# A deep-learning approach for online cell identification and trace extraction in functional two-photon calcium imaging

Luca Sità [1,3✉], Marco Brondi [1,3✉], Pedro Lagomarsino de Leon Roig[1,2], Sebastiano Curreli [1], Mariangela Panniello[1], Dania Vecchia [1] & Tommaso Fellin [1✉]

In vivo two-photon calcium imaging is a powerful approach in neuroscience. However, processing two-photon calcium imaging data is computationally intensive and time-consuming, making online frame-by-frame analysis challenging. This is especially true for large field-of-view (FOV) imaging. Here, we present CITE-On (Cell Identification and Trace Extraction Online), a convolutional neural network-based algorithm for fast automatic cell identification, segmentation, identity tracking, and trace extraction in two-photon calcium imaging data. CITE-On processes thousands of cells online, including during mesoscopic two-photon imaging, and extracts functional measurements from most neurons in the FOV. Applied to publicly available datasets, the offline version of CITE-On achieves performance similar to that of state-of-the-art methods for offline analysis. Moreover, CITE-On generalizes across calcium indicators, brain regions, and acquisition parameters in anesthetized and awake head-fixed mice. CITE-On represents a powerful tool to speed up image analysis and facilitate closed-loop approaches, for example in combined all-optical imaging and manipulation experiments.

[1] Optical Approaches to Brain Function Laboratory, Istituto Italiano di Tecnologia, Genova, Italy. [2] University of Genova, Genova, Italy. [3] These authors contributed equally: Luca Sità, Marco Brondi. ✉email: luca.sita@iit.it; marco.brondi@iit.it; tommaso.fellin@iit.it

Multi-photon imaging in combination with Genetically Encoded Calcium Indicators (GECI) allows the recording of population activity with high spatial resolution in the intact brain in vivo[1–6]. However, multi-photon imaging datasets, in the form of time series (t-series), can be large (0.5 GB to >1 TB) and their processing requires time and computational power (50 GB/h–1 TB/h). More specifically, the precise identification and segmentation of neuronal structures (typically somata) in a given FOV are critical to extract information from raw imaging t-series[6]. This step can be complex because of dense GECI labeling, low signal-to-noise ratio (SNR), presence of motion artifacts, and large number of neurons in the FOV (e.g., in the case of mesoscopic two-photon imaging[7,8]).

Segmentation is typically performed in two ways: (i) manually, based on visual inspection by an expert user and on selection of pixels into regions of interest (ROIs); (ii) automatically, employing supervised or unsupervised methods leveraging on spatial and temporal properties of the fluorescence signal in the t-series[9–23]. Manual segmentation[24,25] is time-consuming and impractical in the case of large datasets and FOVs (e.g., mesoscopic imaging) or when real-time manipulation of experimental conditions is needed[26,27]. State-of-the-art automatic approaches apply pixel correlation[15,16,19], principal/independent component analysis (PCA/ICA)[15,16], constrained non-negative matrix factorization (CNMF)[10,14,17], and deep neural networks (DNN)[10,20–22] to perform FOV segmentation. These approaches are usually applied offline and generally take advantage of both the neuronal spatial footprints and the temporal dynamics of the fluorescence signal associated with the identified spatial footprints. Consequently, their performance benefits from long acquisitions[10,16,23], with highly active cells being more easily segmented than rarely active or inactive ones[10,22]. Moreover, current methods often require the experimenter to set initialization parameters ahead of the segmentation process[10,11,15–17,22]. While most of these parameters are generally easy to adjust (e.g., frame rate and indicator kinetics), some are inaccessible to the user online (e.g., number of expected ROIs in a FOV and spatial constraints on ROI shapes) and must be determined through multiple offline rounds of empirical tuning steps.

The quality of in vivo two-photon calcium imaging is also extremely sensitive to motion artifacts[6]. In particular, the shape and position of imaged cells may change due to motion artifacts correlated with the animal's locomotion, breathing, and heartbeat. In current approaches[10,15–17,22,28], successful neuronal segmentation is typically achieved after correcting for motion artifacts: a process requiring additional time and computational power. The output of the segmentation process is thus a static mask, representing the "average" shape and position of each cell throughout the t-series. This approach is impractical whenever cells should be tracked online on a frame-by-frame basis, for instance, when a neuronal ensemble (i.e., a group of coactive neurons) must be optogenetically manipulated after being identified[29,30]. In fact, neuronal ensembles are dynamic, and different cells may belong to a given ensemble at a certain time point, making it difficult to define a priori the neuronal identities belonging to future ensembles[31]. Finally, downstream of segmentation, the dynamic fluorescence signal from each cell must be extracted and "decontaminated" from background or neuropil signal[6,32]. Different approaches are available to this end[10,11,15–17,32,33], all requiring additional computational time. As a result of all these analytical steps, a total processing time of 30–90 min was reported for most efficient methods when processing FOVs of about 500 μm × 500 μm containing hundreds of cells imaged over tens of thousands of frames[10,11,22]. OnACID[11] and its extended version CaImAn online[10] provide online analysis on streaming data. However, various rounds of offline segmentation with different initialization parameters[10] need to be run in order to obtain a reliable segmentation. Moreover, detection performance is subordinated to the level of cellular activity[10]. These processes introduce a temporal lag that is generally larger than the few minutes required for a single iteration of offline segmentation preprocessing. Altogether, current analytical approaches are: (i) still limited in their ability to perform online analysis, which is necessary for closed-loop experiments; (ii) biased against the identification of rarely active or inactive cells, which could be as informative as more active neurons, for example, in longitudinal all-optical imaging and manipulation approaches; (iii) not validated on large FOVs, such as those generated by mesoscopic two-photon imaging.

Here, we describe CITE-On, a convolutional neural network (CNN) based algorithm trained to perform neuronal somata identification in two-photon imaging recordings, combined with a fast dynamic segmentation and trace extraction pipeline. CITE-On works both offline, after the acquisition is completed, or online and identifies hundreds to thousands of neuronal cell bodies in either modality. Moreover, CITE-On identifies both active and inactive neurons, removing biases towards highly active cells. Finally, CITE-On's light architecture and processing strategy allow fast automatic segmentation, tracking, and trace extraction in mesoscopic two-photon imaging t-series.

## Results

**CITE-On: structure and analysis pipeline.** CITE-On accepts individual frames from two-photon calcium imaging t-series (Fig. 1a) and it includes two main parts: an image detector used to identify neuronal somata based on the publicly available CNN RetinaNet[34] and a custom-built downstream fast analysis pipeline, designed for functional trace extraction. The image detector and the analysis pipeline operate as asynchronous parallel processes in order to provide discrete cell detection update (up to 10 Hz, see text below) and faster than real time functional traces (available at 100 Hz under all experimental conditions tested in this study, see text below). CITE-On required three preprocessing steps ahead of the CNN image detection: (i) frame downsampling; (ii) image upscaling; (iii) replication of the input image into three identical channels (Fig. 1a). The frame downsampling value was set according to the image SNR, while the upscaling factor depended on the ratio between the FOV area and the average area of the neuronal somata (see "Methods" and text below). The length of the frame downsampling window, as well as the value of the upscaling factor, was defined a priori and adjusted according to the data to maximize performance. The three identical images were sent to the CNN for image detection (blue rectangle in Fig. 1a–d). The output of the CNN (yellow highlights in Fig. 1a–d) was a set of boxes tightly surrounding each detected cell soma ("bounding boxes" represented as green squares over the FOVs in Fig. 1a–d). Coordinates and surfaces of each bounding box were used in the analysis pipeline, to generate: (i) cell identity assignment and tracking along the t-series; (ii) frame-by-frame segmentation of neuronal somata; (iii) background subtraction procedure and extraction of neuronal functional trace.

CITE-On works both offline after the acquisition was completed and the whole t-series is available (Fig. 1b), or online, using individual imaging frames as inputs, continuously streamed from the experimental setup during the acquisition of the imaging t-series (Fig. 1c, d). In the offline pipeline, a Fourier-transform approach[35] was used to correct for lateral motion artifacts throughout the t-series (Fig. 1b). The frame downsampling corresponded to the projection of the entire t-series onto its temporal median calculated across all frames (Fig. 1b). Soma detection was then performed once on the preprocessed median

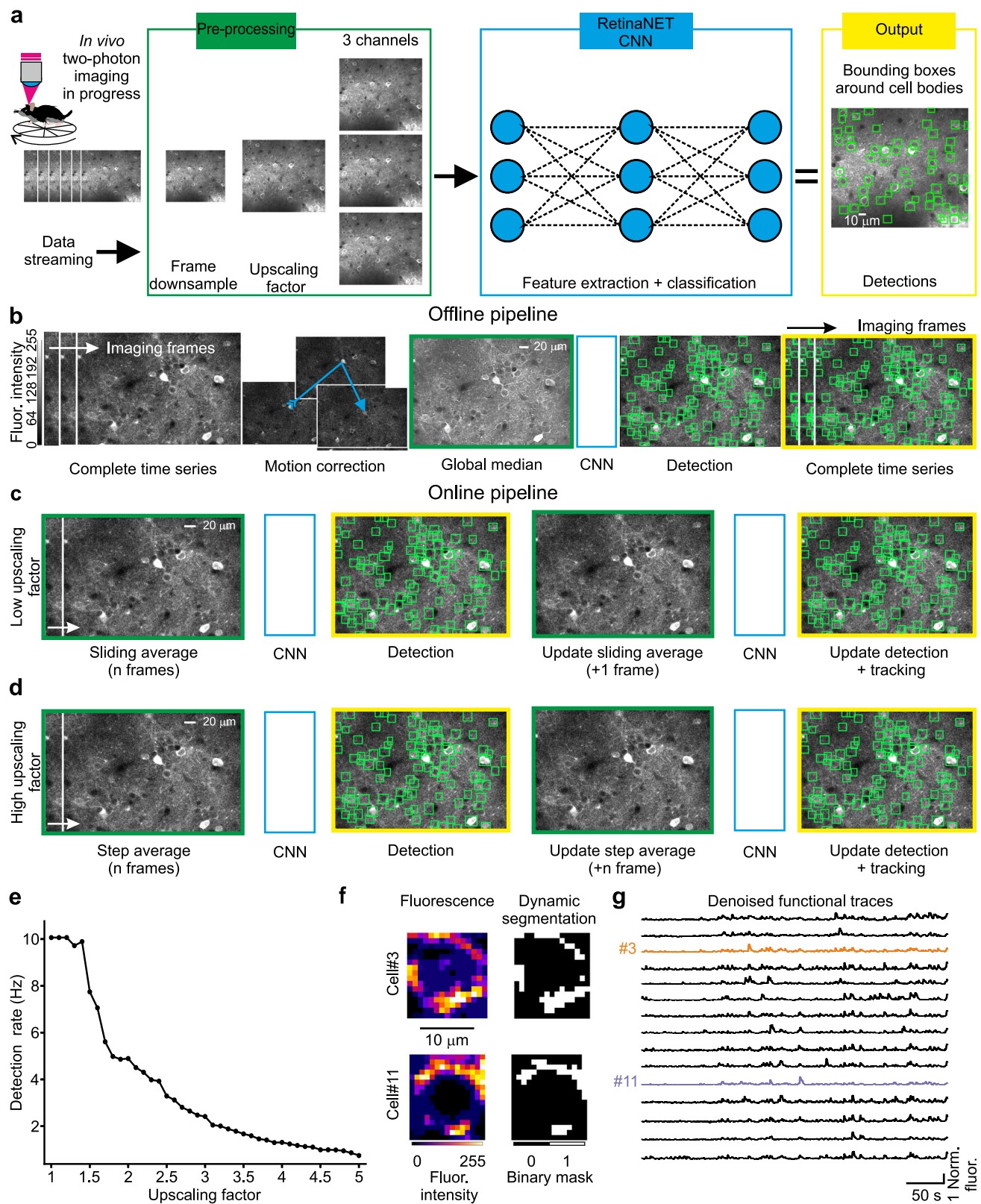

**a** In vivo two-photon imaging in progress · Data streaming · Pre-processing · 3 channels · Frame downsample · Upscaling factor · RetinaNET CNN · Feature extraction + classification · Output · Bounding boxes around cell bodies · 10 μm · Detections

**b** Offline pipeline · Fluor. intensity 0 64 128 192 255 · Imaging frames · Complete time series · Motion correction · Global median · 20 μm · CNN · Detection · Complete time series · Imaging frames

**c** Online pipeline · Low upscaling factor · 20 μm · Sliding average (n frames) · CNN · Detection · Update sliding average (+1 frame) · CNN · Update detection + tracking

**d** High upscaling factor · 20 μm · Step average (n frames) · CNN · Detection · Update step average (+n frame) · CNN · Update detection + tracking

**e** Detection rate (Hz) vs Upscaling factor

**f** Fluorescence · Dynamic segmentation · Cell#3 · Cell#11 · 10 μm · 0 255 Fluor. intensity · 0 1 Binary mask

**g** Denoised functional traces · #3 · #11 · 50 s · 1 Norm. fluor.

image (detection in Fig. 1b), and bounding boxes were generated for each frame of the t-series (Fig. 1b, yellow highlights). Each bounding box was associated with a score, representing network confidence in cell detection. We defined the intersection over union (IoU) for two identified bounding boxes as the proportion of the overlapping area between two boxes out of the sum of the areas of the two boxes. Bounding boxes with intersection over

union (IoU) <20% were considered as separate neuronal identities. When IoU of two bounding boxes was >20%, the bounding box with the highest score was retained.

In the online pipeline, no motion correction was performed, and the user selected between two downsampling strategies depending on the SNR of the data and on the required upscaling factor in the preprocessing step. In the case of relatively high SNR

**Fig. 1 Structure and analysis pipeline of CITE-On. a** Schematic of the image detection process in CITE-On. During ongoing two-photon imaging acquisitions, individual frames are transferred to CITE-On as they are completed (left). A preprocessing step (green rectangle) is required ahead of image detection, including frame downsampling, image upscaling, and triplication of the upscaled image. The result of the preprocessing is then used as input to the CNN (blue rectangle). The CNN output is the detection of neuronal somata in the form of bounding boxes (green squares, greyscale image on the right). **b** CITE-On offline pipeline starts with the complete t-series and the correction of motion artifacts (blue arrow, motion correction). Frame downsampling is performed by computing the global median projection of the t-series. The upscaled and triplicated global median (green) is fed to the CNN (blue), a single detection is performed, and the bounding boxes (detection, green squares) are projected onto each frame of the complete t-series (yellow). The color scale shown on the left in this panel applies to all grayscale images in this figure. **c** In the online pipeline, for data requiring low upscaling factors, a sliding average projection of the first $n$ frames of the ongoing t-series is calculated in the frame downsampling preprocessing step (green). This image is upscaled and triplicated, processed by the CNN (blue), producing the first detection (yellow). As the next frame of the t-series is acquired, a new sliding average is computed, again on $n$ frames, but starting from the second frame of the acquisition and including the $n + 1^{th}$ one (green). The CNN processes this image (blue), updating the detections and starting the tracking system (yellow). **d** For data requiring high upscaling factors, the pipeline is similar to that in **c**, but instead of a sliding average, a step average is calculated on $n$ frames as the frame downsampling preprocessing step (green). Detections are updated every $n$ new frames. **e** Detection rates as a function of the magnitude of the upscaling factor. The maximum detection rate is 10 Hz for upscaling factor between 1 and 1.5. **f** Representative average fluorescence of pixels inside the bounding box relative to two cells (cell #3 and cell #11), calculated in a single frame of the LIV dataset (GCaMP6s, pseudocolor, left). Associated dynamic segmentation mask in the same frame (binary mask, right). **g** Functional traces from $N = 15$ representative cells extracted with online CITE-On pipeline. Traces from cells displayed in **f** are shown in orange and purple.

and low upscaling factors (Fig. 1c), a sliding average (arithmetic mean) was calculated on the first $n$ frames of the t-series and updated with every new individual frame generated by the microscope. Neuronal detections were updated for each imaging frame starting from the $n + 1$th frame. When the SNR was relatively low and the upscaling factor large (Fig. 1d), a step average approach was performed, where the input for the image detector was the average projection of blocks of $n$ frames (i.e., arithmetic mean of the $n$ considered frames along the temporal axis). Additional $n$ frames were thus required for generating the next step average projection, and the detections were updated every $n$ frames. In the sliding-average mode, the maximum detection rate decreased with the upscaling factor, with a peak rate of 10 Hz with upscaling factor = 1 (Fig. 1e). Active detections (i.e., detections in the current sliding average or step average) and past detections (i.e., detections in any previous sliding or step average) were continuously tracked and updated (Fig. 1c, d, update detection and tracking). Specifically, active detections were compared with past detections at each step of the detection update and a new identity was added (and included in the tracking system) every time the surface of an actively detected bounding box had IoU <20% with any of the previously identified boxes. Bounding boxes from active detections with IoU > 20% with those of past detections did not change identities of previously detected boxes, but their positions and shapes were updated according to the position in the most recent detection step. All past detections without updates were retained in the tracking system in the form of their last active detection for the remaining part of the t-series (Supplementary Movie 1).

For both the online and offline pipelines (Fig. 1f, g), bounding boxes were used to generate a dynamic segmentation of the t-series and to identify ROIs. The distribution of fluorescence values inside each bounding box was computed at each frame (Fig. 1f, left). Only pixels with values between the 80th and the 95th percentile of the box's fluorescence distribution were assigned to the ROI corresponding to the cell soma (white pixels of the binary mask in Fig. 1f, right). This range of values was chosen in order to base trace extraction on the pixels with highest intensity (>80th percentile), while avoiding pixels close to saturation (<95th percentile). Since pixel assignment to cell somata in each individual box was updated at each frame, the resulting dynamic segmentation was updated online for every new frame. All the FOV pixels that were not included in any bounding box were assigned to a global background ROI, similarly to ref. [36]. The fluorescence intensity of all pixels belonging to the global background ROI was averaged at each frame to obtain the background signal ($bg$). Moreover, at each frame, the $bg$ was subtracted from the fluorescence of each segmented neuronal ROI, generating $bg$-corrected fluorescence traces (Fig. 1g). Since shape, number, and position of bounding boxes changed as the t-series progressed (according to active detections and tracking), the pixels assigned to $bg$ also changed in number and identity across frames. Identity tracking, segmentation, and functional trace extraction required ≤10 ms per frame (either offline or online).

**Training of the image detector and ground truth generation.** The ResNet50 Feature Extractor CNN incorporated in CITE-On was not originally developed for detecting neuronal somata, but rather for the analysis of natural images, and it was trained on >1 million RGB images across 80 classes (http://www.image-net.org/). We decided to use a transfer learning strategy[37] to adapt this efficient detection architecture to the identification of neuronal somata (i.e., a single class) in greyscale two-photon images. This choice was dictated by the fact that available two-photon calcium imaging datasets (http://neurofinder.codeneuro.org/, http://help.brain-map.org/display/observatory/Data+-+Visual+Coding) are far too small for an ab-initio CNN training. Moreover, they are too homogeneous in terms of calcium indicators used, FOV dimensions, cell density, acquisition frame period, SNR, and background signal contamination[22], making them suboptimal even for a transfer learning strategy. For example, no publicly available large dataset comprises imaging data collected using red-shifted GECIs, such as jRCaMP1a. We thus decided to use a dedicated dataset including only t-series acquired in our laboratory for training and internal validation. In this way, we employed publicly available datasets to test CITE-On performance and its generalization capability on never-before-seen data. The dedicated dataset included 197 t-series from 28 mice acquired using different acquisition parameters (see "Methods"). More specifically, we included 131 t-series from layer IV neurons of the somatosensory cortex expressing either GCaMP6f, GCaMP6s, or GCaMP7f (globally indicated as "LIV") and 66 t-series from the CA1 pyramidal neurons of the hippocampus expressing both jRCaMP1a and GCaMP6f (indicated as "CA1 jRCaMP1a" and "CA1 GCaMP6f", respectively). We included t-series with heterogeneous median fluorescence and SNR in order to reduce potential biases toward bright cells during the training process, while avoiding large differences between groups of data that could have generated better performance on specific subsets of t-series (Supplementary Fig. 1a, b).

To obtain a consensus Ground Truth (GT) annotation of the t-series used for training and validation of the CNN, two human graders manually annotated all t-series, defining the tightest rectangular box fitting each visible cell in each FOV. Manual GT annotation was preferred to automatic segmentation for two main reasons: (i) available automatic segmentation approaches rely on both functional (i.e., fluorescence signal dynamics across frames) and morphological features[9–23], while we wanted the GT annotation to be exclusively based on morphological features (see below); (ii) manual annotation is still frequently considered more accurate than automatic methods[10,22]. Initially, manual annotation on single frames by two graders produced only few neuronal identities because cells were only visible in a minority of frames (Supplementary Fig. 1c, Supplementary Movie 2). This could be due to the low basal emission of some of the indicators used (e.g., GCaMP6f), to the variable expression level of the calcium indicator across cells, and to the sparse activity profile of the imaged cells. In order to increase the visibility of neurons, we created high contrast single images representative of each t-series. To this end, we first corrected each t-series for lateral motion artifacts and then collapsed each acquisition onto its median projection (Supplementary Fig. 1d). These images were sharpened (Supplementary Fig. 1e) and gamma corrected. Brightness and contrast were adjusted in order to obtain a distribution of intensity values spanning the whole bit range. The sharpened images, named enhanced median projections (EMPs) (Supplementary Fig. 1f, g), were used for manual annotation (Supplementary Fig. 1h, Supplementary Movie 3). In training and validation datasets (LIV$_{train}$, CA1$_{train}$, LIV$_{test}$, and CA1$_{test}$ datasets, $N = 197$ t-series), grader #1 annotated 14425 boxes, while grader 2 annotated 12912 (Supplementary Table 1). The bounding boxes produced by grader #1 and grader #2 and their superposition in different experimental preparations are shown in Supplementary Fig. 2. We used mean average precision (mAP), Precision, Recall, and F-1 score as metrics for performance quantification (see "Methods" for definitions). Annotations from the two graders largely overlapped (mAP, $0.77 \pm 0.08$; F-1 score, $0.93 \pm 0.02$; precision: $0.98 \pm 0.01$; recall: $0.88 \pm 0.12$, $N = 197$ EMPs, see also Supplementary Table 2). Given the high similarity of the independent annotations provided by the two graders, we defined the consensus GT for our entire dataset as the GT shared between the two graders (see "Methods"). Given that our dataset contained partially overlapping FOVs and more than one t-series acquired on the same FOV, the dataset was manually split into training (160 t-series) and validation (37 t-series) subsets. To avoid data leakage and to decrease overfitting[38], the t-series relative to a given FOV were first grouped together and then included only in the training dataset or the validation dataset. We trained the CITE-On image detector on the training dataset and evaluated its performance on the validation dataset. The EMPs of all the t-series used for CITE-On training, together with the corresponding consensus GT annotations (green boxes) are shown in Supplementary Movie 4.

**Performance of the image detector**. We trained the CITE-On image detector on our consensus GT annotations achieving the best performance after 17 training epochs (mAP: 0.79 on the validation dataset). We first evaluated CITE-On performance using the offline pipeline on the validation dataset. A representative CITE-On output for a CA1 jRCaMP1a, a CA1 GCaMP6f, and a LIV t-series is shown in Fig. 2a–d and Fig. 2e, respectively. For the whole validation dataset, Precision, Recall, and F-1 score are reported in Fig. 2f and Supplementary Table 3. The EMPs of all the t-series used for CITE-On validation and the corresponding consensus GT annotations (green) are shown in Supplementary Movie 5.

We then used the online pipeline and calculated the F-1 of CITE-On detections on the motion corrected validation dataset, which was used as input to CITE-On at the actual frame rate occurring during acquisition. Our validation data required an upscaling factor of 1, compatible with a maximum detection rate of 10 Hz, while acquisition frame rates varied between 1.5 Hz for LIV and 3 Hz for CA1 acquisitions. We empirically explored the effect of frame downsampling on detection performance using the sliding average approach with different numbers of frames (n). Our aim was to maximize F-1 and score threshold for detections while minimizing $n$. F-1 increased with $n$ between 1 and 20 frames. At this latter value, a local maximum in F-1 was observed (Fig. 3a). Using the sliding average, an initial delay of 6.6 s for LIV data and 14 s for CA1 data was necessary before CITE-On processed each frame in real time at a detection rate of 10 Hz. F-1 values calculated on sliding averages of 20 frames are reported in Fig. 3a. Detections (green boxes) are displayed together with GT annotations (magenta boxes) for both the CA1 jRCaMP1a (Fig. 3b, left) and the CA1 GCaMP6f channels (Fig. 3b, middle). The superimposition of the detections from both channels is shown in Fig. 3b (right). Similarly, in Fig. 3c, we show GT and online detections (magenta and green boxes, respectively) for a representative LIV (GCaMP6f) t-series.

To quantify the impact of motion artifacts on online detection accuracy, we calculated the F-1 score on validation t-series that were not corrected for motion artifacts (Supplementary Table 4). To this end, we translated the GT annotations for each frame according to the shift vectors produced by the motion correction algorithm implemented when building the relative EMPs (see "Methods"). We observed no significant difference between the F-1 obtained on motion corrected and non-corrected validation t-series (Fig. 3d), suggesting that the motion correction step was not necessary to achieve higher performance with our approach. The distribution of motion displacements is shown in Supplementary Fig. 3. Figure 3e shows that the F-1 score increased during online processing and became stable within ~1/3 of the total length of the processed t-series.

To explore the performance of CITE-On in the presence of larger motion artifacts, we created a set of t-series ($N = 90$) with artificial motion artifacts ranging between 4 μm and 20 μm. The artifacts were obtained using frame-by-frame lateral drifts from one ABO motion-corrected t-series (ABO #501271265). The consensus GT annotation was translated frame-by-frame according to the imposed artificial drift. Using this strategy, we were able to study CITE-On performance (Precision, Recall, and F-1 score) as a function of the amplitude and duration of the artificial shift (Supplementary Fig. 4). We observed that the F-1 score was highest for lateral displacements ≤8 μm for all the values of motion artifact duration that we considered. In contrast, CITE-On performance decreased for displacements >8 μm, and this drop was larger for faster artifacts (<2 s).

**Online data processing**. We developed a fast method to dynamically segment each detected cell based on the corresponding bounding box, relying on the instantaneous (i.e., frame-wise) fluorescence statistics of the pixels inside each box (Fig. 4, Supplementary Movie 6). Specifically, the fluorescence intensity distribution of the pixels inside each bounding box was first computed frame-wise. Pixels with fluorescence values between the 80th and 95th percentile of the distribution were then selected as belonging to neuronal somata. The values of selected pixels were averaged (arithmetic mean), and the resulting fluorescence trace was "denoised" by subtracting, at each frame, the *bg* signal. This simple method was computationally light, an important requirement to achieve fast frame-by-frame data processing (trace

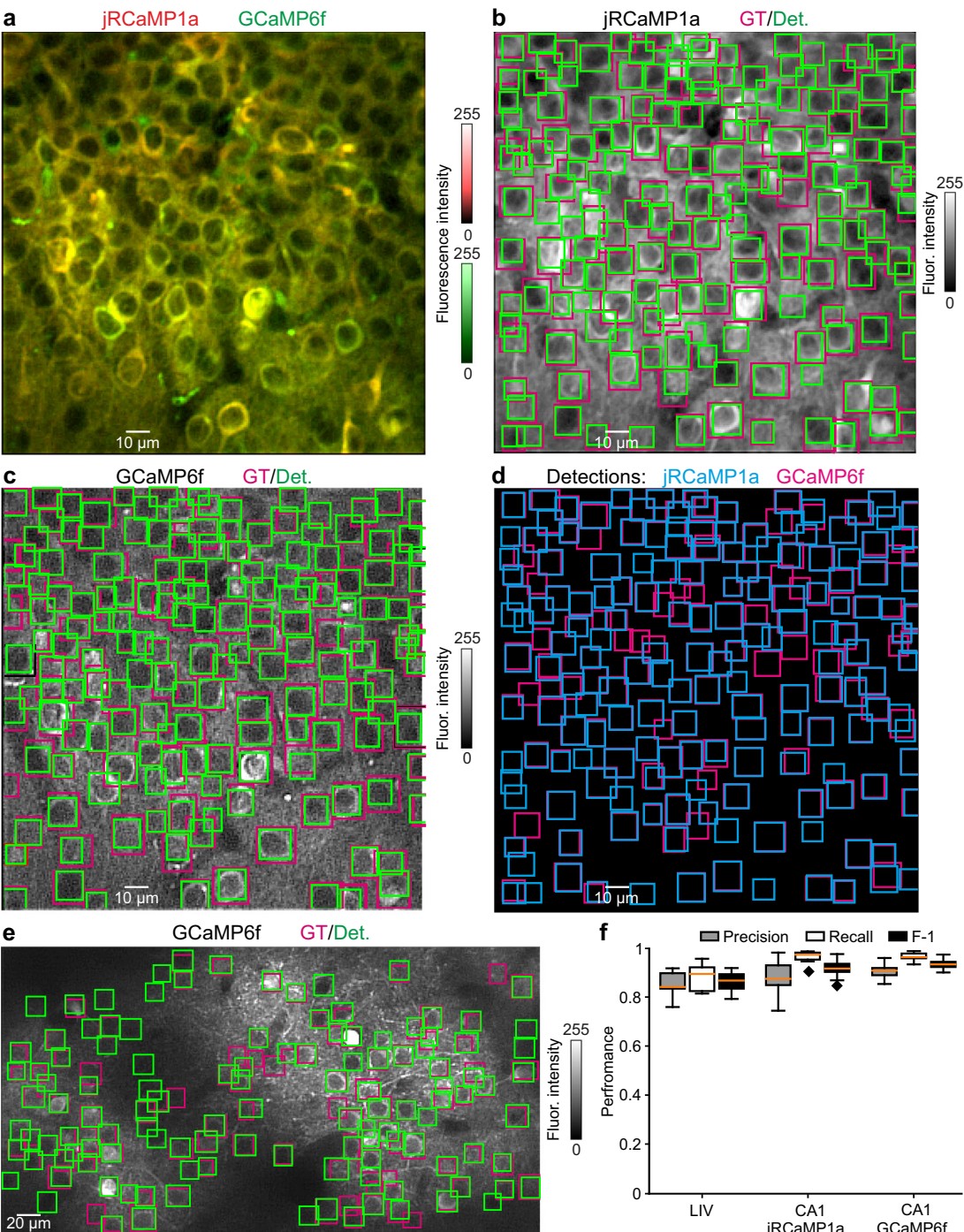

**Fig. 2 CITE-On offline cell detection performance. a** Representative fluorescence median projection showing jRCaMP1a (red) and GCaMP6f (green) expressing CA1 neurons. **b** Ground truth (GT, magenta) and CITE-On detections (Det, green) for the jRCaMP1a channel of the image shown in **a**. **c** Same as in (b), but for the GCaMP6f channel. **d** Superposition of CITE-On detections on jRCaMP1a (cyan) and GCaMP6f (magenta) channels. **e** Representative median projection from the LIV dataset with GT (magenta) and CITE-On detections (green). **f** Boxplots showing performance as Precision (gray), Recall (white), and F-1 score (black) obtained with the offline CITE-On pipeline on the validation t-series of the LIV ($N = 13$), CA1 jRCaMP1a ($N = 12$), and CA1 GCaMP6f ($N = 12$) datasets. The orange line in all boxplots indicates the median, the bounds of the boxes represent the 75th and 25th percentiles (i.e., the interquartile range (IQR)), and the whiskers correspond to the highest value or lowest value of the distribution. If the lowest or highest values are outliers (i.e., >1.5 *IQR from the bounds of the boxes) the whiskers correspond to 1.5 *IQR. Outliers are represented as black diamonds.

extraction rate, 100 Hz). Bounding boxes detected by CITE-On on a representative LIV t-series and a representative CA1 t-series are shown in Fig. 4a. Representative fluorescence traces extracted by CITE-On on the two t-series are displayed in Fig. 4b, c. Figure 4d, e shows the cross-correlation matrix (lower-left triangle) and the dendrogram analysis (upper-right triangle) of all the

identified cells before (Fig. 4d, e left) and after (Fig. 4d, e right) *bg* subtraction. We found that the dendrogram sorting showed blocks with different cross-correlation values for various subgroups of cells. Cross-correlation matrices (see "Methods") before *bg* subtraction displayed substantially larger values of average correlation compared to cross-correlation matrices after *bg*

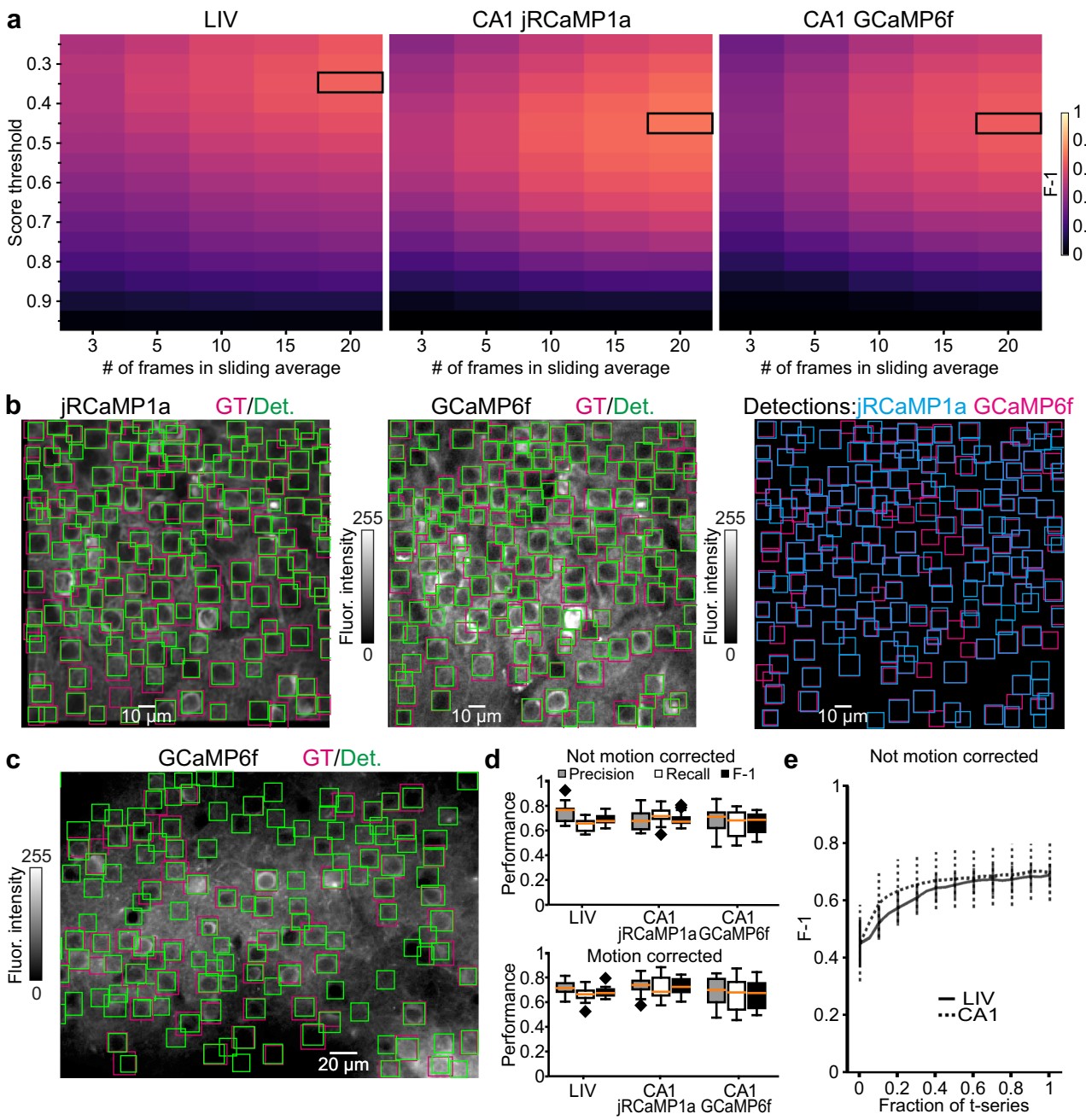

**Fig. 3 CITE-On online cell detection performance. a** Best parameter search for frame downsampling: F-1 score (pseudocolor) as a function of score threshold (vertical axis) and number of frames in the sliding average (horizontal axis) for LIV (left), CA1 jRCaMP1a (middle), and CA1 GCaMP6f (right). The maximal F-1 is indicated with the black rectangle. **b** Fluorescence median projections of one representative FOV for CA1 jRCaMP1a (left) and one FOV for CA1 GCaMP6f (middle). GT (magenta) and online detections at the end of the t-series (Det, green) are also shown. In the rightmost panel, the online detections of jRCaMP1a (cyan) and GCaMP6f (magenta) at the end of the t-series are shown. **c** Same as in **b** but for a representative LIV t-series. **d** Top: boxplots showing online performance as Precision (gray), Recall (white), and F-1 (black) for all t-series in the validation datasets (LIV, $N = 13$; CA1 jRCaMP1a, $N = 12$; CA1 GCaMP6f, $N = 12$). No motion correction was performed. Bottom: same as top, but for the motion-corrected t-series. The orange line in all boxplots (in top and bottom panels) is the median, the bounds of the boxes are the 75th and 25th percentiles (i.e., the interquartile range (IQR)), and the whiskers correspond to the highest value or lowest value of the distribution. If the lowest or highest values are outliers (i.e., >1.5 *IQR from the bounds of the boxes), the whiskers correspond to 1.5 *IQR. Outliers are represented as black diamonds. Results of Kolmogorov–Smirnov test for performance in not motion-corrected t-series vs. motion-corrected t-series from LIV: $p = 0.54$ for F-1, $p = 0.15$ for Precision, $p = 0.38$ for Recall, $N = 13$ t-series. Results of two-sided Kolmogorov–Smirnov test for performance in not motion-corrected t-series vs. motion-corrected t-series from CA1 jRCaMP1a: $p = 0.16$ for F-1, $p = 0.20$ for Precision, $p = 0.20$ for Recall, $N = 12$ t-series. Results of Kolmogorov–Smirnov test for performance in not motion corrected vs. motion-corrected t-series from CA1 GCaMP6f: $p = 0.18$ for F-1, $p = 0.28$ for Precision, $p = 0.22$ for Recall, $N = 12$ t-series. **e** F-1 values as a function of the fraction of the total length of the t-series for not-motion corrected data ($N = 13$ t-series for LIV, $N = 24$ t-series for CA1, including $N = 12$ t-series for CA1 jRCaMP1a and $N = 12$ t-series for CA1 GCaMP6f t-series).

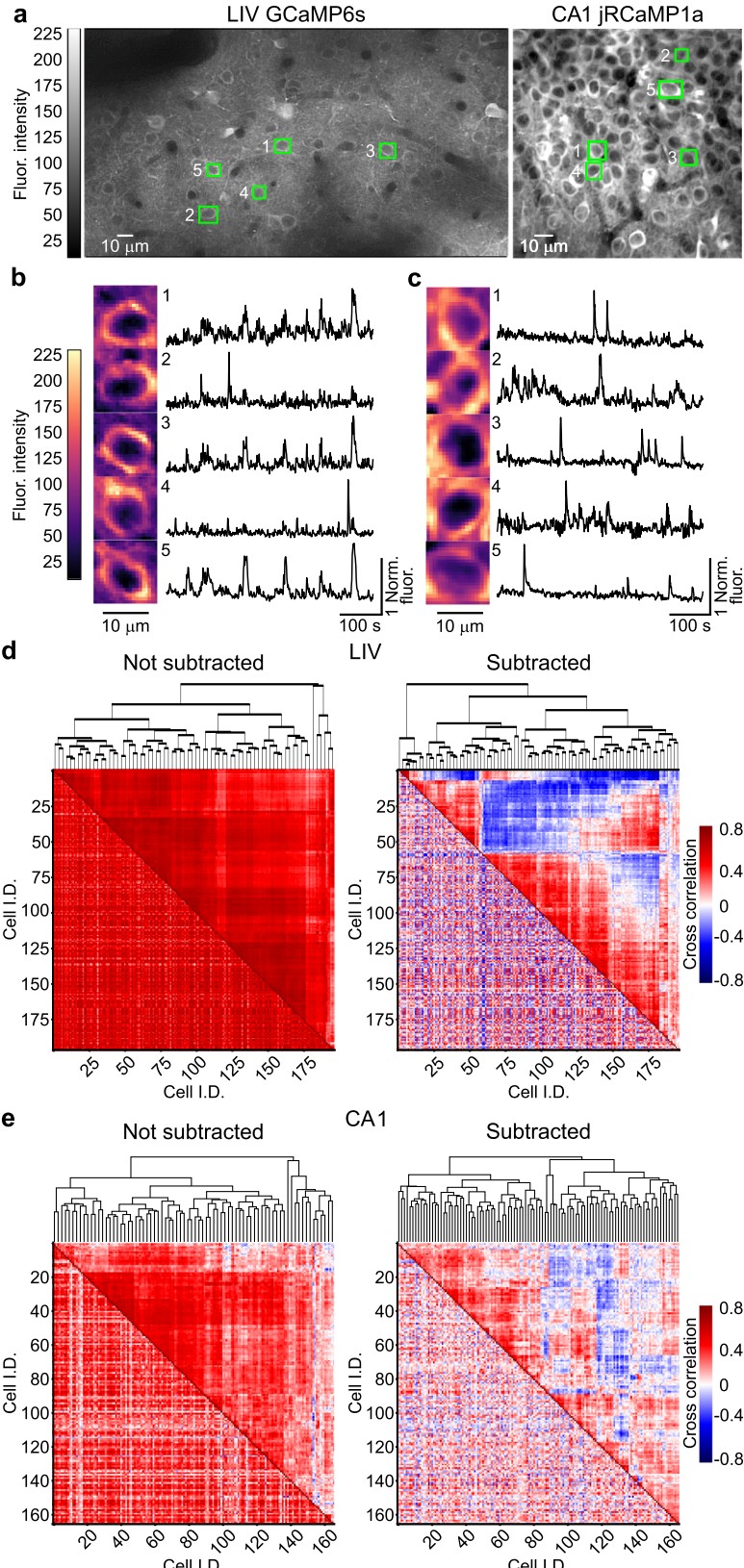

**Fig. 4 Fast extraction of fluorescence traces using CITE-On. a** Fluorescence median projection showing representative FOVs from the LIV GCaMP6s (left) and the CA1 jRCaMP1a (right) datasets. True positive bounding boxes for five CITE-On identified cells in each FOV are shown in green. **b** Left: the five cells indicated in the LIV t-series displayed in **a** are shown at an expanded spatial scale. Right: fluorescence traces for the cells shown in the left panel were thresholded and background subtracted (see "Methods"). **c** Same as in **b** but for the CA1 t-series in **a**. **d** Lower-left triangle: cross-correlation matrix for all functional traces extracted from true positive detection in the LIV GCaMP6s t-series displayed in **a**. Upper-right triangle: corresponding dendrogram sorting. The left matrix shows signals before background subtraction. The right matrix after background subtraction. **e** Same as in **d**, but for the CA1 jRCaMP1a t-series shown in **a**.

subtraction (Fig. 4d, $0.60 \pm 0.14$ before subtraction vs. $0.05 \pm 0.27$ after subtraction, $p < 1\mathrm{E}{-}9$, Wilcoxon signed-rank test, $N = 197$. Figure 4e, $0.33 \pm 0.18$ before subtraction vs. $0.08 \pm 0.19$ after subtraction, $p < 1\mathrm{E}{-}9$, Wilcoxon signed-rank test, $N = 166$). Thus, *bg* subtraction reduced the overall pairwise correlations, as expected when a signal that is common to all neurons is subtracted. Therefore, by reducing the average value of pairwise correlations, *bg* subtraction allowed the identification of neuronal pairs with low cross-correlation values (close to zero or <0), which would be difficult to identify otherwise. The dendrogram sorting of neuronal identities (upper triangles of the cross-correlation matrices in Fig. 4d, e) was based on the relative distance of their cross-correlation values (see "Methods"). Comparing dendrograms before *bg* subtraction with dendrograms after *bg* subtraction highlights how subtracting the *bg* is instrumental to identify spatially localized clusters of cells with distinctive patterns of cross-correlations (both positive and negative), which may be suggestive of functional neuronal ensembles.

We compared traces extracted by CITE-On with those extracted by CaImAn, a state-of-the-art method based on CNMF[10]. Both programs were used in the offline modality and on motion-corrected t-series. We used the bounding boxes generated offline by CITE-On to build binary masks that were used as seeds to initialize the seeded-CNMF algorithm[10]. The spatial components of the CNMF were non-zero only inside the bounding boxes identified by CITE-On. Therefore, the detected factors from seeded-CNMF had one-to-one correspondence with the detected boxes from CITE-On. Using this strategy, we obtained fluorescent traces extracted by CaImAn from putative cells detected in the same locations as those detected by CITE-On, allowing for a one-to-one trace comparison between algorithms. We first observed very high pairwise cross-correlations between the *bg* traces extracted with the two methods (Supplementary Fig. 5a). We then asked how the *bg* signal calculated over the whole FOV (*bg*) correlated with the 'local *bg*', that is, the background activity calculated for each cell from the pixels in the immediate surroundings of the relative bounding box (see "Methods"). The average correlation between *bg* and local *bg* traces was high (Supplementary Fig. 5b). Given the high correlation values observed between *bg* and local *bg* and the lower computational cost of *bg*, we decided to implement the *bg* method only. We then tested how the *bg*-subtracted functional traces calculated by CITE-On compared to those extracted with seeded-CNMF. We did so after denoising traces by averaging fluorescence values across consecutive frames in each t-series (see "Methods"). We again observed high correlation values (Supplementary Fig. 5c–e).

Although the average correlation values for the cells extracted with the two methods were high, some neuronal pairs showed lower correlations. We asked whether the low correlation values emerged from pairs of cells with low SNR. We found that pairwise correlation values for traces extracted with the two methods increased with the SNR of the corresponding neuronal traces (Supplementary Fig. 5f), indicating that indeed the functional traces obtained with the two methods were more similar when the trace SNR was high.

**Offline performance on never-before-seen recordings**. To test the robustness of our image detection approach and its ability to generalize across experimental conditions, we tested CITE-On on three additional datasets, which were not used during the training and validation phases. The three datasets were: the Allen Brain Observatory repository (ABO, 19 t-series divided into 9 superficial, $\mathrm{ABO_{sup}}$, t-series, acquired in visual cortex at depth 175 µm,

and 10 deep, $\mathrm{ABO_{deep}}$, t-series, acquired in visual cortex at depth 275 µm), the Neurofinder Challenge dataset (28 t-series, divided into 19 t-series, $\mathrm{NF_{train}}$, 9 t-series, $\mathrm{NF_{test}}$, from different preparations at different depths), and a dataset acquired in our laboratory using GRIN-based endoscopic two-photon imaging of the ventral posteromedial thalamic nucleus (VPM, 9 t-series). The datasets were first manually annotated de novo to obtain the consensus GT annotation (Supplementary Fig. 6, Supplementary Table 1–2). Because the ratios between FOV and cell surface were variable across the ABO, NF, and VPM datasets and different from our validation dataset, we optimized the upscaling factor and used the one that maximized the F-1 score (Fig. 5a–e). The offline performance (defined in terms of Precision, Recall, and F-1 score) obtained using optimized upscaling factors for each dataset is shown in Fig. 5g and Supplementary Table 3. While CITE-On performance was high for most datasets, we observed lower performance for the $\mathrm{NF_{train}}$ dataset, in agreement with the observation that $\mathrm{NF_{train}}$ has among the lowest SNR of all considered datasets and that CITE-On performance decreases with decreasing SNR (Supplementary Fig. 5f).

We compared the offline detection performance of CITE-On (Supplementary Table 3) with state-of-the-art alternative segmentation approaches such as STNeuroNET[22], CaImAn online[10], CaImAn batch[10], Suite2P[16], HNCcorr[19], UNet2DS[23] on the ABO and NF datasets provided in ref. [22]. To this aim, we did not run the alternative approaches ourselves. Rather, we used the data reported in ref. [22] for all of them. To carry out this comparison, CITE-On was run on the GT annotations provided in ref. [22]. Detection performance (Precision, Recall, and F-1 score) is similar or better for CITE-On (both online and offline) when compared with detection performance reported on the same GT for CaImAn (CaImAn online and CaImAn batch) and most other algorithms (Fig. 6). Moreover, CITE-On performance (online and offline) using our consensus GT tends to be higher than CITE-On performance computed using the GT provided in ref. [22]. The *bg* signals calculated using seeded CaImAn and CITE-On presented high cross-correlation values for all three datasets (Supplementary Fig. 7a). Similarly, high cross-correlations were measured between local *bg* and *bg* signals computed with CITE-On (Supplementary Fig. 7b), as well as between CITE-On and seeded CaImAn extracted functional traces after bg subtraction and smoothing (Supplementary Fig. 7c–e). As in our previous characterization on the validation dataset, cross-correlation values of neuronal traces extracted with CITE-On and CaImAn increased with SNR (Supplementary Fig. 7f). This relationship between cross-correlation and SNR can be ascribed to the larger amount of signal extracted by both CaImAn and CITE-On when the SNR is large.

We compared true positive detections obtained with CITE-On to ABO true positive detections (see "Methods") and to STNeuroNET true positive detections (Fig. 7a–c). Examples of cells identified by CITE-On and present in the true positive detections from both the ABO dataset and the STNeuroNET study are reported in Fig. 7d, together with their relative background-subtracted traces extracted by CITE-On. Examples of cells identified only by CITE-On are reported in Fig. 7e. We analyzed the functional traces of all the CITE-On true positives (i.e., including cells detected in ABO and STNeuroNET) after background subtraction (Fig. 7f). Similar to what is described for our validation dataset, the dendrogram sorting showed blocks with different cross-correlation values for various subgroups of cells. On average, the number of CITE-On detected cells exceeded the number of identities available in public repositories (Supplementary Table 5).

Since state-of-the-art segmentation methods have a bias against inactive cells, we investigated whether CITE-On only cells were inactive. We quantified the number of calcium events

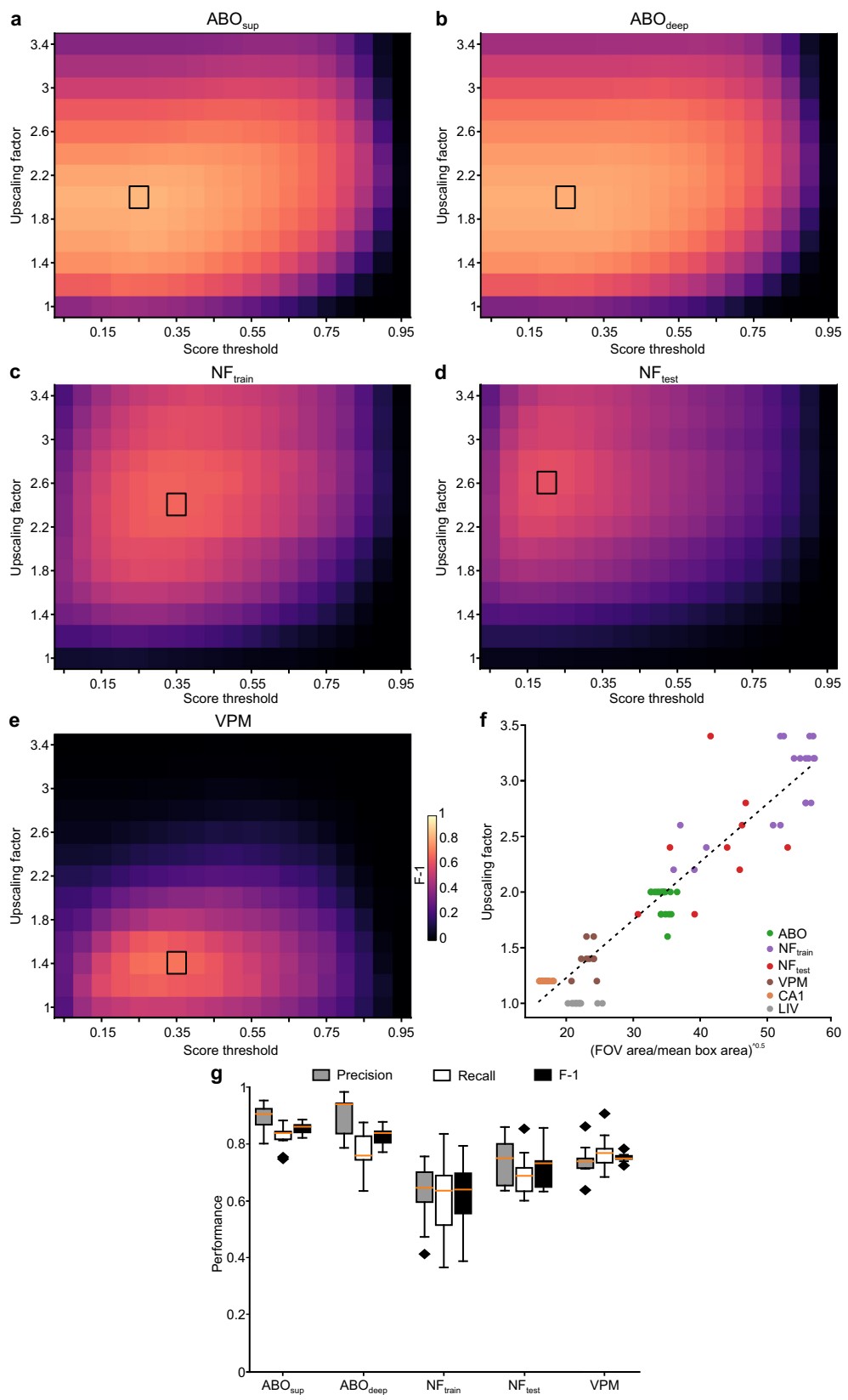

per detection in the ABO true positives (ABO TP), STNeuroNET true positives (STNuroNET TP), CITE-On true positives (CITE-On TP), and CITE-On only true positives (CITE-On only, Supplementary Fig. 8). The number of cells with few detected calcium events was larger for CITE-On TP (Supplementary Fig. 8). Moreover, the distribution of calcium events per detection

for CITE-On only cells (Supplementary Fig. 8d) shows that some identities were silent (as expected), but also that a large fraction of them displayed detectable activity (91% displayed at least one calcium event and 69% showed at least ten calcium events in the whole ABO dataset). The number of detected calcium events with CITE-On (all detections, true positives, false positives, and CITE-

**Fig. 5 CITE-On offline cell detection performance on never-before-seen data. a–e** Best parameter search for frame upscaling factor: F-1 score (pseudocolor) as a function of upscaling factor and score threshold for the Allen Brain Observatory (ABO) $ABO_{sup}$ (**a**), $ABO_{deep}$ (**b**), and Neurofinder (NF) $NF_{train}$ (**c**), $NF_{test}$ (**d**) and VPM (**e**) datasets. The maximal F-1 is indicated by the black rectangle. The pseudocolor scale in (**e**) applies to **a–d**. **f** Optimized upscaling factor as a function of the ratio between the FOV area and the bounding box area for all acquisitions in the validation datasets. Each dot represents a single t-series (green, $ABO_{sup}$ and $ABO_{deep}$ together, $N = 19$; purple, $NF_{train}$, $N = 19$; red, $NF_{test}$, $N = 9$; brown, VPM, $N = 9$; orange, CA1 jRCaMP1a and GCaMP6f together, $N = 24$; gray, LIV, $N = 13$). The dotted line represents the linear fit of the data ($R^2 = 0.942$). **g** Boxplots showing performance as Precision (gray), Recall (white), and F-1 (black) for all t-series in the $ABO_{sup}$ ($N = 9$), $ABO_{deep}$ ($N = 10$), $NF_{train}$ ($N = 19$), $NF_{test}$ ($N = 9$), and VPM ($N = 9$) datasets. The orange line in all boxplots is the median, the bounds of the boxes are the 75th and 25th percentiles (i.e., the interquartile range (IQR)), and the whiskers correspond to the highest value or lowest value of the distribution. If the lowest or highest values are outliers (i.e., >1.5 *IQR from the bounds of the boxes) the whiskers correspond to 1.5 *IQR. Outliers are represented as black diamonds.

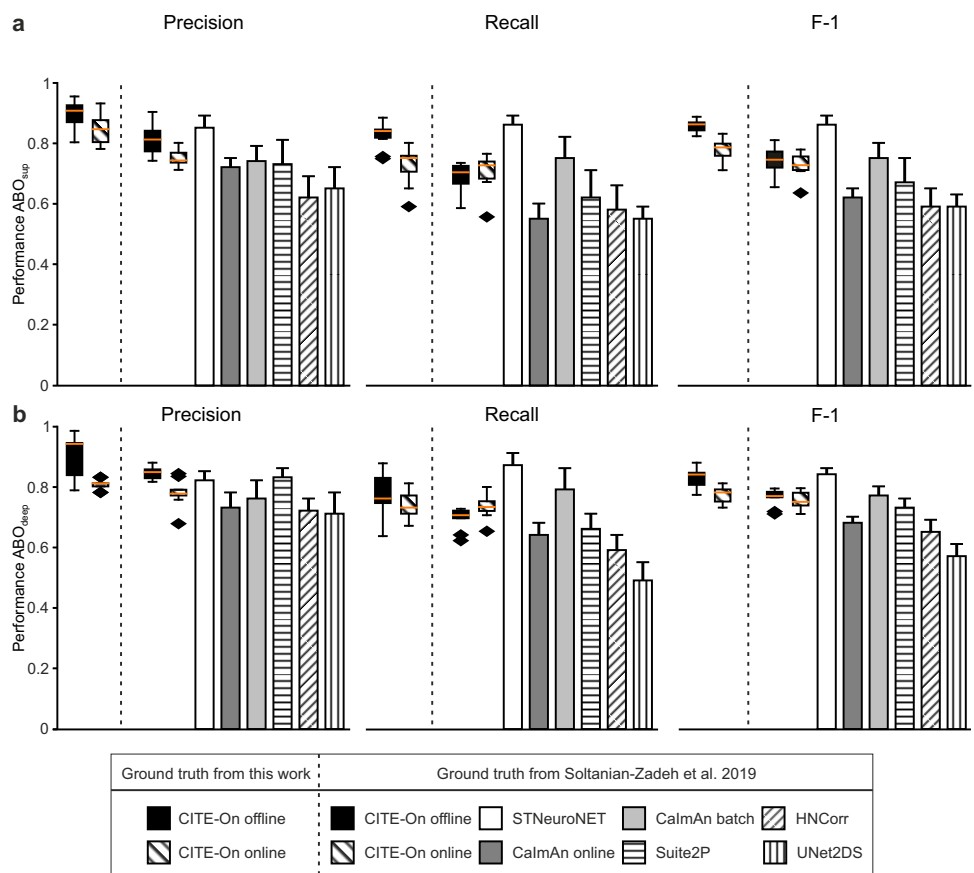

**Fig. 6 CITE-On cell detection performance compared to state-of-the-art methods. a, b** Precision (left), Recall (middle), and F-1 score (right) in cell detection for CITE-On and other methods (for other methods plotted data are reported from ref. [22]). CITE-On performance is evaluated on $ABO_{sup}$ (**a**, $N = 9$ t-series), $ABO_{deep}$ (**b**, $N = 10$ t-series). CITE-On performance is calculated using our consensus ground truth ("ground truth from this work", left of the vertical dotted line) or using the ground truth reported in Soltanian-Zadeh et al.[22] (right of the vertical dotted line). CITE-On performance is shown in bloxplots, the performance of other methods is shown as mean ± s.d from ref. [22]. The orange line in all boxplots is the median, the bounds of the boxes are the 75th and 25th percentiles (i.e., the interquartile range (IQR)), and the whiskers correspond to the highest value or lowest value of the distribution. If the lowest or highest values are outliers (i.e., >1.5 *IQR from the bounds of the boxes) the whiskers correspond to 1.5 *IQR. Outliers are represented as black diamonds.

On only identities), ABO (true and false positives), and with STNeuroNET (true and false positives) is reported in Supplementary Fig. 8e. These data indicate that the identities captured exclusively by CITE-On were mostly active neurons. We observed no significant difference between the medians of the distribution of SNR values for CITE-On only cells and for all CITE-On TP cells (Supplementary Fig. 8f, Wilcoxon rank sum test, $p = 0.20$, $N = 439$ total number of CITE-On only cells and $N = 4934$ total number of CITE-On TP cells). The average number of cell detections in the different datasets is reported in Supplementary Table 5.

**Online performance on never-before-seen recordings.** We ran CITE-On online and compared the results to our consensus GT annotation on each frame of the ABO, NF, and VPM datasets. When computing the F-1 score, we tested different sizes of the sliding average, in order to define the smallest number of frames required to achieve real-time processing. The absolute maximum F-1 score was achieved in 10 frames for the ABO dataset (both $ABO_{sup}$, and $ABO_{deep}$, Fig. 8a, b), 20 frames for the VPM dataset (Fig. 8e), and 200 frames for the NF (train and test) dataset (Fig. 8c, d). For the ABO dataset, 0.3 s were necessary to acquire 10 frames (0.00086% of the whole time series of average duration

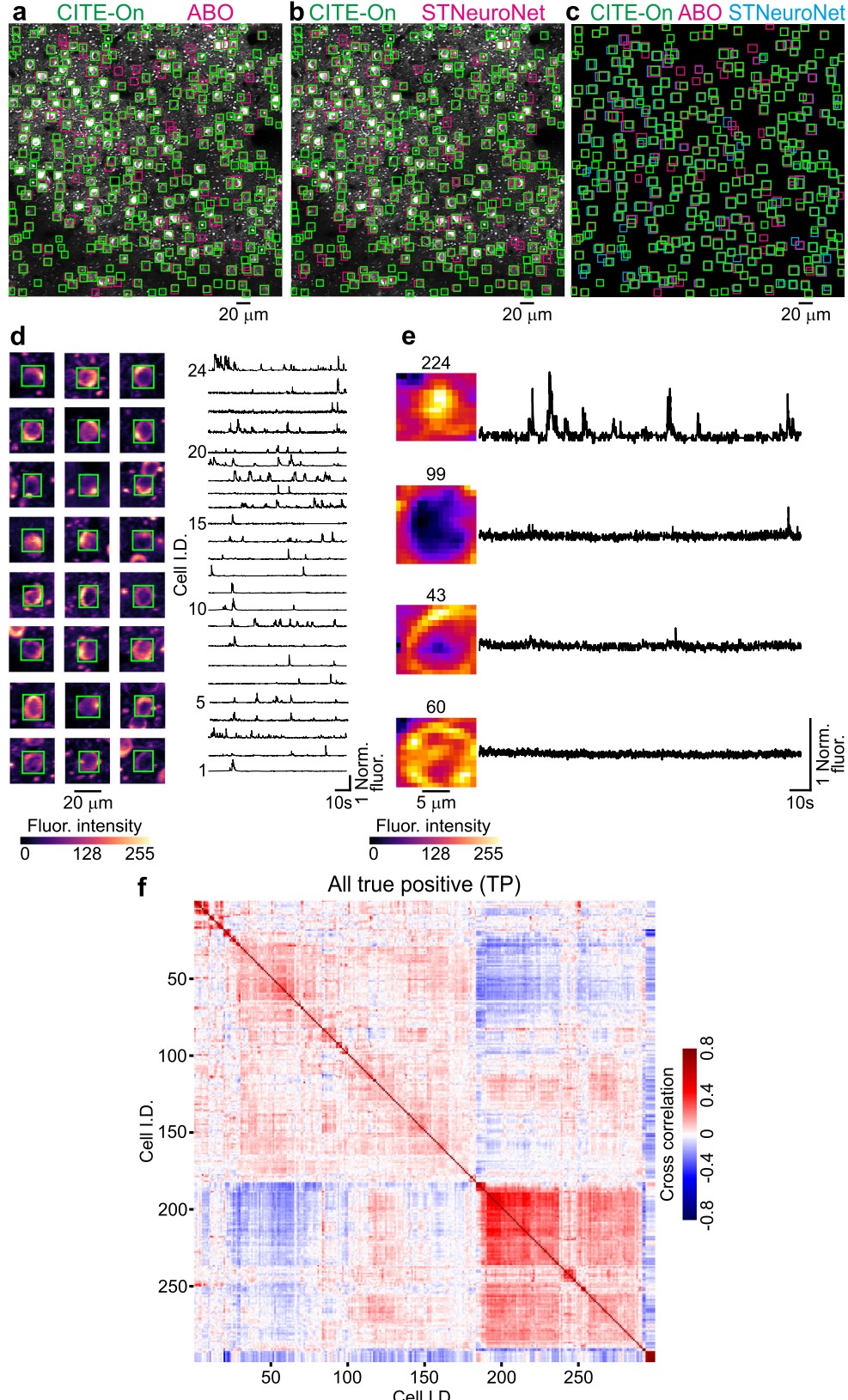

115635 ± 130 frames) with a CITE-On detection rate of 5 Hz and an upscaling factor of 2. The NF dataset required upscaling factors of 2.4 and 2.6 for $NF_{train}$ and $NF_{test}$, respectively. A time window of 28.5 s was required to acquire the 200 frames necessary to reach peak F-1 value (5.4% of the whole t-series of average duration 3697 ± 1874 frames) with a CITE-On detection rate of 4 Hz for $NF_{train}$ and 3 Hz for $NF_{test}$. Given the relatively low online performance for these latter datasets, we decided to measure the F-1 score using an alternative frame downsampling strategy: step average. Using this approach, we found that detection performance remained high (Fig. 8i and Supplementary Table 4). The F-1 score calculated online as a function of the

**Fig. 7 CITE-On data processing of never-before-seen recordings. a** Median projection of a representative t-series from the ABO dataset showing GCaMP6f expressing cortical neurons. CITE-On true positives (CITE-On, green) and true positives provided by the Allen Brain Observatory (ABO, magenta) are shown. **b** Same as in **a** with CITE-On true positives (green) and STNeuroNET true positives (magenta). **c** Superposition of CITE-On (green), ABO (magenta), and STNeuroNET (cyan) true positives. **d** Left: 24 representative cells detected by CITE-On and identified as true positives in ABO and STNeuroNET. The CITE-On-identified bounding box is represented in green. Right: corresponding CITE-On extracted fluorescence traces. **e** Same as in **d** for four representative CITE-On only cells. These fours cells were not included in the GT of the ABO dataset and of the STNeuroNET GT reported in the ref. [22], either as true or false positives. **f** Cross-correlation matrix for all functional traces extracted from true positive (TP) detections in the t-series displayed in **a**. Cell identities are grouped with hierarchical dendrogram sorting. The pseudocolor scale indicates the cross-correlation value.

length of the t-series for ABO ($ABO_{sup}$ and $ABO_{deep}$ together), NF ($NF_{train}$ and $NF_{test}$), and VPM data is shown in Fig. 8j. Stable F-1 scores were observed within 30% of the total length of the t-series. Processing time required for running the online pipeline on the ABO, VPM, and NF datasets was not different from that described previously for our validation datasets. CITE-On performance (both in the offline and the online modality) strongly depended on the mean SNR and, to a lower extent, on the mean fluorescence intensity (Supplementary Fig. 9a–d). CITE-On performance did not show strong dependence on the pixel size and the strategy used for expressing the calcium indicator (Supplementary Fig. 9e–h).

**Analysis of large FOV mesoscopic images**. Given the speed of the CITE-On architecture, we tested if it could be applied to detect cells in the mesoscopic imaging t-series described in Sofroniew et al. [7]. Because the dimensions of the input image were too large (1792 pixels x 1682 pixels, or 4.3 mm × 4.0 mm, at 0.42 pixel/μm) to fit the CNN architecture using the appropriate upscaling factor based on the FOV/neuron surface ratio (upscaling factor for direct processing, 12.8), we tiled the entire mesoscopic FOV in 272 subfields (subfield dimension, 128 pixel x 128 pixels, 28 pixels overlap). Each subfield was appropriately upscaled (upscaling factor, 1; score threshold, 0.4; detection rate, 10 Hz). To increase speed, we multiplexed the CNN detector process and processed subfields in batches of eight images in parallel until completion. Identity duplicates were suppressed using a non-maximum suppression algorithm, where boxes having an IoU > 20% were considered duplicates, and only the one with the highest score was retained (see "Methods"). Image detection outputs were finally recombined to reconstruct the entire FOV. In Fig. 9a, we report CITE-On detected bounding boxes on the entire FOV obtained using the offline pipeline. Two FOV patches are magnified (Fig. 9b) to show the shape of the detected cells (total number of detected somata, 4842). In Fig. 9c, we show representative fluorescence traces obtained with CITE-On from five cells from the whole mesoscopic FOV. Figure 9d shows the cross-correlation matrix (lower-left triangle) of all the identified cells for the first 700 frames of the t-series. The dendrogram analysis (upper-right triangle of Fig. 9d) highlighted several distinct functional modules observed in the identified neuronal population.

The single subfields were processed by the CITE-On detector at 10 Hz (upscaling factor, 1), while segmentation, tracking, and functional trace extraction were performed at 100 Hz (see previous results). Parallel processing of all the 272 sub-fields required 12.6 s for each detection step. Therefore, with a step average downsampling approach including 25 frames (13.2 s of mesoscopic imaging time since acquisition rate was 1.9 Hz), and while extracting traces faster than the incoming frames, we minimized the CITE-On detection lag with respect to the running acquisition. With this strategy, we achieved an online F-1 score of 0.54 (Precision: 0.77; Recall: 0.42) with a score threshold of 0.25 (quantified on 4 patches from the entire FOV). Although the performance was lower in mesoscopic data compared to other

datasets, these results demonstrate that CITE-On can be applied to fast processing of mesoscopic two-photon t-series with good efficiency.

**Discussion**

In this study, we developed a CNN-based algorithm, CITE-On, for fast analysis of two-photon imaging recordings. CITE-On performed online identification of neuronal somata, tracking of identities across frames, dynamic segmentation, and functional trace extraction with background subtraction. CITE-On could generalize across calcium indicators, brain regions, acquisition parameters, and it was successfully applied to data obtained using different surgical and optical preparations (e.g., chronic superficial imaging window, chronic deep imaging window, and endoscopic GRIN lens-based deep imaging).

Our image detection strategy was based on RetinaNET; a CNN originally developed to detect natural images [34]. On one side, this choice was justified by RetinaNET's excellent performance in object recognition and by its availability. On the other hand, it required us to exploit RetinaNET on a set of greyscale two-photon fluorescence images of neurons, which were remarkably different from those RetinaNET was originally trained on. To compensate for this difference and to have a large and heterogeneous dataset for training and validation of the detection algorithm, we built a dedicated library of hundreds of two-photon imaging t-series acquired with different GECIs, in different regions of the mouse brain, at different frame rates, using different surgical/optical preparations, and showing variable image quality. In this dataset, a reliable GT consensus was reached using the annotations of two human graders, which allowed us to evaluate CITE-On performance. To obtain a GT annotation insensitive to the graders' biases, a potential alternative approach could have been to generate an in silico dataset for network training and validation [39]. No online libraries of this kind are currently available, but we foresee that this approach may represent an extremely useful method to optimize future CNN-based approaches for the analysis of two-photon functional data. It is also worth noticing that utilizing public datasets already used for training and validation of alternative processing toolboxes [22] would have given us the possibility to take advantage of third-party GT annotations. However, we decided not to do so because: (i) CITE-On would have likely inherited the annotation bias toward more active cells, which is shared by existing publicly available repositories; (ii) by using public datasets exclusively for the validation of the CITE-On image detector (rather than for training, too), we avoided any chance of data leakage, and we demonstrated that CITE-On generalizes to never-before-seen data, although with lower performance.

In the absence of GT annotations, CITE-On can be run in three consecutive steps (see also Methods). First, the optimal upscaling factor must be set. This step can be performed on the median projection of the motion-corrected t-series by manually adjusting the upscaling factor in order to have tight bounding boxes surrounding neuronal somata. The second step consists in gradually increasing the offline score threshold to decrease the number of

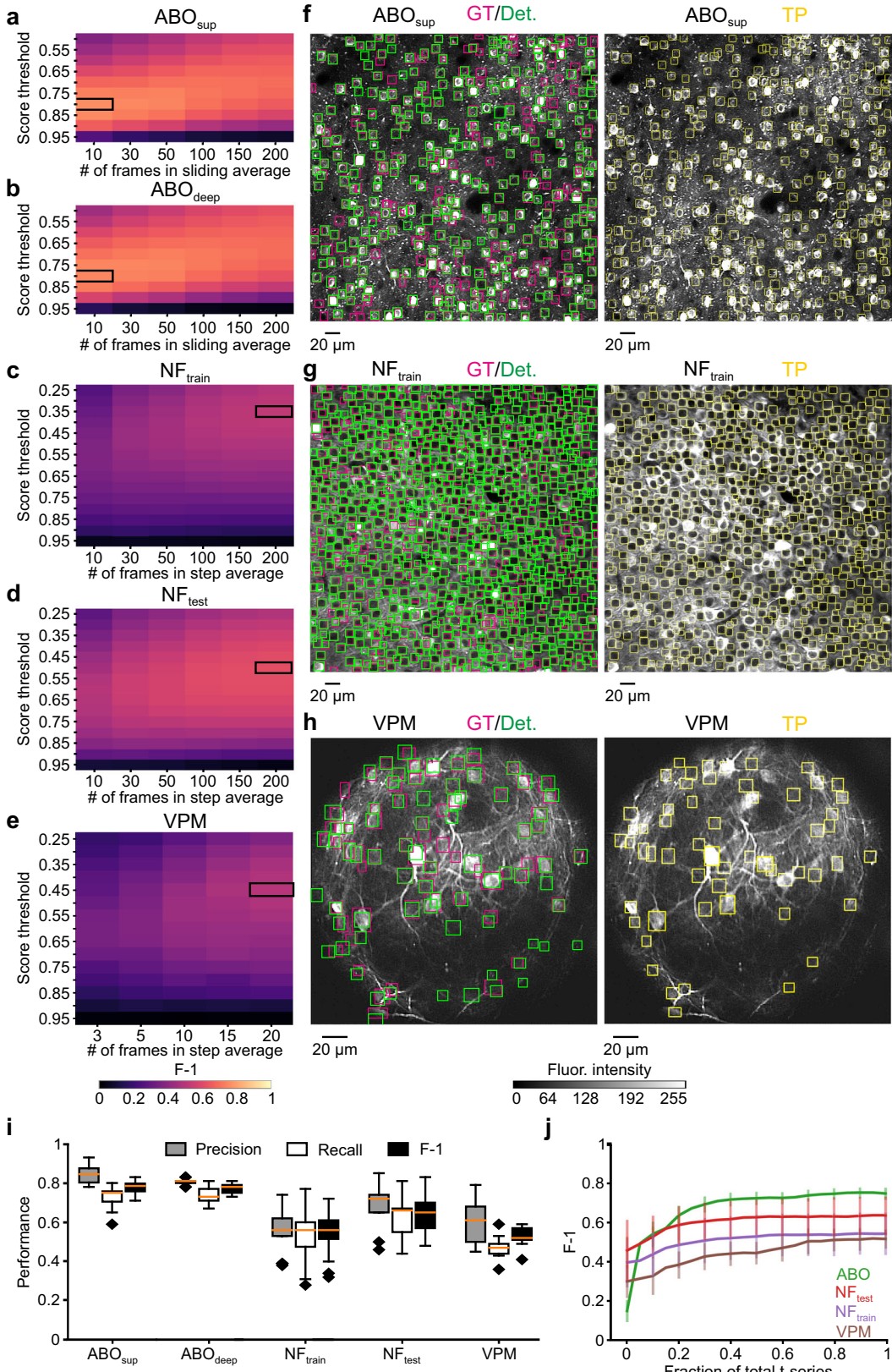

false positives. The third step involves running a grid search to determine: (i) the most appropriate number of averages, and (ii) the online score threshold giving the highest F-1 score compared to the offline detection obtained in the first step described above. This three-step procedure is implemented in a Jupyter notebook,

which is available in the online repository (https://gitlab.iit.it/fellin-public/cite-on).

CITE-On performed similarly to state-of-the-art algorithms[10,11,16,19,22,40] on publicly available datasets, and importantly, it did so in a much shorter time. In fact, only a few

**Fig. 8 CITE-On online cell detection performance on never-before-seen datasets. a–e** Best parameter search for frame downsampling: F-1 score (pseudocolor) as a function of score threshold and number of frames for the $ABO_{sup}$ (**a**), $ABO_{deep}$ (**b**), $NF_{train}$ (**c**), $NF_{test}$ (**d**), and VPM (**e**) datasets. The maximal F-1 is indicated by the black rectangle. The pseudocolor scale in (**e**) applies to (**a–d**). For the $ABO_{sup}$, $ABO_{deep}$ datasets the sliding average frame downsampling approach was used, while for the $NF_{test}$, $NF_{train}$, and VPM datasets, the step average approach was implemented. **f–h** Left: median projection of a representative t-series from the $ABO_{sup}$ (**f**), $NF_{train}$ (**g**), and VPM (**h**) datasets. GT (magenta) and online CITE-On detections (green bounding boxes) are shown. Right: bounding boxes (yellow) corresponding to true positives (TP) are shown. The greyscale in **h** applies also to **f**, **g**. **i** Boxplots showing online detection performance of Precision (gray), Recall (white), and F-1 (black) for all t-series in the $ABO_{sup}$ ($N = 9$), $ABO_{deep}$ ($N = 10$), $NF_{train}$ ($N = 19$), $NF_{test}$ ($N = 9$), and VPM ($N = 9$) datasets. The orange line in all boxplots is the median, the bounds of the boxes are the 75th and 25th percentiles (i.e., the interquartile range (IQR)), and the whiskers correspond to the highest value or lowest value of the distribution. If the lowest or highest values are outliers (i.e., >1.5 *IQR from the bounds of the boxes) the whiskers correspond to 1.5 *IQR. Outliers are represented as black diamonds. **j** F-1 values as a function of the fraction of processed t-series for ABO (green, $N = 19$ t-series), $NF_{test}$ (red, $N = 9$ t-series), $NF_{train}$ (purple, $N = 19$ t-series), and VPM (brown, $N = 9$ t-series) datasets. Ten frames sliding averages for ABO; detection rate, 5 Hz. Step median of 20 frames and 200 frames for VPM and NF datasets; detection rate, 0.3 Hz and 0.035 Hz for VPM and NF datasets, respectively.

seconds were needed to have online, frame-by-frame, accurate ROI segmentation, identity tracking, *bg* subtraction, and functional trace extraction. Four main characteristics were crucial for CITE-On's high performance. First, CITE-On relied exclusively on morphological features to identify neurons. Second, neuronal identification was dynamic and it adapted to changes in shape, position, and activity of the detected cells frame-by-frame, avoiding time-consuming motion correction procedures. Third, once bounding boxes were identified in individual frames, we used a simple computationally effective strategy to extract pixels belonging to neuronal ROIs based on their brightness. Fourth, we implemented a fast background subtraction strategy, limiting computational costs. When applied in the online modality, these characteristics were crucial to achieve real-time frame-by-frame trace extraction, something current approaches do not achieve[9–25], while maintaining high cell detection performance. The observation that functional fluorescence traces extracted by CITE-On were highly correlated when the SNR was large with those extracted on the same bounding boxes by a state-of-the-art method, i.e., CaImAn[10], confirmed the validity of our computationally effective approach.

Thanks to the features described above, CITE-On efficiently processes full mesoscopic two-photon t-series (FOV dimension, 4.3 mm × 4 mm). It did so by dividing each image into subfields and processing subfields in parallel. CITE-On's detector processed single subfields at 10 Hz, while segmentation, tracking, and functional trace extraction were performed at 100 Hz. Parallel processing of all the 272 sub-fields generating a whole mesoscopic FOV required 12.6 s for each detection step. Thus after 12.6 s, trace extraction could be performed at 100 Hz on thousands of cells. Besides its application online, the offline application of CITE-On is also going to be extremely powerful for the identification of the thousands of neurons imaged in mesoscopic two-photon functional imaging.

CITE-On online works on individual images. These were either updated frame-by-frame after obtaining them as the result of a sliding average approach, or they were updated every *n*-frames when a step average strategy was used. In both cases, CITE-On online has an initial lag in detecting identities, due to the time required to compute the first sliding average or step average. During this initial lag, detections are not available, and if no previous detections had been computed, no functional trace extraction is performed. In the case of the LIV, CA1, VPM, and ABO datasets, the initial lag was 6.6 s, 14 s, 7 s, and 0.3 s, respectively, using the sliding average approach. For the NF and mesoscope datasets, on which a step averaged approach was taken, a time window of respectively 28.5 s and 12.6 s was required for the computation to be performed. In both cases, the shape of each bounding box was updated every time active detections were updated. This process occurred in real time for

the LIV, CA1, VPM, and ABO datasets. In all cases, dynamic segmentation and functional trace extraction were performed at 100 Hz, which was faster than real time. CITE-On did not retrospectively update functional traces corresponding to a newly identified ROI, and functional traces started being extracted only when the associated ROI was detected.

Closed-loop all-optical experiments are fundamental to investigate whether models of network dynamics, circuit connectivity, and causality are accurate[26]. Recently, all-optical closed-loop experiments have been validated[27]. For example, using this approach specific groups of neurons were activated based on the readout of ongoing activity in a reference cell. However, the closed-loop strategy described in[27] was based on a priori identification of the reference cell. Because CITE-On allows efficient frame-by-frame cell identification and trace extraction, it will enable a new type of experiment in which the loop is closed based on real-time identification and readout of any neuron or group of neurons in the FOV.

It is important to underline that CITE-On performance is lower in the online modality than in the offline modality. Moreover, during online processing, CITE-On has on average lower performance in the initial third of each given t-series (Fig. 3), with F-1 scores ~0.6. The F-1 score then plateaus in the second and third of the t-series. Given that different datasets had different sampling rates, average length of the acquisitions, and considering the non-stationary activity of neurons, CITE-On required a different number of frames to reach plateau F-1 score across different datasets. This value was ~40,000 frames for a t-series acquired at 30 Hz with SNR comparable to that of the ABO dataset, ~1200 frames for 30 Hz frame rate movies with SNR comparable to that of the $NF_{test}$ dataset, ~260 frames for a 2.66 Hz frame rate movie with SNR comparable to that of the VPM dataset, ~150 frames for a 1.5 Hz frame rate movie with SNR comparable to that of the LIV dataset, and ~250 frames for a 3 Hz frame rate movie with SNR comparable to that of the CA1 dataset.

Cross-correlation values between traces extracted with CITE-On and those extracted with seeded CaImAn depended on the trace SNR (i.e., correlations were lower for small values of SNR). This result may indicate that CITE-On is less accurate in extracting functional traces from low SNR cells. However, this effect may also be due to the reduced accuracy of CaImAn in extracting functional traces from low SNR cells[10]. A way to discriminate between these possibilities would be to pair single-cell electrophysiology (to record the cell's spiking activity) with two-photon calcium imaging and test which method (CITE-On vs. CaImAn) extracts functional traces that best match the AP firing profile of cells with low SNR. However, low SNR cells are typically absent in current available datasets of combined imaging and electrophysiological measurements due to the difficulty of

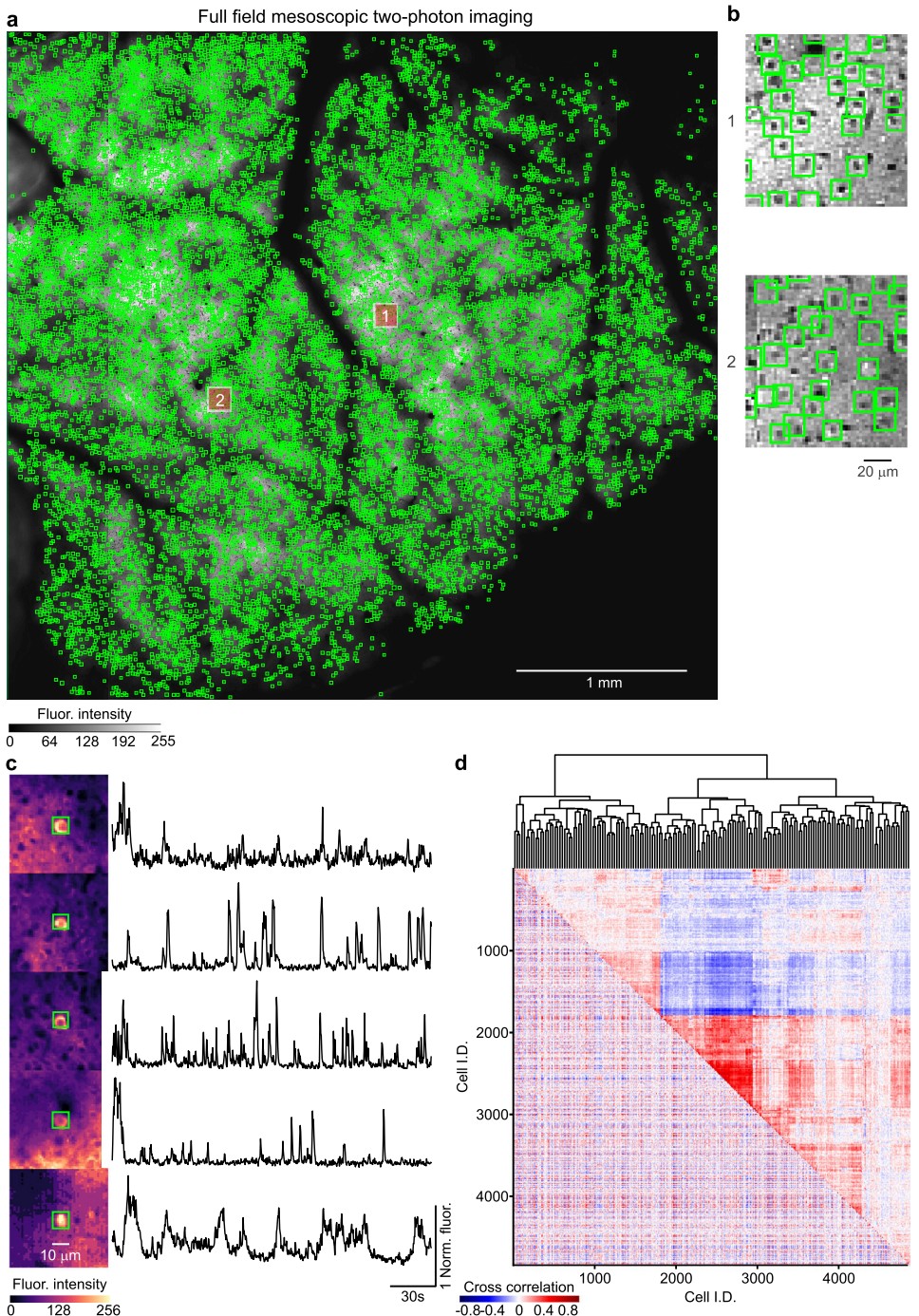

**Fig. 9 CITE-On analysis of mesoscopic two-photon imaging t-series. a, b** Median projection of a mesoscopic imaging t-series showing GCaMP6s expressing neurons (mesoscopic data from ref. [7]). Green boxes indicate cells detected by CITE-On (total: 4842 cells). Two regions are highlighted by the red and white squares, and are shown at an expanded spatial scale in **b**. Greyscale in **a** applies also to **b**. **c** Left: five representative cells detected by CITE-On. The brightest cell in each FOV was selected. Right: corresponding CITE-On extracted fluorescence traces in the first 230 s of the t-series. **d** Cross-correlation matrix (bottom-left triangle) calculated on the background-subtracted traces extracted by CITE-On on all detected cells in the first 7000 frames and relative dendrogram (top-right triangle).

performing imaging-guided electrophysiological recordings on low SNR cells. Although the pairwise correlations between low SNR cells extracted by CITE-On and CaImAn are low, we believe that these should not be removed from the CITE-On online analysis, for two reasons. First, excluding these identities altogether may lead to a substantial bias towards more active cells or cells with higher levels of indicator expression. Second, SNR is a

dynamic property, and each cell may display both high and low SNR periods during chronic imaging.

Since online CITE-On detection efficiency increased with increasing SNR of the input image (Supplementary Fig. 9), we decided to feed the CNN with images resulting from averaging a subset of frames from the running t-series, rather than individual frames. This approach increased the detectability of neuronal

somata (Supplementary Fig. 1c–d) because the average across frames reduced uncorrelated noise emerging from each individual frame. In order to process data online, we opted for a sliding average approach. Here, the input image fed to the CNN was obtained by averaging the last $n$ frames of the t-series, and it was updated at every new incoming frame. The input image was then appropriately upscaled and processed as described for the offline pipeline. In the presence of large FOVs and small pixel size, the upscaling process could be slower than the time required for the calculation of the sliding average. In this case, a step average approach was used, where the input image was computed on blocks of $n$ frames and updated every $n$ frames.

The result of the regression process, performed by the CNN, on the anchor boxes against putative position and size of a cell, determines the dependence of CITE-On performance on neuronal size (and on the upscaling factor value). When the upscaling factor is optimized, the size of the anchor boxes matches that of the feature(s) to detect, and the detection process ends in good agreement with the GT annotation. On the other hand, if the upscaling factor is not optimized, the sizes of the anchor boxes and the features to detect do not match, and lower performance is expected.

CITE-On has some similarities but also several differences compared to CaImAn online[10]. First, both CITE-On online and CaImAn online require the optimization of some initialization parameters[10]. In CITE-On, initialization parameters (e.g., upscaling factor and score threshold) are set knowing the dimension and the resolution of the target t-series. In contrast, CaImAn online requires the user to provide input parameters such as number of expected components, maximum number of neurons added per frame, threshold on SNR for accepting new identities[10]. These can be difficult to estimate simply based on the acquisition settings, and are usually optimized in repeated rounds of offline segmentation. Second, CaImAn online requires a preprocessing step for offline segmentation, which typically runs on 1000–3000 frames[10]. The quality of this initial offline segmentation has a large impact on the subsequent online processing. To obtain a reliable (convergent) segmentation using CaImAn online, it is often necessary to run multiple rounds of offline segmentation with different initialization parameters[10] and the result is subordinated to the level of cellular activity[10]. Once the offline segmentation converges, CaImAn online starts the online analysis, updating ROIs (in terms of position, shape, and newly identified identities) with a temporal lag that is generally larger than the 2–4 min required for a single iteration of offline segmentation preprocessing[10]. CITE-On reduces this initial lag. After this time window, frames were processed in real time. Third, in the online modality, CITE-On does not require the correction of motion artifacts when lateral displacements are within 4 µm/s (Supplementary Fig. 4). In contrast, CaImAn online requires a frame-by-frame motion correction routine that is fast and efficient (on average 5 ms per frame[10]), but is dependent on the surface (i.e., number of pixels) of the FOV. The correction of motion artifacts using CaImAn online may thus introduce significant delays when processing large FOVs (e.g., mesoscopic imaging data). Fourth, CITE-On performs tracking, dynamic segmentation, and functional trace extraction at 100 Hz independently of the number of detected neurons and their activity. This feature allows maintaining high online performance on FOVs characterized by large numbers of neurons (e.g., those obtained from mesoscopic imaging) and sparse activity. Frame-by-frame processing with CaImAn online, instead, depends on the number of ROIs to be updated or added on the basis of the initial offline segmentation[10]. Fifth, CITE-On does not use local pixel correlation for cell identification, which may be advantageous when separating nearby synchronous cells. In contrast, the

CaImAn online fast deconvolution approach may be more efficient in separating adjacent cells with different temporal profiles of fluorescence emission[10]. Finally, CaImAn online was tested on two datasets with rather homogenous acquisition parameters, and its application to different experimental conditions was not fully characterized[10]. Here, we demonstrate that CITE-On generalizes across indicators (i.e., GCaMP6s, GCaMP6f, GCaMP7f, and jRCaMP1a) and across data acquired in different brain regions and with different pixel sizes, SNR, and frame rates. CITE-On performance on never-before-seen data tended to be, however, lower. Thanks to the properties described above CITE-On is a flexible online analysis tool to apply in different experimental conditions.

Because cell identification was based only on spatial features, CITE-On identified both active and silent cells. This unique characteristic of CITE-On is important because it adds further flexibility in designing imaging experiments. Neurons that are silent in a t-series may change their level of activity in subsequent acquisitions depending on the behavioral state of the animal or because of external manipulations[41–43]. Thus, being able to track cells regardless of their activity level is key, for instance, for investigating the sensory information carried by neurons that significantly change their activity throughout longitudinal imaging experiments. Biasing the cell identification toward active neurons, as currently done by most approaches, intrinsically skews the proportion of analyzed cells towards those that are responsive to a given stimulation in a certain brain region. In this regard, it is also interesting to note that neurons that were detected only by CITE-On and not by other state-of-the-art approaches comprised silent cells but, unexpectedly, also active cells.

In summary, we developed CITE-On, a tool to effectively process two-photon imaging data frame-by-frame, while maintaining similar cell detection and trace extraction performance of existing offline state-of-the-art methods. Future developments of CITE-On will likely include its optimization for one-photon imaging[44,45], its application to genetically encoded voltage indicators[46] as well as to volumetric two-photon imaging[47].

## Methods

**Animals**. All experiments were carried out in accordance with the guidelines of the European Communities Council Directive and were approved by the National Council on Animal Care of the Italian Ministry of Health (authorization #34/2015-PR, #1134/2015-PR, and #61/2019-PR).

Wild type (wt) C57BL/6 J mice were purchased from Charles River Laboratories (Calco, Italy; strain code #632), transgenic Scnn1a-cre (B6;C3-Tg(Scnn1a-cre)3Aibs/J; JAX #009613), and Ai95D (B6;129S-*Gt(ROSA)26Sor*<sup>tm95.1(CAG-GCaMP6f)Hze</sup>/J; JAX #024105) were purchased from Jackson Laboratories (Bar Harbor, USA). Scnn1a-cre mice express the enzyme Cre in a subpopulation of layer IV neurons[48] and of VPM cells[49]. Animals were housed in individually ventilated cages under a 12-h light:dark cycle, with controlled room temperature and humidity (22–23 °C, 60%, respectively). A maximum of five animals per cage was allowed with ad libitum access to food and water. Mice of both sexes were used for experiments.

**Viral injections and surgical procedures**. We expressed GCaMP6 or GCaMP7 through the following viral vectors AAV1.Syn.Flex.GCaMP6s.WPRE.SV40 (Addgene viral prep # 100845-AAV1), AAV1.Syn.Flex.GCaMP6f.WPRE.SV40 (Addgene viral prep # 100833-AAV1) purchased from the University of Pennsylvania Viral Vector Core, or AAV1.Syn.Flex.GCaMP7f.WPRE.SV40 (Addgene viral prep #104492-AAV1) purchased from Addgene. For CA1 imaging, we expressed jRCaMP1a using co-injection of AAV1.CAG.Flex.NES-jRCaMP1a.WPRE.SV40 (Addgene viral prep # 100846-AAV1) and AAV1.CamKII 0.4.Cre.SV40 (Addgene viral prep # 105558-AAV1) purchased from the University of Pennsylvania Viral Vector Core.

For LIV imaging, we used a total of 29 mice. Specifically, 17 Scnn1a-cre mice injected with a viral vector transducing GCaMP6s, six Scnn1a-cre mice injected with a virus transducing GCaMP6f, three Scnn1a-cre mice injected with a virus transducing GCaMP7s, and three Scnn1a-cre crossed with Ai95D mice. Mice between post-natal days 30 and 33 were anesthetized with 2% isoflurane (IsoFlu, Zoetis, IT) in 0.8 (L/min) oxygen, placed into a stereotaxic apparatus (Stoelting Co, Wood Dale, IL), and maintained on a warm platform at 37 °C for the whole duration of the anesthesia. Viral

injection in mice used for LIV imaging was carried out similarly to refs. [36,50] and ref. [51]. Briefly, a scalp incision was performed after local administration of Lidocaine (2%), and then two small holes were drilled on the skull above the right/left somatosensory cortex at 1.2 mm and 2 mm posterior (P) to the bregma suture, 2.8 mm and 3 mm lateral (L) to the sagittal sinus. A micropipette was slowly inserted into the cortical tissue until the tip reached a depth of 0.3 mm below the pia mater[52]. 200 nL of GCaMP6 virus were injected at 20–60 nl/min by means of a hydraulic injection apparatus driven by a syringe pump (UltraMicroPump, WPI, Sarasota, FL). The pipette was then further lowered to reach 0.4 mm below the pia mater, and a second injection was performed. This procedure was repeated for the second injection site. The injected solution contained $10^{12}$ viral genomes/ml diluted 1:1 in artificial cerebrospinal fluid (aCSF: 127 mM NaCl, 3.2 mM KCl, 2 mM CaCl$_2$, 1 mM MgCl$_2$ and 10 mM HEPES, pH 7.4). The exposed skull was then cleaned, and the skin sutured and cleansed with Iodopovidone (Betadine®, Meda Pharma, Milan, Italy). Mice were monitored until full recovery from the anesthesia. In mice used for imaging in awake conditions, a custom metal bar was sealed to the skull using Vetbond (3 M, St. Paul, MN, USA) and dental cement (Paladur, Kulzer GmbH, Hanau, Germany). The exposed bone was covered using the silicone elastomer KWIK-Cast (World Precision Instruments, Friedberg, DE), and an intraperitoneal injection of antibiotic (BAYTRIL, Bayer, DE) was performed. Two to four weeks after virus injection, mice used for imaging in LIV during anesthesia were injected with urethane (1.65 g/kg, 16.5% in saline solution, i-p.). A scalp incision was performed after local administration of Lidocaine (2%). A circular craniotomy was opened over the somatosensory cortex, in the area where green fluorescence was clearly visible using an epifluorescent microscope. The surface of the brain was kept moist with aCSF. A heating pad underneath the animal was set at 35.5–37 °C. Respiration rate, eyelid reflex, vibrissae movements, and reactions to tail pinching were monitored throughout the surgery. Mice were then moved under the two-photon microscope, kept at 37 °C with a heating pad, and the brain surface irrigated with aCSF. Imaging began 1 h after the end of the surgery. Before imaging LIV activity in awake animals, mice were habituated to head-fixation similarly to[53]. In brief, habituation lasted for 7–10 days, during which they were head restrained for increasing periods (from 15 min to 1 h), while running or standing on a custom-made treadmill. On the day of the experiment, the habituated mouse was anesthetized with a mixture of isoflurane and oxygen (2%–0.8 L/min), and a was craniotomy performed similarly to that described above. After surgery, the animal was head-fixed and allowed to recover under the microscope for at least 1 h before imaging.

For VPM imaging, we used a total of 4 mice. Viral injections and aberration-corrected microendoscopes insertion in mice used for VPM imaging were performed in Scnn1a-cre mice as in[49]. Mice were anesthetized as previously described. A single craniotomy was performed at stereotaxic coordinates P 1.7 mm, L 1.6 mm. A micropipette was lowered to a depth of 3 mm below the brain surface. 0.5–1 μl of GCaMP6s virus-containing solution (containing $10^{12}$ viral genomes/ml diluted 1:4 in aCSF) were injected at 30–50 nl/min. Following virus injection, a craniotomy (area: 600 μm × 600 μm) was performed at stereotaxic coordinates P 2.3 mm, L 2 mm. A thin column of brain tissue was displaced with a glass cannula (ID = 300 μm, OD = 500 μm; Vitrotubs, Vitrocom Inc, Mounting Lakes, NJ) and a microendoscope was slowly inserted into the cannula track using a custom holder, down to 3 mm from the brain surface. The microendoscope was finally secured by acrylic adhesive and dental cement. Imaging was performed 2–4 weeks after endoscope implantation.

For CA1 imaging, we used a total of 2 male mice (8–10 weeks old). Before surgery, mice were medicated with an intramuscular bolus of Dexamethasone (Dexadreson, 4 gr/kg). After scalp incision, a 0.5 mm craniotomy was drilled at stereotaxic coordinates P 1.75 mm, L 1.35 mm. A micropipette was lowered 1.40 mm below the brain surface. 0.8 μl of viral solution (containing a mixture of CamKII-Cre and jRCaMP1a viruses at $10^{12}$ viral genomes/ml diluted 1:1 in aCSF) was injected at 100 nL/min in Ai95D crossed with Glast-cre-ERT2 (Slc1a3$^{tm1(cre/ERT2)Mgoe}$) mice[54]. Inducible Glast-cre was not activated after viral injection, resulting in CA1 neuronal labeling with jRCaMP1a and GCaMP6f. A stainless-steel screw was attached to the skull, and a chronic hippocampal window was implanted as described in[55,56]. A 3 mm circular craniotomy was opened, centered at coordinates P 2.00 mm, L 1.80 mm. The dura mater was removed using fine forceps, and the cortical tissue overlaying the hippocampus slowly aspirated using a blunt needle coupled to a vacuum pump. During aspiration, the exposed tissue was continuously irrigated with aCSF. Aspiration was stopped once the fibers of the external capsule were exposed. A cylindrical optical window made of a thin-walled stainless-steel cannula (OD, 3 mm; ID, 2.77 mm; height, 1.50–1.60 mm) attached to a 3 mm diameter round coverslip was fitted to the craniotomy in contact to the external capsule. A thin layer of silicone elastomer was used to surround the interface between the brain tissue and the steel surface of the optical window. A custom stainless-steel headplate was attached to the skull using epoxy glue. All the components were finally fixed in place using black dental cement, and the scalp incision was sutured to adhere to the implant. All the animals received an intraperitoneal bolus of antibiotic (BAYTRIL, Bayer, DE) to prevent postsurgical infections.

**Functional two-photon imaging.** Two-photon imaging was performed using a chameleon ultra II pulsed laser (80 MHz pulse frequency, Coherent Inc, Santa Clara, CA, USA) tuned at 920 nm for GCaMP6/7 imaging and at 990 nm for dual-color imaging. Excitation power was 30–110 mW as measured under the microscope objective and controlled via a Pockel cell (Conoptics Inc, Danbury CT, USA,). An Ultima II scanhead (Bruker Corporation, Milan, Italy) equipped with 3 mm raster scanning galvanometers (6215H, Cambridge Technology, Bedford,

MA) and standard photomultiplier tubes (Hamamatsu Photonics, Tokyo, Japan) and an Ultima Investigator (Bruker Corporation, Milan, Italy), equipped with 6 mm raster scanning galvanometers, movable objective mount, and multi-alkali photomultiplier tubes were used. The three objectives were: 25x/1.05 NA (Olympus Corp., Tokyo, JP) for LIV imaging, 20x/0.5 NA (Zeiss, Oberkochen, Germany) for VPM endoscopic imaging, and 16x/0.8 NA (Nikon, Tokyo, Japan) for CA1 experiments.

For LIV imaging, dwell time was 4.4 μs, photomultiplier voltage was 777 V, zoom factor was always 1, pixel size was 0.77 μm, acquisition frame rate ranged between 0.5–3 Hz for a 512 pixels x 512 pixels image. Fluorescence values spanned 95% of the available dynamic range (16 bit). For dual-color CA1 imaging, pixel dwell time was set at 4 μs; photomultiplier voltage was 777 V; zoom factor was always; pixel size was 0.634 μm; acquisition frame rate was 3.03 Hz for a 256 pixels x 256 pixels image. For VPM imaging, the setup was similar to the one described in[49,57], pixel dwell time was set at 4 μs, photomultiplier voltage was 810 V, zoom factor was always 1, pixel size was 2.19 μm, acquisition frame rate was 2.66 Hz for a 196 pixels x 196 pixels image. Imaging sessions lasted 1 h for CA1, VPM, and awake LIV experiments. They lasted 4 h for the anesthetized LIV condition. After awake imaging sessions, animals were returned to their home cages.

**Training and validation datasets.** In the absence of a generally accepted wide-scale annotated dataset of two-photon calcium imaging, we built a dataset of in vivo t-series collected using raster scanning acquisitions. A total of 197 t-series (average frame number per time series: 597 ± 262, average frame rate: 2.3 ± 1.5 Hz) were included in the dataset: 66 t-series from CA1 imaging of principal neurons stained with both GCaMP6f (33 t-series) and jRCaMP1a (33 t-series); 131 t-series from cortical LIV imaging of principal neurons stained with virally injected GCaMP6s (113 t-series), GCaMP6f (8 t-series), GCaMP7f (5 t-series), GCaMP6f expressed in transgenic animals (Scnn1a-cre x Ai95D; 5 t-series). Training and validation datasets contained 160 (118 from LIV, 21 from CA1 GCaMP6f, and 21 from CA1 jRCaMP1a) and 37 t-series (13 from LIV, 12 from CA1 GCaMP6f, and 12 from CA1 jRCaMP1a), respectively. To avoid data leakage between training and validation datasets, we grouped together t-series acquired from the same FOV and included these data either in the training or validation datasets.

**Additional datasets.** Four additional datasets were selected and used for validation purposes only:

1. VPM microendoscopic imaging t-series in awake head restrained mice (9 t-series).
2. The publicly available Allen Brain Observatory (ABO) visual coding dataset (19 t-series, https://observatory.brain-map.org/visualcoding). T-series identification numbers: 501271265, 501484643, 501574836, 501704220, 501729039, 501836392, 502115959, 502205092, 502608215, 503109347, 504637623, 510214538, 510514474, 510517131, 527048992, 531006860, 539670003, 540684467, 545446482. The ABO repository contains both the raw imaging data used in this work and the annotation of true positive cell identity produced by the curators of the ABO dataset as described in:

http://help.brain-map.org/download/attachments/10616846/VisualCoding_Overview.pdf?version=5&modificationDate=1538066962631&api=v2() The ABO ground truth annotation was used in Fig. 7a–c and Supplementary Fig. 8.
3. The publicly available Neurofinder (NF) challenge dataset (28 t-series, https://github.com/codeneuro/neurofinder). T-series identification numbers: neurofinder.00.00, neurofinder.00.01, neurofinder.00.02, neurofinder.00.03, neurofinder.00.04, neurofinder.00.05, neurofinder.00.06, neurofinder.00.07, neurofinder.00.08, neurofinder.00.09, neurofinder.00.10, neurofinder.00.11, neurofinder.01.00, neurofinder.01.01, neurofinder.02.00, neurofinder.02.01, neurofinder.03.00, neurofinder.04.00, neurofinder.04.01, neurofinder.00.00.test, neurofinder.00.01.test, neurofinder.01.00.test, neurofinder.01.01.test, neurofinder.02.00.test, neurofinder.02.01.test, neurofinder.03.00.test, neurofinder.04.00.test, neurofinder.04.01.test.
4. A single t-series of mesoscopic full field imaging from ref. [7].

No preprocessing was performed on the VPM, ABO, NF, and mesoscopic t-series. All t-series were manually annotated de novo by the two graders working independently. The consensus ground truth was obtained as described for the training and validation dataset below (see also Supplementary Table 1–2).

**Image processing and consensus labeling.** The CITE-On image detector was based on purely morphological features extracted from imaging data. No information from the dynamic fluorescence signal in the t-series was used to detect putative cells. Each imaging t-series was corrected for lateral displacements using the MOCO[35] implementation present in Fiji (ImageJ 1.52p). T-series were aligned using the raw median projection as a reference, without downsampling and with a maximum possible shift of 13 pixels. The 8-bit median projection of each t-series was then computed on the motion-corrected t-series. The resulting images (one per t-series) were globally sharpened to better visualize cell shapes using a [[−1, −1, −1], [−1, 12, −1], [−1, −1, −1]] kernel. A gamma correction of 0.3 was applied, and the dynamic range was linearly adjusted, normalizing across the whole 8-bit depth. Processed images were named "enhanced median projections" (EMPs) and were used to define

our GT labeling. Two graders independently labeled each EMP. LabelImg (http://github.com/tzutalin/labelImg) was used to define a single object class by manually drawing bounding boxes around every visible cell soma in the EMP. The surface of each bounding box was manually defined in order to tightly surround the cell shape. Boxes were allowed to overlap. Coordinates and surface of each bounding box for all EMPs were saved in a standard VOC format where each file reported the top left and bottom-right coordinates (in pixels) for each bounding box. For each t-series, the annotations of the two graders were overlapped, and the two graders accepted all boxes that overlapped with a threshold >0.5 of the intersection of the boxes' surface ($GT_{overlap}$). For boxes with overlap <0.5, the two graders together first analyzed the boxes only included by grader 1 ($GT_{grader1only}$) and then those included only by grader 2 ($GT_{grader2only}$). Boxes were retained when both graders were in agreement, otherwise the identity was rejected. This procedure generated two novel sets of boxes ($GT_{grader1only\_consensus}$ and $GT_{grader2only\_consensus}$). The final consensus GT was the one shared between the two graders, i.e., $GT_{overlap} + GT_{grader1only\_consensus} + GT_{grader2only\_consensus}$.

**Image detector training**. CITE-On is based on a fully convoluted single-shot image detector, RetinaNet[34]. Briefly, a feature pyramid network was constructed from residual layers of the ResNet50 feature extractor[58]. This feature pyramid was then fed to two separate sets of convolution filters: one computing the label score (classification subnet), the second performing bounding box regression from anchor boxes (regression subnet). We used the consensus GT to train the RetinaNet model from its Keras implementation (https://github.com/fizyr/keras-retinanet) using a transfer learning approach (i.e., starting with a model pre-trained on natural images), and achieving best performance after 17 epochs (validation mAP = 0.79). Starting from the network trained on a large-scale dataset of natural images, we fine-tuned the weights of the classification and regression subnets, while freezing the weights of the feature pyramids. We used "plain" median projections obtained from the motion-corrected t-series and linearly normalized across the bit range. The resulting projections were then upsampled in order to obtain images where the short side was 800 pixels long, while the long side did not exceed 1333 pixels. Since the input layer of the network accepted a three-channel image, the same image was replicated for each channel without changing any parameter. These last two image conversions were necessary as the network was originally trained on RGB images. The network was trained with a regression L1 loss function (Mean Absolute Errors (MAE) https://rishy.github.io/ml/2015/07/28/l1-vs-l2-loss/) and with focal loss (http://arxiv.org/abs/1708.02002) using the Adam optimizer[59] with learning rate $10^{-5}$ and clipnorm $10^{-3}$ (http://github.com/keras-team/keras/issues/510)[34] modified by reducing the learning rate on loss plateau with a factor of 0.1. The network was trained for 17 epochs, each consisting of 1000 training steps of batch size 1.

CITE-On is a Python library and is freely available, along with the trained network, on our institutional GitLab (https://gitlab.iit.it/fellin-public/cite-on). For ease of access, the repository is accompanied by sample Jupyter notebooks detailing every aspect of both the offline and the online pipelines and containing guidelines for parameter optimization. For offline and simulated online analysis, a Jupyter instance (preferably with GPU acceleration) is needed in order to run the notebooks. For true online implementation, programming experience in Python is required only to interface CITE-On with the microscope acquisition software.

**CITE-On offline pipeline**. Two-photon calcium imaging t-series were first corrected for lateral artifacts using MOCO (as described above). The median projections were then computed, normalized, upscaled to the target input size, and converted to 8-bit RGB images. The resulting images were fed to the image detection network. Upscaling factor and score threshold were the only two parameters defined a priori. The parameter "upscaling factor" was defined as the geometric transformation of the input image before it is fed to the CNN input layer. The score threshold was defined as the minimum value of score needed for each box to be considered as true. The upscaling factor retained the original aspect ratio of the input image, while the absolute size of all image features changed (e.g., neuronal somata). In RetinaNET[34], a set of anchor boxes were used to predict the size and position of the bounding box for an object, independently from the input image size. Moreover, each location on a given feature map in RetinaNet had nine anchor boxes (at three scales and three ratios). The relationship between the anchor boxes and the dimension of the features on which the CNN performed the detection (i.e., neuronal somata) was therefore important. The upscaling process was set out to optimize this relationship and improve detection performance. We did not perform an ab initio training of a CNN because of the lack of suitably large two-photon datasets annotated for neuronal somata morphology. Rather, because RetinaNET is trained on millions of natural images, we opted for a transfer learning approach. This strategy prevented the modification of the anchor boxes size. The upscaling factor to be used for each input image (or groups of images with features of similar size) was empirically defined. Specifically, while exploring a range of upscaling factors and score thresholds, we used a grid search approach aimed to maximize the F-1 score (Fig. 5a–e). This optimization step can be refined for any new dataset containing features of a size different from those used in this study. Alternatively, new CITE-On users can optimize their upscaling factor by using the simple empirical relation described in Fig. 5f. Indeed, the upscaling factor linearly depends on the square root of the ratio between the FOV area and the average feature surface (Fig. 5f). Upscaling factor was adjusted in order to have the smallest feature in each image inscribed in a 32 pixels x 32 pixels box. This was because the smallest anchor box encoded in the network was 32 pixels x 32 pixels.

In order to optimize the upscaling factor, we systematically explored the effect of varying its value in all used datasets (Fig. 5a–e). We defined an optimal upscaling factor of 1 for the training and validation datasets (LIV and CA1 datasets). The optimal upscaling factor was 2 for the ABO dataset, and it was between 1.7 and 3.1 for the NF datasets. In the VPM dataset, the magnification factor of each image was altered as a function of the radial distance due to the optical properties of the corrected microendoscopes[49]. We corrected this distortion with an additional preprocessing step. The optimal upscaling factor for the corrected VPM dataset was 1.4. Each bounding box was associated with a score, representing network confidence in cell detection. Bounding boxes with intersection over union (IoU) <20% were considered as separate neuronal identities. When IoU of two bounding boxes was >20%, the bounding box with the highest score was retained. Results of the image detector were filtered by applying a threshold on the output score provided by the network and optimized for each dataset.

Performance was evaluated using Precision, Recall, and F-1 scores defined as follows:

$$Precision = TP/(TP + FP) \tag{1}$$

$$Recall = TP/(TP + FN) \tag{2}$$

$$F-1 = 2*(Precision*Recall)/(Precision + Recall) \tag{3}$$

$$mAP = <IoU> \tag{4}$$

where TP indicates true positive detections, FP indicates false positive detections, and FN indicates false negative detections. TP and FP detections were defined, according to the confusion matrix, as bounding boxes identified by CITE-On, which had (TP) or did not have (FP) a corresponding bounding box in the consensus GT (using a cut-off threshold of 0.5 on the surface overlap). FN were bounding boxes detected by CITE-On with no GT counterpart. IoU indicates the intersection over union.

**CITE-On online pipeline**. In the online pipeline, individual raw imaging frames were continuously grabbed from a streaming source (e.g., live microscope output) and processed. To simulate this process, we individually imported in the CITE-On pipeline each frame of each raw t-series. Single frames were passed on to the trace extractor and to a buffer. The buffer stored the number of frames sufficient to produce an average projection. Once the buffer was filled, the projection was computed, sent to the image detector, and the buffer emptied.

The parameter "frame downsampling" was defined as the number of imaging frames used by CITE-On online to calculate the local average (either in the sliding approach or the step average approach). The local average was then used as the input image for the CNN. In the offline pipeline, the input image for the CNN was calculated on all the frames of the t-series. In the online pipeline, the frame downsampling value determined the SNR of the input image to the CNN and therefore influenced performance (Supplementary Fig. 9). To set the frame downsampling value on new acquisitions, users should follow these steps: (i) based on the SNR of the acquisition under consideration, refer to Supplementary Fig. 9 to estimate a certain range of obtainable F-1 scores; (ii) use the estimated F-1 value to extract, from Fig. 3a, the optimal range of frame downsampling to use. Further optimization of the frame downsampling value may be performed offline (if this modality is compatible with the experimental design). Offline validation would also allow building the user's internal GT that may be then used to update the current model of CITE-On with additional training data and potentially increase detection performance.

Detections were performed using the same procedure described for the offline pipeline. Detection results were fed to a custom tracking algorithm detecting all the overlaps between current and previous detections, and designed in order to maximize the overlap between putative matching boxes. For every detection matching a previous one, the coordinates of the relative bounding box were updated to the last one. For each new detection having an IoU <25% with all the previous detections, a new identity was created. In case of identities not actively detected in the current frame, relative coordinates were updated using a rigid shift calculated as the mean shift obtained from the active identities. In this way, we aimed to minimize the effect of motion artifacts and identity switch without implementing online motion correction approaches. A simple dynamic segmentation was then performed for each identity. At each raw frame, the interval between the 80th and the 95th percentile of the pixel fluorescence intensity distribution inside each bounding box was averaged to extract the raw functional trace. At each frame, background signal corresponded to the average fluorescence of all the pixels in the FOV not belonging to any bounding box. This frame-wise value was subtracted from all the individual raw functional traces. In order to optimize real-time performance for high frame rate acquisitions (above 3 Hz, including all ABO and some NF t-series), the entire pipeline was implemented in a multiprocessing scheme where one process was responsible for data loading, one for image preprocessing, and one for sending its output to the CNN detector (accelerated over GPU) and tracking identities, while the remaining CPU cores (compatibly with imaging acquisition software requirements) were dedicated to real-time trace extraction given the parallel nature of the problem[60]. This implementation scheme allowed for cell detection update (up to 10 Hz) and

functional trace extraction update from all identities (100 Hz) to operate as parallel and asynchronous processes.

**Parameter exploration for the object detector.** To find the best operating parameters for the object detector, we quantified offline CITE-On performance while systematically exploring various plausible values of upscaling factor and score threshold. For all datasets, we evaluated the F-1 across a set of upscaling factors between 0.6 and 3.4 in steps of 0.2. We also explored score thresholds between 0.05 and 0.95, in steps of 0.05. This mapping strategy allowed us to define the optimal combination of score threshold and upscaling factor for each input data. The upscaling factor was dependent on the ratio between the average box surface and the whole FOV surface, while the score threshold presented nonobvious dependence on trivial image statistics. Therefore, we determined the upscaling factor according to the acquisition parameters (FOV area and mean bounding box area, Fig. 5f) and the relative score thresholds, in order to maximize the F-1 score for each dataset. For the online pipeline, we used the same upscaling factors utilized in the offline pipeline and proceeded by exploring the dependency of F-1 on the score threshold and on the number of averaged frames in each detection.

**Generation of artificial motion artifacts.** Artificial motion artifacts were generated on a representative t-series from the ABO dataset. The FOV was first cropped by 20 µm (i.e., the size of the maximal displacement tested) on each side to remove the black bands introduced by the shift. A parameter search was then run to determine the best combination of upscaling factor, number of averages, and score threshold for online analysis. We simulated a planar shift from left to right using an affine transformation with a translation matrix. Starting at the 10,000th frame of the acquisition, we gradually applied the shift at each frame following a linear profile in time ranging from 30 ms to 333.33 s with logarithmic sampling. This procedure was repeated for each value of the shift ranging between 4 and 20 µm. In Supplementary Fig. 4, we report Precision, Recall, and F-1 score values for the online pipeline run on the shifted and cropped t-series against the cropped version of the corresponding GT annotation.

**Trace extraction: comparison between CITE-On and other methods.** We compared CITE-On trace extraction with trace extraction performed with a popular state-of-the-art method based on CNMF, CaImAn[10]. Briefly, we provided binary masks corresponding to the CITE-On detected bounding boxes and used these masks to seed the CNMF algorithm. Seeded-CNMF first calculated the temporal background component using pixels that were not included in any mask. We compared this background component to the CITE-On background traces used for trace correction. The subsequent step of the seeded-CNMF algorithm estimated temporal components and spatial footprints, constrained to be non-zero only at the location of the binary masks. Using this strategy, we obtained fluorescent traces from putative cells detected in the same locations as those detected by CITE-On, allowing for a one-to-one trace comparison between algorithms. It is important to note that, for this analysis, we set the order of the autoregressive model of the CNMF to zero because we were not interested in trace deconvolution but only in correcting for background contamination. To better compare the correlation due to the trace true signal and reduce the noise contribution, after *bg* subtraction, we smoothed both the seeded-CNMF and CITE-On extracted traces using a Gaussian filter. For each dataset, the standard deviation of the Gaussian kernel was set to 1 frame.

**Local vs. global background signal correlation.** To compare local and global background noise contributions, we used the same approach for background noise subtraction but considering only the pixels in the vicinity of each cell. The vicinity of a cell was defined as all the pixels in a concentric rectangular box double the size of the box detected by CITE-On, with no overlap with other bounding boxes. We then calculated the cross-correlation at lag zero between the local noise for each cell and the global background trace.

**Detection of calcium events.** To detect calcium events in Supplementary Fig. 8, traces extracted with CITE-On were processed as follows: (i) each functional trace was filtered using a Savitzky-Golay filter, (second order, time bin size = 15 frames); (ii) the filtered trace was processed using the *scipy.find_peaks* function (prominence = 7, height = (3, None)) to find only positive-going peaks; (iii) the *scipy.find_peaks* function returned all time points at which a peak in the fluorescence was detected, according to the two parameters in (ii). Each detected fluorescence peak was defined as a calcium event.

**Computation of the SNR.** To compute the SNR of a t-series, we divided the average fluorescence intensity of all pixels by the standard deviation of the fluorescence intensity of all pixels. To compute the SNR of single ROIs, we divided the average fluorescence intensity of all pixels within the ROI by the standard deviation of the fluorescence intensity of the selected pixels. We ran this computation on background-subtracted traces extracted by either CITE-On or CaImAn.

**Computation of cross-correlation and dendrogram sorting.** We used an agglomerative clustering procedure, where error sum of squares function (i.e., Ward distance) is used to define the distance between couples of functional traces in order to define the hierarchical dendrogram[61,62]. Dendrogram-sorted correlation matrices shown in this work (Figs. 4d, e, 7f, 9d) were generated using the clustermap function in the Seaborn package (mwaskom/seaborn v0.9.0 (2018), https://doi.org/10.5281/zenodo.1313201). This function first computes the Manhattan distance between correlations for each pair of cells and then uses Ward linkage on the aforementioned distances to sort the correlation matrix.

**Tiled detection on mesoscopic images.** For large-scale datasets (1972 pixels x 1682 pixels) such as the mesoscopic imaging dataset[7], requiring large amounts of GPU memory (>500 GB), we implemented a tiled detection approach. We divided each mesoscopic FOV into a configurable number of tiles with a configurable overlap factor in order to batch process all tiles up to the limit of the available GPU memory. Once all detections were computed, they were appropriately shifted back to the original position in the FOV, and a non-maximum suppression[63] was performed in order to remove duplicate boxes in regions of the FOV where overlap between tiles occurred.

**Hardware and software for data analysis.** All the data analysis procedures presented in this work were performed on a Dell Precision 7920 desktop with an Intel Xeon Silver 4110 @ 2.1 GHz 8 core CPU, 32 GB DDR4-2666 ECC RAM, NVIDIA Quadro RTX5000 GPU, 512 GB NVMe SSD, and 2 TB 7200 rpm HDD.

All processing steps, including network training and validation, were carried out under Keras/Tensorflow software libraries[64]. Image processing and data analysis were carried out using Python Language ref. [65].

**Statistics.** Values were expressed as mean ± sd, unless otherwise stated. The number of samples ($N$) and $p$-values are reported in the figure legends or in the text. No statistical methods were used to pre-determine sample size, but the sample size was chosen based on the previous studies[3,66]. All recordings with no technical issues were included in the analysis, and blinding was not used in this study. Statistical analysis was performed with the scientific Python ecosystem (SciPy 1.4, NumPy 1.19) under Python 3.7, Python Software Foundation, Python Language ref. [65] (available at https://www.python.org). A Kolmogorov–Smirnov test was run on each experimental sample to test for normality. The significance threshold was always set at $p = 0.05$. When comparing two paired populations of non-normally distributed data, a two-sample Kolmogorov–Smirnov test or a Wilcoxon signed-rank test were used. For unpaired comparison, a Student's $t$-test or a Wilcoxon rank sum test was used for normally and non-normally distributed data, respectively. All tests were two-sided, unless otherwise stated.

**Reporting summary.** Further information on research design is available in the Nature Research Reporting Summary linked to this article.

## Data availability
All t-series of our datasets (LIV, CA1, and VPM), our GT annotations for our datasets and for external datasets, and the trained model are freely available at our institutional data repository at the following link: https://doi.org/10.48557/TRGQOD. Additional publicly available datasets used in this work can be found at the following links: Allen Brain Observatory (ABO) visual coding dataset (see "Methods" section for t-series identification numbers): https://observatory.brain-map.org/visualcoding—Neurofinder (NF) challenge dataset: https://github.com/codeneuro/neurofinder—Mesoscopic full field imaging t-series: https://github.com/sofroniewn/2pRAM-paper.

## Code availability
All the code presented in this work together with the trained network model and tutorial examples are freely available from our institutional GitLab server at the following link: https://gitlab.iit.it/fellin-public/cite-on.

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

## Acknowledgements

We thank Dr. C. Arlt and Dr. C. Harvey for sharing mesoscopic imaging data, Dr. A. Sattin for sharing VPM recordings, Dr. S. Fiorini and Dr. A. Barla for discussion and comments on algorithm development, Dr. D.S. Kim and the GENIE project and Dr. J.M. Wilson for constructs, and Dr. S. Succol for technical support. This work was supported by the ERC-CoG (NEURO-PATTERNS, 647725), NIH Brain Initiative (U19 NS107464), and H2020-ICT (DEEPER, 101016787). M.P. is a Marie Sklodowska-Curie fellow (EnlightenedLoom, 101024523).

## Author contributions

L.S., M.B., P.L.d.L.R. developed software and performed analysis. M.B., S.C. performed experiments. D.V. performed data curation. L.S., M.B., T.F. conceived the project. T.F. coordinated the project. L.S., M.B., M.P., T.F. wrote the paper with inputs from other authors. All authors approved the final version of the manuscript.

## Competing interests

The authors declare no competing interests.
