## [Peer Review File · Nature Communications]

A deep-learning approach for online cell identification and trace extraction in functional two-photon calcium imagingReviewer #3 replaced reviewer #1 the second round of review due their contribution to the initial report submitted by reviewer #1.

REVIEWER COMMENTS

Reviewer #1 (Remarks to the Author):

Summary

=====

In this paper the authors describe a pipeline for analysis of two-photon calcium imaging data, including cell detection, segmentation and trace extraction with background subtraction. They trained a publicly available neural network to identify individual neurons in images. Further, they used the regions identified by the image detector for trace extraction of neural activity. The whole pipeline is computationally light-weight and therefore easy to use online (during image acquisition). The approach is generalisable to datasets outside of their lab, but the cell detection performance becomes less accurate. This is a simple, novel and useful tool to implement as part of an experimental pipeline requiring real-time feedback of neural activity to inform further physical, electrical or optical stimulations. As the CITE-On approach is simple, it is easy to understand and has the potential to be very accessible for online analysis of two-photon calcium imaging data. However, the authors should carry out major revisions to improve the description of the approach and to provide further evidence to back up some of their claims, including minor new analyses of existing data. With these changes the paper will be clearer, have more impact, and the authors will have discussed the negative as well as the positive aspects of their approach.

Major remarks

=====

1 - Generalisability overstated or insufficiently quantified

The authors claim the method is generalisable, but CITE-On performance degrades when used on never-before-seen datasets, the authors have not quantified what characteristics caused the degradation in performance. The authors should:

a) quantify differences in datasets more thoroughly (e.g. by identifying factors such as the mean intensity, SNR, pixel size, injected vs. transgenic) and provide more data (e.g. a stack of all timeseries median projections with annotations from ground truth and CITE-On as in Fig 3b) AND/OR

b) tone down the claims of generalisability, clearly mentioning the degraded accuracy (supp table 4)

2 - Feasibility and ease of utility of CITE-On in a new lab environment insufficiently described
CITE-On is published as a tool which is generalisable and therefore useful to the community, however the feasibility, ease of utility and accessibility of the approach/code is insufficiently described. The authors should:

a) describe the code in more detail (e.g. what language is it written in, is it open access on GitHub, will it be a package/programme or just plain code, what level of programming expertise would a new lab require to utilise CITE-On?)

b) discuss the feasibility of using CITE-On on novel datasets during daily experiments (i.e. how will a new user set their upscaling/downsampling/score threshold parameters without any ground truth performance comparisons?)

c) openly discuss the impact that poorer CITE-On performance would have on the accuracy of experiments requiring online analysis (i.e. if 20-30% of cells are missed, experiments will be systematically biased), including poorer performance at the start of a timeseries (i.e. requiring substantial time prior to experiment initiation)

3 - Insufficient quantification of robustness to motion artifacts

Robustness to motion correction is unsubstantiated and/or unproven. The authors show that their data has very minor motion artifacts (supp fig 3), yet claim that CITE-On does not require motion correction.

a) the claim should be toned down

OR

b) data that specifically has a large amount of motion artifacts should be used to compare performance with and without motion correction (Fig 3D)

AND

c) supp movie 4 should be repeated for a cell during motion to show whether the trace extraction is affected while the bounding box is in the process of updating

4 - Comparison with other approaches is insufficiently detailed and stated unfairly

To ensure a fair comparison was conducted with other state-of-the-art methods, and to ensure it can be repeated, the authors should:

a) state the parameters used for alternative methods (offline and online) and whether they were optimised

b) explain how calcium events were detected for supp fig 7, and include direct comparisons (number of calcium events, trace and ROI) of the same cells between each method

c) clarify the discussion of CITE-On and OnACID in lines 479-496 (i.e. CITE-On requires tuning of downsampling/upscaling/score threshold, these are not mentioned as 'initialisation parameters', it also performs most accurately after ~30% of the timeseries similar to OnACID requiring 1000-3000 frames, further description of CITE-On is given without comparison to OnACID for each point)

d) the generalisation of OnACID is discussed in comparison to CITE-On, but there is no evidence that OnACID can not generalise, therefore this specific argument should be toned down

e) clarify the differences (or lack of) between OnACID and CaImAn online, one name is used in the results, and the other is used in the discussion

5 - Further explanation, interpretation and guidance required for reader

Missing details and definitions make the paper difficult to read/understand and also impact on the ease of utility. The authors should:

a) describe the following object detection performance metrics in the main text before line 214 and expand on lines 956/957: F-1, Precision, Accuracy and Recall

b) define the parameters in more detail and discuss the logic used to make adjustments to them in the main text as they currently seem arbitrarily chosen without clear logic (i.e. 'frame downsampling' = temporal downsampling, linked to SNR and upscaling - but why is it linked, how does it help performance? 'image upscaling' = spatial upscaling, was this done because of RetinaNet image size input requirements, and why does FOV vs. neuron size matter for the performance?)

c) explain why sliding vs. step average helps with high vs. low SNR recordings, example images may help

d) explain why the arbitrary thresholds of the 80th and 95th percentile pixel values were used for trace extraction, first introduced on line 156-158

e) elaborate on why analyses were performed and what results show i.e. line 267-271 describes that images, annotations, traces, cross correlation matrices and dendrogram analysis (with and without background subtraction) are present in the figure, however there is no attempt made to help the reader understand how well the pipeline performed, what cross correlation matrices are for, what dendrogram analysis shows, how either of those analyses are carried out, why with and without background subtraction was looked at and how the results differed. Here is a non-exhaustive list of other places to improve: lines 307-309, 320-322, 344-348

Minor remarks

=====

GENERAL

- Colour-blind friendly colormaps would help with accessibility (all red/green combinations changed to magenta/green for images + bounding boxes)

- Mentions of CITE-On's performance is written in the past tense, but would read better in the present tense as CITE-On is capable of performing in the same way every time i.e. 'CITE-On also identifies inactive neurons'
- Line 33/34 statement is too strong, no attempt was made to determine if the functional measurements were accurate (i.e. no ground truth ephys)
- Line 35 needs clarity. As it is written, it sounds like the CITE-On online pipeline performs as well as other methods do offline. In the paper, no comparison was made between offline results and CITE-On online, so is this sentence alluding to the offline performance of CITE-On?
- Line 85/86 states the limited ability of current approaches to do online analysis, but no mention of CaImAn or OnACID in the introduction by name, they should be mentioned as current state-of-the-art online approaches and how they work specifically as the novelty of the paper relates directly to these existing approaches
- Line 88/89 is a weak argument, the approaches may not be validated on large FOVs, but it doesn't mean they can't perform well, did the authors try to validate them on large FOVs?
- Line 121-123: the description of 'offline' and 'online' analysis could be useful further up the text, those words are used in the abstract
- Line 129/130: a definition of intersection over union would be beneficial here, i.e. 'proportion of overlapping area out of the total area of both bounding boxes'
- Line 160-162: worth mentioning that this is an accepted method, and if not, why the authors did it
- Line 215-217: why were both graders' annotations used as the ground truth (union of the set)? A consensus ground truth should be those that are shared between grader 1 + 2, not the union of the set.
- Line 225-226: the use of GT annotations to train the RetinaNet model wasn't mentioned in the methods
- Line 239: 'absolute maximum was observed' is stated, but there is no data above 20 frames in Fig 3a, so the reader can't judge it as the maximum being observed
- Line 255: Figure 3e is not used to explain anything, the authors should add narrative to these results
- Line 275/275: in this case CITE-On was compared with CaImAn in a non-native mode (seeded), was there any comparison made between CaImAn online and CITE-On online, comparing against the same ground truth? If not, this would be the state-of-the-art comparison and should be done
- Line 290-293: the effect of trace SNR on trace extraction performance is explored (and compared to CaImAn), but not discussed anywhere else in the paper. It appears cells with low SNR can not have their traces reliably extracted (as both methods are not correlated), does this mean those cells with low SNR shouldn't be analysed online? The authors should discuss this
- Lines 313/314: it sounds like internal ground truth annotations were used for CITE-On but different ground truth annotations were used to compare to the other methods in Fig 7, is that true? If not, please edit for clarity. If so, this will become a major issue where internally generated ground truth annotations was not used for all methods.
- Line 339-341: ABO calcium event frequency distribution is described, but CITE-On-only calcium event frequency distribution is missing, it should also be provided for the reader to see the comparison made in the sentence
- Line 344: Fig 7g is compared to 7f, is it fair to compare dendrogram sorted correlation matrices between two different sized groups of cells across the same area? The authors should include a correlation matrix from a random set of bounding boxes of the same size as the smaller group to prove that this result doesn't occur by chance
- Line 395-398: CITE-On can be applied efficiently, but the authors should be honest that the accuracy/precision/recall/F-1 etc. is not good, additionally there is no opportunity to see any breakdown of the performance results from mesoscopic imaging, but this is seen as a novelty of the paper
- Line 402: the authors make a strong claim that CITE-On is accurate, however in some cases it has a precision in the region of 0.6. The authors should leave out the word 'accurate' here and directly compare CITE-On's accuracy to state-of-the-art methods below
- Line 435-437: while it is true that cell detection updates frame-by-frame, there is a lag between the real data and the cell detection bounding box updates (due to the sliding/step frame average). The authors should make this distinction for the reader, as cell positions are not updated 'frame-by-frame'

- Line 509: the authors make speculation of CITE-On only cells being low SNR, why not measure and report this in the results section?
- Line 854: the authors should mention somewhere here that ground truth annotation was used to train the network and used for the loss function
- Line 865: 'score threshold' is not described anywhere, the authors should add a few lines describing it
- Line 866/867: what is meant by 'smallest feature in each image', the authors should describe this further to help with the understanding of the upscaling factor
- Line 932-934: potentially move to/repeat in the main text as this is useful information to know

CLARIFICATIONS AND GRAMMAR

----- MAIN TEXT

- Line 44 definition: 't-series' as 'timeseries' the first time the word is used
- Line 44 clarity: 'heavy' is meant as 'large', 'uses up a lot of disk space', 'high bandwidth', 'requires a lot of storage space'
- Line 45 clarity: 'time and computational power' requires more accurate wording - how much?
- Line 46/47 clarity: 'truthful and reliable' implies that calcium imaging is truthful and reliable, but rather it is 'representative'
- Line 48 clarity: does 'staining' mean 'expression' or 'labelling'?
- Line 53 clarity: does 'static and dynamic' mean 'spatial and temporal'?
- Line 54 grammar: 'in case of' missing 'the' for 'in the case of'
- Line 60 grammar: 'associated to' = 'associated with'?
- Line 69-89 clarity: the first sentence is on motion artifacts, but most of the paragraph is really about online segmentation of neurons and not specifically motion artifacts, potentially re-order for clarity
- Line 81 clarity: describe background signal as neuropil fluorescence?
- Line 84 clarity: 500 x 500 um, not um^2 ?
- Line 84 grammar: missing 'of' in 'thousands frames'
- Line 92 clarity: 'light-weight' in what way
- Line 103 clarity: 'RetinaNet dedicated to the identification of neuronal somata' as a novice, this reads as though RetinaNet is setup for neuronal somata, consider restructuring the sentence
- Line 104 clarity: 'light-weight' in what way
- Line 111 clarity: does 'surface' means 'area'?
- Line 117 clarity: does 'surfaces' means 'areas'?
- Line 118 clarity: does 'dynamic' means 'frame-by-frame'?
- Line 124 clarity: can the authors correct for all 'planar' motion, or do they mean lateral motion was corrected?
- Line 136 clarity: 'average', what kind of average?
- Line 139/140 clarity: 'average projection' what kind of average across what axis?
- Line 162/163 grammar: comma placement should be after 'Moreover'
- Line 164 clarity: 'functional' is probably the wrong word here, potentially 'uncontaminated'?
- Line 169 clarity: 'Training of the image detector and ground truth generation' title suggestion as they are talked about in that order
- Line 180-183 clarity: make it clear the dataset was made internally
- Line 220-221 clarity: sounds like similar FOVs were grouped and all put in to only one of either the training or validation datasets, but doesn't currently read like that
- Line 239 clarity: extra word 'value'?
- Line 264 clarity: 'averaged' with what method?
- Line 273-275 clarity: were CaImAn and CITE-On used in the 'online' mode, or offline? Were both using motion-corrected data?
- Line 313 clarity: potentially include the word 'ref' to indicate that the authors are referring to reference 22, or mention the authors of ref 22 by name?
- Line 326 clarity: is 'ABO' a method as well as a dataset, I didn't see it introduced
- Line 329 grammar: 'Similarly to what described' missing the word 'is', 'to what is described'
- Line 329: description of dendrogram could be mentioned when dendrograms are first introduced, rather than later on here
- Line 331-332 clarity: supp fig 7 shows number of detected calcium events, not cells, and supp table 1 isn't related to this sentence, maybe supp table 3 was meant?

- Line 342 grammar: 'resulted' should be 'was'?
- Line 347/348 clarity: 'detections' here refers to 'calcium event detections'?
- Line 351 clarity: 'ran CITE-On online using out GT annotation' - do the authors mean comparing it to their GT annotations?
- Line 367 grammar: 'what' is meant to be 'that'?
- Line 371 grammar: 'light-weight of' rewording needed, 'applied to' instead of 'applied for'?
- Line 372 clarity: 'ref 7' or refer to authors names?
- Line 373 clarity: 1792 x 1682 pixels at 1 um pixel size isn't 4.8 mm x 4.8 mm, some clarity needed here?
- Line 411 clarity: what is meant by the word 'readiness' here?
- Line 412 clarity: remove 'images,'
- Line 431 clarity: 'CITE-On performed as state-of-the-art' should be 'CITE-On performed as well as'?
- Line 438 clarity: 'cost-effective' in what way? Computational or monetary?
- Line 444-454: repetition of previous paragraph, suggest removing one or merging them and removing repetition
- Line 461 clarity: repetition between 'CITE-On analyzed full mesoscopic images' and the previous sentence
- Line 462 grammar: 'image in subfields' should be 'in to'?
- Line 466 grammar: 'it' should be 'its'
- Line 467 grammar: 'thousands neurons' missing 'of'?
- Line 503 clarity: 'beard'?
- Line 504-506 clarity: sentence is confusing, last half needs rewording, maybe 'skews the detection of cells to those that are responsive...'?
- Line 728 grammar: 'what' should be 'that'
- Line 797-798 clarity: what is meant by 'manually split t-series including different FOVs in the datasets'?
- Line 818-819 clarity: '... as described for the training and validation dataset' add the word 'below' as it is yet to be described at this point in the methods
- Line 849 grammar: 'were' should be 'where'?
- Line 856/857: super-script the number -5 and -3?
- Line 866/867: 32 x 32 pixels, or < 6 x 6 pixels?
- Line 896 extra word: 'anyway'?
- Line 915 clarity: 'net size multipliers' == 'upscaling factors'?
- Line 921 clarity: which 'acquisition parameters'?
- Line 922 clarity: 'appropriate upscaling factors' based on what?
- Line 923 grammar: should 'from' be 'on' in both cases?
- Line 944 clarity: 'local noise' means 'local background'?
- Line 947 clarity: '($>3000 \text{ pixel}^2$)' is meant to be 3000 x 3000 pixels?
- Line 947 grammar: 'as' should be 'such as'?
- Line 949 grammar: 'in a' should be 'in to a'?
- Line 952 grammar: 'were' should be 'where'?

MAIN FIGURES AND LEGENDS

- Fig 1F extra scale bar not required?
- Line 987: 'acquisition' should be 'acquisitions'?
- Line 991/992: add comma between 'squares' and 'black'?
- Line 1003: this grayscale bar is actually in panel B, not C
- Line 1011/1012: the traces aren't shown in green, they are purple + orange?
- Fig 2 title: add 'cell detection' as this relates to that performance only?
- Fig 3 title: add 'cell detection' as above
- Line 1027: referring to panel B, the two FOVs are swapped, the middle is actually jRCaMP1a
- Line 1028: capitalisation of 'DET' not needed?
- Line 1028: 'the online detections', is this the result at the end of the t-series, mid-way or the start?
- Line 1030/1031: bring 'validation' and 'datasets' back together and put the datasets in parentheses?
- Line 1039: missing hyphen in 'not-motion corrected'
- Fig 4A/B/C: the cells in panel B/C do not look like any of the cells in A (left/right panels), are

they the cells highlighted? If so, why are the images different? Explain in figure legend

- Line 1047 clarity: mention that they were thresholded as well as background subtracted (i.e. the threshold was 80th-95th percentile intensity values)
- Line 1050 grammar: 'shown' should be 'shows'?
- Fig 5 title: include 'cell detection' as with Fig 2/3 titles
- Fig 6 title: again, include 'cell detection'
- Line 1061 grammar: should 'On Line' be 'online'?
- Fig 7E does not appear to be referenced in the text
- Line 1074 rewording to say 'not counted in the GT of ABO or STNeuroNET'?
- Line 1075-1077: there isn't a dendrogram shown, but cross correlation matrix looks to have been sorted by a dendrogram? Rewording required
- Fig 8 title: add 'cell detection'
- Line 1094: should 'SPM' be 'VPM'?
- Lines 1099/1100: there are no yellow squares, red + white instead?
- Fig 9C: the cells in each of the cropped images appear to be brighter than every cell around them, how were the crops made, are they a different average/substack? Report in the figure legend

SUPPLEMENTAL FIGURES AND LEGENDS

- Supp fig 1: SNR calculation method is not mentioned, is it mean/std throughout the paper, mention this in the methods?
- Supp fig 1 legend: underscore used instead of left parentheses
- Supp fig 1 legend: 'sharpened median projection (cyan)' mentioned, but not shown in the figure
- Supp fig 3 title: 'artefacts', but have used 'artifacts' in main text
- Supp fig 3C legend: should 'Percentage of total' be 'Distribution of total'?
- Supp fig4A-C fig6A-C: make the y-axis matched across the three panels in each figure, wasn't immediately obvious that the correlation was so good on panel A for both figures
- Supp fig 7E: The SD is so large compared to the mean for all of the samples that I don't think any of the statistical tests are providing much information, remove them? If the authors want to keep them, did they do a multiple test correction here?
- Supp table 3 + 4 titles: include 'cell detection' i.e. '...cell detection performance'?
- Supp table 4: Precision and Recall mean + SDs are replicates of each other, I don't think this is possible?
- Supp table 5 title: are the detections referring to 'calcium events' here? Mention that in the title and legend

Reviewer #2 (Remarks to the Author):

The manuscript by Sità et al. describes a new software suite aimed at analyzing two-photon calcium imaging data online at rates up to 100 Hz. The strategy uses neural network-based algorithm for fast automatic cell identification and segmentation, as well as lean strategies for cell identity tracking and trace extraction online. Neural network was trained on datasets acquired in the lab and carefully tested on data repository available from different sources. The performance is equal or superior to all other existing suites, suggesting this is a valuable addition to the available toolbox. I have however a few questions relative to the testing and performance.

- Cross-correlation of traces extracted by CITE-On pipeline and by the best-performing available pipeline CaImAn is used as an important indicator of the reliability of CITE-On. However, the main determinant for cross-correlation appears to be the SNR of the trace. While this may seem trivial at first, it somewhat invalidates the approach. Indeed, each signal is composed of uncorrelated noise at each pixel of the image (mostly shot-noise and instrumental noise) on top of correlated modulations in time and space (whether they are true signals or motion artefacts). Because the two algorithms have separate estimates of the fluorescence, uncorrelated noise for the same cell

and the same frame will depend on different pixel weights and should therefore remain mostly uncorrelated between the traces, the proportion of uncorrelated noise over trace true signal modulation (in amplitude and frequency of occurrence), the SNR, will dictate the overall cross-correlation. Thus, measuring cross-correlation without any normalization relative to the uncorrelated noise does not give any indication on the similarity of the CITE-On and CaImAn traces. Although high SNR gives excellent correlation, nothing warrants that true trace correlation is as good at lower signal to noise levels. This analysis should be redone accordingly by estimating the correlations of the denoised traces.

- It is not entirely clear from the manuscript how the online version of the pipeline works. The authors state that there is no seed needed. Does it mean that the neural network can identify cells from the first averaged frame without any pre-acquisition? It is stated that cells are gradually identified as they become active. If so, is the signal recalculated for the first frames online?
- What is the dependence of cell tracking and signal quality on the amplitude of brain motion? One may anticipate that if the movement exceeds the overlap criterion for cell box identity a new cell will be created. This could be tested on a synthetic set of data obtained by tempering with an experimental set, by adding known image shifts.

Point-by-point response to the referees' comments

We thank the Reviewers for their constructive comments. We extensively edited the manuscript and performed all the additional analyses and modifications required to address all of their comments. Based on the additional work suggested by the reviewers, we believe that we significantly improved our manuscript and greatly strengthened our findings. Please find below our point-by-point response to their comments. The original Reviewer's comments are shown in *italics*, while the authors' response is displayed in red. Page and line numbers refer to the revised version. Edits to the original manuscript are shown in red in the revised text.

Reviewer #1

Reviewers' comment:

Summary

In this paper the authors describe a pipeline for analysis of two-photon calcium imaging data, including cell detection, segmentation and trace extraction with background subtraction. They trained a publicly available neural network to identify individual neurons in images. Further, they used the regions identified by the image detector for trace extraction of neural activity. The whole pipeline is computationally light-weight and therefore easy to use online (during image acquisition). The approach is generalisable to datasets outside of their lab, but the cell detection performance becomes less accurate.

This is a simple, novel and useful tool to implement as part of an experimental pipeline requiring real-time feedback of neural activity to inform further physical, electrical or optical stimulations. As the CITE-On approach is simple, it is easy to understand and has the potential to be very accessible for online analysis of two-photon calcium imaging data. However, the authors should carry out major revisions to improve the description of the approach and to provide further evidence to back up some of their claims, including minor new analyses of existing data. With these changes the paper will be clearer, have more impact, and the authors will have discussed the negative as well as the positive aspects of their approach.

Author response: we thank the referee for the positive evaluation of our work. Below we detail the changes that we made to the manuscript to address all of the reviewer's concerns.

Reviewers' comment:

Major remarks

1 - Generalisability overstated or insufficiently quantified

The authors claim the method is generalisable, but CITE-On performance degrades when used on never-before-seen datasets, the authors have not quantified what characteristics caused the degradation in performance. The authors should:

a) quantify differences in datasets more thoroughly (e.g. by identifying factors such as the mean intensity, SNR, pixel size, injected vs. transgenic) and provide more data (e.g. a stack of all time series median projections with annotations from ground truth and CITE-On as in Fig 3b).

Author response: to address request (a) of the referee's comment, we now show how the performance of CITE-On offline and CITE-On online varies according to the following factors:
- mean fluorescence intensity (new Supplementary Fig. 9a-b),

- mean SNR (new Supplementary Fig. 9c-d),
- pixel size (new Supplementary Fig. 9e-f),
- viral vs transgenic expression (new Supplementary Fig. 9g-h).

CITE-On performance (both in the offline and the online modality) strongly depended on the mean SNR and, to a lower extent, on the mean fluorescence intensity. CITE-On performance did not show strong dependence on the pixel size, and the strategy used for expressing the calcium indicator. The description of these results is now presented at lines 425-429 and reporter here below:

“CITE-On performance (both in the offline and the online modality) strongly depended on the mean SNR and, to a lower extent, on the mean fluorescence intensity (Supplementary Fig. 9a-d). CITE-On performance did not show strong dependence on the pixel size, and the strategy used for expressing the calcium indicator (Supplementary Fig. 9e-h).”

Moreover, as requested by the referee, we now provide two new movies: Supplementary Movie 4 and 5. Supplementary Movie 4 contains the enhanced median projections (EMPs, see Methods) of all the t-series used for CITE-On training, together with the corresponding consensus ground truth annotations (magenta boxes). Supplementary Movie 5 shows the EMPs of all the t-series used for CITE-On validation, with both the corresponding consensus ground truth annotations (magenta) and the CITE-On offline detections (green).

Reviewers' comment:

AND/OR

b) tone down the claims of generalisability, clearly mentioning the degraded accuracy (supp table 4)

Author response: we edited the text to more clearly state the degradation in accuracy of CITE-On on never-before-seen datasets. The edited text reads:

Lines 487-491:

“However, we decided not to do so because: *i*) CITE-On would have likely inherited the annotation bias toward more active cells, which is shared by existing publicly available repositories; *ii*) by using public datasets exclusively for the validation of the CITE-On image detector (rather than for training, too), we avoided any chance of data leakage, and we demonstrated that CITE-On generalizes to never-before-seen data, although with lower performance.”

Lines 628-635:

“Finally, CaImAn online was tested on two datasets with rather homogenous acquisition parameters, and its application in different experimental conditions was not fully characterized¹⁰. Here, we demonstrate that CITE-On generalizes across indicators (i.e. GCaMP6s, GCaMP6f, GCaMP7f, and jRCaMP1a) and across data acquired in different brain regions and with different pixel size, SNR, and frame rate. CITE-On performance on never-before-seen data tended to be, however, lower. Thanks to the properties described above CITE-On is a flexible online analysis tool to apply in different experimental conditions.”

Reviewers' comment:

2 - Feasibility and ease of utility of CITE-On in a new lab environment insufficiently described
CITE-On is published as a tool which is generalisable and therefore useful to the community, however the feasibility, ease of utility and accessibility of the approach/code is insufficiently described. The authors should:

a) describe the code in more detail (e.g. what language is it written in, is it open access on GitHub, will it be a package/programme or just plain code, what level of programming expertise would a new lab require to utilise CITE-On?)

Author response: we appreciate the referee's comments. We now add the requested information at lines 862-868. Specifically:

“CITE-On is a Python library and is freely available, along with the trained network, on our institutional gitlab (<https://gitlab.iit.it/fellin-public/cite-on>). For ease of access, the repository is accompanied by sample Jupyter notebooks detailing every aspect of both the offline and the online pipelines, and containing guidelines for parameter optimization. For offline and simulated online analysis, a Jupyter instance (preferably with GPU acceleration) is needed in order to run the notebooks. For true online implementation, programming experience in Python is required only to interface CITE-On with the microscope acquisition software.”

Reviewers' comment:

b) discuss the feasibility of using CITE-On on novel datasets during daily experiments (i.e. how will a new user set their upscaling/downsampling/score threshold parameters without any ground truth performance comparisons?)

Author response: we added a paragraph to the discussion describing how a new user should utilize CITE-On and set parameters in the absence of ground truth annotation. The new text reads (lines 493-501):

“In the absence of GT annotations, CITE-On can be run in three consecutive steps (see also Methods). First, the optimal upscaling factor must be set. This step can be performed on the median projection of the motion-corrected t-series by manually adjusting the upscaling factor in order to have tight bounding boxes surrounding neuronal somata. The second step consists in gradually increasing the offline score threshold to decrease the number of false positives. The third step involves running a grid search to determine: *i*) the most appropriate number of averages, and *ii*) the online score threshold giving the highest F-1 score compared to the offline detection obtained in the first step described above. This three-step procedure is implemented in a Jupyter notebook, which is available in the online repository (<https://gitlab.iit.it/fellin-public/cite-on>).”

Reviewers' comment:

c) openly discuss the impact that poorer CITE-On performance would have on the accuracy of experiments requiring online analysis (i.e. if 20-30% of cells are missed, experiments will be systematically biased), including poorer performance at the start of a timeseries (i.e. requiring substantial time prior to experiment initiation)

Author response: we thank the referee for their comment. In the Discussion (lines 552-562), we further highlighted the limitations imposed by lower CITE-On performance at the beginning of the t-series and in the online modality vs. the offline modality. The new paragraph reads:

“It is important to underline that CITE-On performance is lower in the online modality than in the offline modality. Moreover, during online processing, CITE-On has on average lower performance in the initial third of each given t-series (Fig. 3), with F-1 scores ~ 0.6 . The F-1 score then plateaus in the second and third third of the t-series. Given that different datasets had different sampling rates, average length of the acquisitions, and considering the non-stationary activity of neurons, CITE-On required a different number of frames to reach plateau F-1 score across different datasets. This value was ~ 40000 frames for a t-series acquired at 30 Hz with SNR comparable to that of the ABO dataset, ~ 1200 frames for 30 Hz frame rate movie with SNR comparable to that of the NF_{test} dataset, ~ 260 frames for a 2.66 Hz frame rate movie with SNR comparable to that of the VPM dataset, ~ 150 frames for a 1.5 Hz frame rate movie with SNR comparable to that of the LIV dataset, and ~ 250 frames for a 3 Hz frame rate movie with SNR comparable to that of the CA1 dataset.”

Reviewers' comment:

3 - Insufficient quantification of robustness to motion artifacts

Robustness to motion correction is unsubstantiated and/or unproven. The authors show that their data has very minor motion artifacts (supp fig 3), yet claim that CITE-On does not require motion correction.

a) the claim should be toned down

OR

b) data that specifically has a large amount of motion artifacts should be used to compare performance with and without motion correction (Fig 3D)

Author response: to address the referee's point and explore the performance of CITE-On in the presence of larger motion artifacts, we created a set of t-series ($N = 90$) with artificial motion artifacts ranging between 4 μm and 20 μm . The artifacts were obtained using frame-by-frame lateral drifts from one ABO motion-corrected t-series (ABO #501271265). The consensus ground truth annotation was translated frame-by-frame according to the imposed artificial drift. Using this strategy, we were able to study CITE-On performance (Precision, Recall, and F-1 score) as a function of the amplitude and duration of the artificial shift (new Supplementary Fig. 4). We observed that the F-1 score was highest for lateral displacements $\leq 8 \mu\text{m}$ for all the values of motion artifact durations that we considered. In contrast, CITE-On performance decreased for displacements $> 8 \mu\text{m}$ and this drop was more evident for faster artifacts (< 2 s). Importantly, motion artifacts typically observed in awake, behaving mice, are $< 6 \mu\text{m}$ (Griffiths et al. Nat. Methods 2020, Greenberg and Kerr J Neurosci. Methods 2009, Dombek et al. Neuron 2007.). The results of these new simulations and analyses are reported at lines 275-283 and new Supplementary Fig. 4. The new text reads:

“To explore the performance of CITE-On in the presence of larger motion artifacts, we created a set of t-series ($N = 90$) with artificial motion artifacts ranging between 4 μm and 20 μm . The artifacts were obtained using frame-by-frame lateral drifts from one ABO motion-corrected t-series (ABO #501271265). The consensus GT annotation was translated frame-by-frame

according to the imposed artificial drift. Using this strategy, we were able to study CITE-On performance (Precision, Recall, and F-1 score) as a function of the amplitude and duration of the artificial shift (Supplementary Fig. 4). We observed that the F-1 score was highest for lateral displacements $\leq 8 \mu\text{m}$ for all the values of motion artifact duration that we considered. In contrast, CITE-On performance decreased for displacements $> 8 \mu\text{m}$ and this drop was larger for faster artifacts ($< 2 \text{ s}$).

A section in the Material and Methods related to this new analysis has also been added (lines 984-994). Specifically:

“Generation of artificial motion artifacts

Artificial motion artifacts were generated on a representative t-series from the ABO dataset. The FOV was first cropped by $20 \mu\text{m}$ (i.e., the size of the maximal displacement tested) on each side to remove the black bands introduced by the shift. A parameter search was then run to determine the best combination of upscaling factor, number of averages, and score threshold for online analysis. We simulated a planar shift from left to right using an affine transformation with a translation matrix. Starting at the 10000th frame of the acquisition, we gradually applied the shift at each frame following a linear profile in time ranging from 30 ms to 333.33 s with logarithmic sampling. This procedure was repeated for each value of the shift ranging between 4 and $20 \mu\text{m}$. In Supplementary Fig. 4, we report Precision, Recall, and F-1 score values for the online pipeline run on the shifted and cropped t-series against the cropped version of the corresponding GT annotation.”

Reviewers’ comment:

AND

c) supp movie 4 should be repeated for a cell during motion to show whether the trace extraction is affected while the bounding box is in the process of updating

Author response: to address the referee’s request, we replaced the t-series in Supplementary movie 4 (now Supplementary Movie 6) with an acquisition from the CA1 validation dataset. The new t-series had motion artifacts larger than those in the example previously shown. In the new movie, the tracking efficiency for the detected identities, as well as the effect of motion artifacts on trace extraction, can be better appreciated. The functional trace in panel d shows several downward transients, which appear when the cell is maximally displaced with respect to the position of the corresponding bounding box (see also panel b). Downward deflections most likely appear when pixels with low SNR, belonging to the region surrounding each cell, are included in the box. These artifacts are relatively small in amplitude and do not prevent the identification of large and upward fluorescence transients.

Reviewers’ comment:

4 - Comparison with other approaches is insufficiently detailed and stated unfairly

To ensure a fair comparison was conducted with other state-of-the-art methods, and to ensure it can be repeated, the authors should:

a) state the parameters used for alternative methods (offline and online) and whether they were optimised

Author response: we thank the referee for their comment and we apologize if we were not sufficiently clear in the previous version of the manuscript. We did not run alternative methods (STNeuroNET, CaImAn online, CaImAn batch, Suite2P, HNCorr, Unet2DS) ourselves. Rather, we used the data reported in (Soltanian-Zadeh et al. PNAS 2019) for all alternative approaches. We decided not to run all the alternative methods because it would have required extensive parameter optimization for each method: a work already presented in (Soltanian-Zadeh et al. PNAS 2019). Details of the preprocessing steps, as well as of the optimization of each parameter for all alternative methods, can thus be found in the Appendix and related Supplementary methods of (Soltanian-Zadeh et al. PNAS 2019). We modified the text to clarify our strategy at lines 359-368 and we point to the Appendix and related Supplementary methods of (Soltanian-Zadeh et al. PNAS 2019) for details about parameter optimization of alternative methods. The new text reads:

“We compared the offline detection performance of CITE-On (Supplementary Table 3) with state-of-the-art alternative segmentation approaches such as STNeuroNET²², CaImAn online¹⁰, CaImAn batch¹⁰, Suite2P¹⁶, HNCcorr¹⁹, UNet2DS²³ on the ABO and NF datasets provided in reference²². To this aim, we did not run the alternative approaches ourselves. Rather, we used the data reported in reference²² for all of them. To carry out this comparison, CITE-On was run on the GT annotations provided in reference²². Detection performance (Precision, Recall, and F-1 score) is similar or better for CITE-On (both online and offline) when compared with detection performance reported on the same GT for CaImAn (CaImAn online and CaImAn batch) and most other algorithms (Fig. 6). Moreover, CITE-On performance (online and offline) using our consensus GT tends to be higher than CITE-On performance computed using the GT provided in reference²².”

Reviewers' comment:

b) explain how calcium events were detected for supp fig 7, and include direct comparisons (number of calcium events, trace and ROI) of the same cells between each method

Author response: to explain how calcium events were detected, we added the following paragraph to the text (lines 1020-1026):

“Detection of calcium events

To detect calcium events in Supplementary Fig. 8, traces extracted with CITE-On were processed as follows: *i)* each functional trace was filtered using a Savitzky-Golay filter, (second order, time bin size = 15 frames); *ii)* the filtered trace was processed using the `scipy.find_peaks` function (prominence = 7, height = (3, None) to find only positive-going peaks); *iii)* the `scipy.find_peaks` function returned all time points at which a peak in the fluorescence was detected, according to the two parameters in (ii). Each detected fluorescence peak was defined as a calcium event.”

In Supplementary Fig. 8, we did not use multiple methods to identify cells. Rather, we ran CITE-On on a representative ABO t-series and computed the distribution of calcium events in true positives cells using three different ground truth annotations (the ABO annotation, Supplementary Fig. 8a, available in the ABO repository; the STN annotation, Supplementary Fig. 8b, available in Soltanian-Zadeh et al. PNAS 2019; our own annotations, Supplementary Fig. 8c). We then plotted the distribution of calcium events in boxes that were detected by CITE-On and which were not included as true positives in any of the three annotations (Supplementary Fig. 8d). Surprisingly,

these identities were not completely silent, but displayed calcium events (Supplementary Fig. 8d). The purpose of Supplementary Fig. 8 was then to show that some of the identities detected by CITE-On, and defined as false positives according to all the annotations used, could still display calcium activity. We modified the text at lines 394-401 to better clarify our aim. The text reads:

“Moreover, the distribution of calcium events *per* detection for CITE-On only cells (Supplementary Fig. 8d) shows that some identities were silent (as expected), but also that a large fraction of them displayed detectable activity (91 % displayed at least one calcium event and 69 % showed at least ten calcium events in the whole ABO dataset). The number of detected calcium events with CITE-On (all detections, true positives, false positives, and CITE-On only identities), ABO (true and false positives), and with STNeuroNET (true and false positives) is reported in Supplementary Fig. 8e. These data indicate that the identities captured exclusively by CITE-On were mostly active neurons.”

Reviewers' comment:

c) clarify the discussion of CITE-On and OnACID in lines 479-496 (i.e. CITE-On requires tuning of downsampling/upscaling/score threshold, these are not mentioned as 'initialisation parameters', it also performs most accurately after ~30% of the timeseries similar to OnACID requiring 1000-3000 frames, further description of CITE-On is given without comparison to OnACID for each point)

Author response: we modified the indicated paragraph as suggested by the referee. Please note that we now refer to CaImAn online rather than OnACID as *per* our response to one of the next points of this referee. The new paragraph reads (lines 599-635):

“CITE-On has some similarities but also several differences compared to CaImAn online¹⁰. First, both CITE-On online and CaImAn online require the optimization of some initialization parameters¹⁰. In CITE-On, initialization parameters (e.g., upscaling factor and score threshold) are set knowing the dimension and the resolution of the target t-series. In contrast, CaImAn online requires the user to provide input parameters such as number of expected components, maximum number of neurons added *per* frame, threshold on SNR for accepting new identities¹⁰. These can be difficult to estimate simply based on the acquisition settings, and are usually optimized in repeated rounds of offline segmentation. Second, CaImAn online requires a preprocessing step for offline segmentation, which typically runs on 1000-3000 frames¹⁰. The quality of this initial offline segmentation has a large impact on the subsequent online processing. To obtain a reliable (convergent) segmentation using CaImAn online, it is often necessary to run multiple rounds of offline segmentations with different initialization parameters¹⁰ and the result is subordinated to the level of cellular activity¹⁰. Once the offline segmentation converges, CaImAn online starts the online analysis, updating ROIs (in terms of position, shape, and newly identified identities) with a temporal lag that is generally larger than the 2-4 minutes required for a single iteration of offline segmentation preprocessing¹⁰. CITE-On reduces this initial lag. After this time window, frames were processed in real time. Third, in the online modality, CITE-On does not require the correction of motion artifacts when lateral displacements are within 4 $\mu\text{m/s}$ (Supplementary Fig. 4). In contrast, CaImAn online requires a frame-by-frame motion correction routine that is fast and efficient (on average 5 ms per frame¹⁰), but is dependent on the surface (i.e. number of pixels) of the FOV. The correction of motion artifacts using CaImAn online may thus introduce significant

delays when processing large FOVs (e.g., mesoscopic imaging data). Fourth, CITE-On performs tracking, dynamic segmentation, and functional trace extraction at 100 Hz independently of the number of detected neurons and their activity. This feature allows maintaining high online performance on FOVs characterized by large numbers of neurons (e.g., those obtained from mesoscopic imaging) and sparse activity. Frame-by-frame processing with CaImAn online, instead, depends on the number of ROIs to be updated or added on the basis of the initial offline segmentation¹⁰. Fifth, CITE-On does not use local pixel correlation for cell identification, which may be advantageous when separating nearby synchronous cells. In contrast, the CaImAn online fast deconvolution approach may be more efficient in separating adjacent cells with different temporal profiles of fluorescence emission¹⁰. Finally, CaImAn online was tested on two datasets with rather homogenous acquisition parameters, and its application in different experimental conditions was not fully characterized¹⁰. Here, we demonstrate that CITE-On generalizes across indicators (i.e. GCaMP6s, GCaMP6f, GCaMP7f, and jRCaMP1a) and across data acquired in different brain regions and with different pixel size, SNR, and frame rate. CITE-On performance on never-before-seen data tended to be, however, lower. Thanks to the properties described above CITE-On is a flexible online analysis tool to apply in different experimental conditions.”

Reviewers’ comment:

d) the generalisation of OnACID is discussed in comparison to CITE-On, but there is no evidence that OnACID can not generalise, therefore this specific argument should be toned down

Author response: we thank the referee for their comment. We edited the text to clarify that CaImAn online (see also response to the previous and following points) has not been tested under all the conditions CITE-On was tested on. This means that CITE-On generalizes across the different indicators, brain regions, and acquisition parameters described in the text, but does not mean that CaImAn online cannot generalize. The new text reads (lines 628-635):

“Finally, CaImAn online was tested on two datasets with rather homogenous acquisition parameters, and its application in different experimental conditions was not fully characterized¹⁰. Here, we demonstrate that CITE-On generalizes across indicators (i.e. GCaMP6s, GCaMP6f, GCaMP7f, and jRCaMP1a) and across data acquired in different brain regions and with different pixel size, SNR, and frame rate. CITE-On performance on never-before-seen data tended to be, however, lower. Thanks to the properties described above CITE-On is a flexible online analysis tool to apply in different experimental conditions.”

Reviewers’ comment:

e) clarify the differences (or lack of) between OnACID and CaImAn online, one name is used in the results, and the other is used in the discussion

Author response: in the text, we now refer only to CaImAn online (Giovannucci et al. eLife 2019), the software that extends and improves the OnACID algorithm (Giovannucci et al. NIPS 2017).

Reviewers’ comment:

5 - Further explanation, interpretation and guidance required for reader

Missing details and definitions make the paper difficult to read/understand and also impact on the ease of utility. The authors should:

a) describe the following object detection performance metrics in the main text before line 214 and expand on lines 956/957: F-1, Precision, Accuracy and Recall

Author response: we modified the main text and the methods as requested by the referee. Specifically:

1. at line 225, we list the metrics (Precision, Recall, and F-1 score) we used to evaluate detection performance.
2. we expanded the methods at lines 911-923. The new text reads:

“Performance was evaluated using Precision, Recall, and F-1 scores defined as follows:

$$\text{Precision} = \text{TP} / (\text{TP} + \text{FP}) \quad (\text{Eq. 1})$$

$$\text{Recall} = \text{TP} / (\text{TP} + \text{FN}) \quad (\text{Eq. 2})$$

$$\text{F-1} = 2 * (\text{Precision} * \text{Recall}) / (\text{Precision} + \text{Recall}) \quad (\text{Eq. 3})$$

$$\text{mAP} = \langle \text{IoU} \rangle \quad (\text{Eq. 4})$$

where TP indicates true positive detections, FP indicates false positive detections, and FN indicates false negative detections. TP and FP detections were defined, according to the confusion matrix, as bounding boxes identified by CITE-On, which had (TP) or did not have (FP) a corresponding bounding box in the consensus GT (using a cut-off threshold of 0.5 on the surface overlap). FN were bounding boxes detected by CITE-On with no GT counterpart. IoU indicates the intersection over union.”

Reviewers’ comment:

b) define the parameters in more detail and discuss the logic used to make adjustments to them in the main text as they currently seem arbitrarily chosen without clear logic (i.e. ‘frame downsampling’ = temporal downsampling, linked to SNR and upscaling - but why is it linked, how does it help performance? ‘image upscaling’ = spatial upscaling, was this done because of RetinaNet image size input requirements, and why does FOV vs. neuron size matter for the performance?)

Author response: we edited the text to add the definitions and clarifications requested by the reviewer. Specifically, at lines 933-945 and 874-896, we added:

“The parameter “frame downsampling” was defined as the number of imaging frames used by CITE-On online to calculate the local average (either in the sliding approach or the step average approach). The local average was then used as the input image for the CNN. In the offline pipeline, the input image for the CNN was calculated on all the frames of the t-series. In the online pipeline, the frame downsampling value determined the SNR of the input image to the CNN, and therefore influenced performance (Supplementary Fig. 9). To set the frame downsampling value on new acquisitions, users should follow these steps: *i*) based on the SNR of the acquisition under consideration, refer to Supplementary Fig. 9 to estimate a certain range of obtainable F-1 scores; *ii*) use the estimated F-1 value to extract, from Fig. 3a, the optimal range of frame downsampling to use. Further optimization of the frame downsampling value may be performed offline (if this modality is compatible with the experimental design). Offline validation would also allow building

the user's internal GT that may be then used to update the current model of CITE-On with additional training data, and potentially increase detection performance.”

“The parameter “upscaling factor” was defined as the geometric transformation of the input image, before it is fed to the CNN input layer. The score threshold was defined as the minimum value of score needed for each box to be considered as true. The upscaling factor retained the original aspect ratio of the input image, while the absolute size of all image features changed (e.g. neuronal somata). In RetinaNET³⁴, a set of anchor boxes were used to predict the size and position of the bounding box for an object, independently from the input image size. Moreover, each location on a given feature map in RetinaNet had nine anchor boxes (at three scales and three ratios). The relationship between the anchor boxes and the dimension of the features on which the CNN performed the detection (i.e., neuronal somata) was therefore important. The upscaling process was set out to optimize this relationship and improve detection performance. We did not perform an *ab initio* training of a CNN because of the lack of suitably large two-photon datasets annotated for neuronal somata morphology. Rather, because RetinaNET is trained on millions of natural images, we opted for a transfer learning approach. This strategy prevented the modification of the anchor boxes size. The upscaling factor to be used for each input image (or groups of images with features of similar size) was empirically defined. Specifically, while exploring a range of upscaling factors and score thresholds, we used a grid search approach aimed to maximize the F-1 score (Fig. 5a-e). This optimization step can be refined for any new dataset containing features of size different from those used in this study. Alternatively, new CITE-On users can optimize their upscaling factor by using the simple empirical relation described in Fig. 5f. Indeed, the upscaling factor linearly depends on the square root of the ratio between the FOV area and the average feature surface (Fig. 5f). Upscaling factor was adjusted in order to have the smallest feature in each image inscribed in a 32 pixels x 32 pixels box. This was because the smallest anchor box encoded in the network was 32 pixels x 32 pixels.”

In the discussion (lines 592-597), we added:

“The result of the regression process, performed by the CNN, on the anchor boxes against putative position and size of a cell, determines the dependence of CITE-On performance on neuronal size (and on the upscaling factor value). When the upscaling factor is optimized, the size of the anchor boxes matches that of the feature(s) to detect, and the detection process ends in good agreement with the GT annotation. On the other hand, if the upscaling factor is not optimized, the sizes of the anchor boxes and the features to detect do not match, and lower performance is expected.”

In the Methods section (lines 898-904), we added:

“In order to optimize the upscaling factor, we systematically explored the effect of varying its value in all used datasets (see Fig. 5a-e). We defined an optimal upscaling factor of 1 for the training and validation datasets (LIV and CA1 datasets). The optimal upscaling factor was 2 for the ABO dataset, and it was between 1.7 and 3.1 for the NF datasets. In the VPM dataset, the magnification factor of each image was altered as a function of the radial distance, due to the optical properties of the corrected microendoscopes⁴⁹. We corrected for this distortion with an additional preprocessing step. The optimal upscaling factor for the corrected VPM dataset was 1.4.”

Reviewers' comment:

c) explain why sliding vs. step average helps with high vs. low SNR recordings, example images may help

Author response: to address the reviewer's request, we added the following paragraph (lines 580-590):

“Since online CITE-On detection efficiency increased with increasing SNR of the input image (Supplementary Fig. 9), we decided to feed the CNN with images resulting from averaging a subset of frames from the running t-series, rather than individual frames. This approach increased the detectability of neuronal somata (see Supplementary Fig. 1c-d) because the average across frames reduced uncorrelated noise emerging from each individual frame. In order to process data online, we opted for a sliding average approach. Here, the input image fed to the CNN was obtained by averaging the last n frames of the t-series, and it was updated at every new incoming frame. The input image was then appropriately upsampled and processed as described for the offline pipeline. In the presence of large FOVs and small pixel size, the upscaling process could be slower than the time required for the calculation of the sliding average. In this case, a step average approach was used, where the input image was computed on blocks of n frames and updated every n frames.”

Reviewers' comment:

d) explain why the arbitrary thresholds of the 80th and 95th percentile pixel values were used for trace extraction, first introduced on line 156-158

Author response: we now provide the information requested by the referee and modified the text as follows (lines 164-169):

“The distribution of fluorescence values inside each bounding box was computed at each frame (Fig. 1f, left). Only pixels with values between the 80th and the 95th percentile of the box's fluorescence distribution were assigned to the ROI corresponding to the cell soma (white pixels of the binary mask in Fig. 1f, right). This range of values was chosen in order to base trace extraction on the pixels with highest intensity ($> 80^{\text{th}}$ percentile), while avoiding pixels close to saturation ($< 95^{\text{th}}$ percentile).”

Reviewers' comment:

e) elaborate on why analyses were performed and what results show i.e. line 267-271 describes that images, annotations, traces, cross correlation matrices and dendrogram analysis (with and without background subtraction) are present in the figure, however there is no attempt made to help the reader understand how well the pipeline performed, what cross correlation matrices are for, what dendrogram analysis shows, how either of those analyses are carried out, why with and without background subtraction was looked at and how the results differed. Here is a non-exhaustive list of other places to improve: lines 307-309, 320-322, 344-348

Author response: we thank the referee for pointing our attention to unclear sections of the manuscript. We extensively edited the portions of the text indicated by the reviewer. Specifically: lines 294-312 now read:

“Bounding boxes detected by CITE-On on a representative LIV t-series and a representative CA1 t-series are shown in Fig. 4a. Representative fluorescence traces extracted by CITE-On on the two t-series are displayed in Fig. 4b-c. Fig. 4d-e shows the cross correlation matrix (lower left triangle) and the dendrogram analysis (upper right triangle) of all the identified cells before (Fig. 4d-e left) and after (Fig. 4d-e, right) *bg* subtraction. We found that the dendrogram sorting showed blocks with different cross correlation values for various subgroups of cells. Cross correlation matrices (see Methods) before *bg* subtraction displayed substantially larger values of average correlation compared to cross correlation matrices after *bg* subtraction (panel d: 0.60 ± 0.14 before subtraction vs. 0.05 ± 0.27 after subtraction, $p < 1E-9$, Wilcoxon signed-rank test, $N = 197$. Panel e: 0.33 ± 0.18 before subtraction vs. 0.08 ± 0.19 after subtraction, $p < 1E-9$, Wilcoxon signed-rank test, $N = 166$). Thus, *bg* subtraction reduced the overall pairwise correlations, as expected when a signal that is common to all neurons is subtracted. Therefore, by reducing the average value of pairwise correlations, *bg* subtraction allowed the identification of neuronal pairs with low cross correlation values (close to zero or < 0), which would be difficult to identify otherwise. The dendrogram sorting of neuronal identities (upper triangles of the cross correlation matrices in Fig. 4d-e) was based on the relative distance of their cross correlation values (see Methods). Comparing dendrograms before *bg* subtraction with dendrograms after *bg* subtraction highlights how subtracting the *bg* is instrumental to identify spatially localized clusters of cells with distinctive patterns of cross correlations (both positive and negative), which may be suggestive of functional neuronal ensembles.”

Lines 352-357 now read:

“The offline performance (defined in terms of Precision, Recall, and F-1 score) obtained using optimized upscaling factors for each dataset is shown in Fig. 5g and Supplementary Table 3. While CITE-On performance was high for most datasets, we observed lower performance for the NF_{train} dataset, in agreement with the observation that NF_{train} has among the lowest SNR of all considered datasets and that CITE-On performance decreases with decreasing SNR (Supplementary Fig. 5f).”

Lines 373-376 now read:

“As in our previous characterization on the validation dataset, cross correlation values of neuronal traces extracted with CITE-On and CaImAn increased with SNR (Supplementary Fig. 7f). This relationship between cross correlation and SNR can be ascribed to the larger amount of signal extracted by both CaImAn and CITE-On when the SNR is large.”

Lines 397-401 now read:

“The number of detected calcium events with CITE-On (all detections, true positives, false positives, and CITE-On only identities), ABO (true and false positives), and with STNeuroNET (true and false positives) is reported in Supplementary Fig. 8e. These data indicate that the identities captured exclusively by CITE-On were mostly active neurons.”

Reviewers' comment:

Minor remarks

GENERAL

- Colour-blind friendly colormaps would help with accessibility (all red/green combinations changed to magenta/green for images + bounding boxes)

Author response: thanks for this suggestion. It has been implemented in all figures.

Reviewers' comment:

- Mentions of CITE-On's performance is written in the past tense, but would read better in the present tense as CITE-On is capable of performing in the same way every time i.e. 'CITE-On also identifies inactive neurons'

Author response: done.

Reviewers' comment:

- Line 33/34 statement is too strong, no attempt was made to determine if the functional measurements were accurate (i.e. no ground truth ephys)

Author response: following the comment of the referee, we modified the indicated sentence.

The old sentence reads:

“CITE-On processes thousands of cells online, including data from mesoscopic two-photon imaging, and provides accurate functional measurements from most neurons in the FOV.”

The new sentence reads (lines 32-34):

“CITE-On processes thousands of cells online, including during mesoscopic two-photon imaging, and extracts functional measurements from most neurons in the FOV.”

Reviewers' comment:

- Line 35 needs clarity. As it is written, it sounds like the CITE-On online pipeline performs as well as other methods do offline. In the paper, no comparison was made between offline results and CITE-On online, so is this sentence alluding to the offline performance of CITE-On?

Author response: the referee is correct. The sentence refers to CITE-On performance offline and it has been modified to make it clear.

The old sentence reads:

“Applied to publicly available datasets, CITE-On achieves performance similar to that of state-of-the-art methods for offline analysis.”

The new sentence reads (lines 34-36):

“Applied to publicly available datasets, the offline version of CITE-On achieves a performance similar to that of state-of-the-art methods for offline analysis.”

Reviewers' comment:

- Line 85/86 states the limited ability of current approaches to do online analysis, but no mention of CaImAn or OnACID in the introduction by name, they should be mentioned as current state-of-the-art online approaches and how they work specifically as the novelty of the paper relates directly to these existing approaches

Author response: thanks for this comment. We now modified the indicated sentence (lines 82-89) as follows:

“As a result of all these analytical steps, a total processing time of 30 to 90 minutes was reported for most efficient methods when processing FOVs of about 500 μm x 500 μm containing hundreds of cells imaged over tens of thousands of frames^{10, 11, 22}. OnACID¹¹ and its extended version CaImAn online¹⁰ provide online analysis on streaming data. However, various rounds of offline segmentations with different initialization parameters¹⁰ need to be run in order to obtain a reliable segmentation. Moreover, detection performance is subordinated to the level of cellular activity¹⁰. These processes introduce a temporal lag that is generally larger than the few minutes required for a single iteration of offline segmentation preprocessing.”

Reviewers' comment:

- Line 88/89 is a weak argument, the approaches may not be validated on large FOVs, but it doesn't mean they can't perform well, did the authors try to validate them on large FOVs?

Author response: the paragraph the referee is referring to is reported here below (lines 89-94):

“Altogether, current analytical approaches are: *i*) still limited in their ability to perform online analysis, which is necessary for closed-loop experiments; *ii*) biased against the identification of rarely active or inactive cells, which could be as informative as more active neurons, for example, in longitudinal all-optical imaging and manipulation approaches; *iii*) not validated on large FOVs, such as those generated by mesoscopic two-photon imaging.”

With point *iii*) we do not mean to say that current approaches cannot perform well on large FOVs. Rather, that they haven't been validated on this type of data yet. We think our statement is fair and correct and we would suggest keeping the text as is.

Reviewers' comment:

- Line 121-123: the description of 'offline' and 'online' analysis could be useful further up the text, those words are used in the abstract

Author response: we agree with the referee and modified the text according to their suggestion. Specifically, at lines 96-102 we now write:

“Here, we describe CITE-On, a CNN-based algorithm trained to perform neuronal somata identification in two-photon imaging recordings, combined with a fast dynamic segmentation and trace extraction pipeline. CITE-On works both offline, after the acquisition is completed, or online and identifies hundreds to thousands of neuronal cell bodies in either modality. Moreover, CITE-

On identifies both active and inactive neurons, removing biases towards highly active cells. Finally, CITE-On's light architecture and processing strategy allows, for the first time, fast automatic segmentation, tracking, and trace extraction in mesoscopic two-photon imaging t-series."

Reviewers' comment:

- Line 129/130: a definition of intersection over union would be beneficial here, i.e. 'proportion of overlapping area out of the total area of both bounding boxes'

Author response: we agree with the reviewer and modified the indicated sentence as follows (lines 135-138):

"We defined the intersection over union (IoU) for two identified bounding boxes as the proportion of the overlapping area between two boxes out of the sum of the areas of the two boxes. Bounding boxes with intersection over union (IoU) < 20 % were considered as separate neuronal identities."

Reviewers' comment:

- Line 160-162: worth mentioning that this is an accepted method, and if not, why the authors did it

Author response: a similar global background was used in (Brondi et al. Cell Reports 2020). We now cite this previous study in the indicated sentence at lines 171-172:

"All the FOV pixels that were not included in any bounding box were assigned to a global background ROI, similarly to reference ³⁶."

Reviewers' comment:

- Line 215-217: why were both graders' annotations used as the ground truth (union of the set)? A consensus ground truth should be those that are shared between grader 1 + 2, not the union of the set.

Author response: we apologize with the referee for not being clear on this point in the original version of the manuscript. The consensus ground truth was obtained as follows:

Grader #1 and Grader #2 first annotated all the t-series independently. For each t-series, the two annotations were then overlapped and the two graders accepted all boxes that overlapped with a threshold > 0.5 of the intersection of the boxes' surface (GT_{overlap}). For boxes with overlap < 0.5, the two graders together first analysed the boxes only included by grader 1 ($GT_{\text{grader1only}}$) and then those included only by grader 2 ($GT_{\text{grader2only}}$). Boxes were retained when both graders were in agreement, otherwise the identity was rejected. This procedure generated two novel sets of boxes ($GT_{\text{grader1only_consensus}}$ and $GT_{\text{grader2only_consensus}}$). The final consensus GT was the one shared between the two graders, i.e., $GT_{\text{overlap}} + GT_{\text{grader1only_consensus}} + GT_{\text{grader2only_consensus}}$.

In the original version of the manuscript we called "ground truth" the union of the (consensus) annotations of the two graders. We now realize the text was unclear and partially incorrect. We modified the Methods (lines 826-838) to precisely detail the procedure described above and make

clear that the consensus ground truth was the one shared by both graders. We thank the referee for raising this point.

The added new text reads:

“Two graders independently labelled each EMP. LabelImg (<http://github.com/tzutalin/labelImg>) was used to define a single object class by manually drawing bounding boxes around every visible cell soma in the EMP. The surface of each bounding box was manually defined in order to tightly surround the cell shape. Boxes were allowed to overlap. Coordinates and surface of each bounding box for all EMPs were saved in a standard VOC format where each file reported the top left and bottom right coordinates (in pixels) for each bounding box. For each t-series, the annotations of the two graders were overlapped and the two graders accepted all boxes that overlapped with a threshold > 0.5 of the intersection of the boxes' surface (GT_{overlap}). For boxes with overlap < 0.5 , the two graders together first analyzed the boxes only included by grader 1 ($GT_{\text{grader1only}}$) and then those included only by grader 2 ($GT_{\text{grader2only}}$). Boxes were retained when both graders were in agreement, otherwise the identity was rejected. This procedure generated two novel sets of boxes ($GT_{\text{grader1only_consensus}}$ and $GT_{\text{grader2only_consensus}}$). The final consensus GT was the one shared between the two graders, i.e., $GT_{\text{overlap}} + GT_{\text{grader1only_consensus}} + GT_{\text{grader2only_consensus}}$.”

Reviewers' comment:

- Line 225-226: *the use of GT annotations to train the RetinaNet model wasn't mentioned in the methods*

Author response: we modified the Methods at lines 845-848 to specify what requested by the referee. Specifically:

“We used the consensus GT to train the RetinaNet model from its keras implementation (<https://github.com/fizyr/keras-retinanet>) using a transfer learning approach (i.e., starting with a model pre-trained on natural images), and achieving best performance after 17 epochs (validation mAP = 0.79).”

Reviewers' comment:

- Line 239: *'absolute maximum was observed' is stated, but there is no data above 20 frames in Fig 3a, so the reader can't judge it as the maximum being observed*

Author response: we agree with the referee. We removed the word “absolute” and replaced it with the word “local” at lines 255-256. The indicated sentence now reads:

“F-1 increased with n between 1 and 20 frames. At this latter value, a local maximum in F-1 was observed (Fig. 3a).”

Reviewers' comment:

- Line 255: *Figure 3e is not used to explain anything, the authors should add narrative to these results*

Author response: the following sentence was added (lines 272-273) to explain the results presented in Fig. 3e.

“Fig. 3e shows that the F-1 score increased during online processing and became stable within approximately $\frac{1}{3}$ of the total length of the processed t-series.”

Reviewers' comment:

- Line 275/275: in this case CITE-On was compared with CaImAn in a non-native mode (seeded), was there any comparison made between CaImAn online and CITE-On online, comparing against the same ground truth? If not, this would be the state-of-the-art comparison and should be done

Author response: at line 275 of the original manuscript (new Supplementary Fig. 5), we wanted to compare fluorescence trace extraction based on identities recognized by CITE-On to a state-of-the-art method for trace extraction, e.g., CaImAn. To this aim, we believe it is essential that the same identities are considered with the two methods and this is the main reason why we opted for the “seeded-CaImAn”. Our results show that functional traces extracted with CITE-On and CaImAn have high cross correlation values, validating the method of trace extraction, which was implemented in CITE-On.

The comparison of performance of CaImAn online and CITE-On online against the same ground truth is now shown in new Fig. 6. The text describing the new results reads (lines 364-368):

“Detection performance (Precision, Recall, and F-1 score) is similar or better for CITE-On (both online and offline) when compared with detection performance reported on the same GT for CaImAn (CaImAn online and CaImAn batch) and most other algorithms (Fig. 6). Moreover, CITE-On performance (online and offline) using our consensus GT tends to be higher than CITE-On performance computed using the GT provided in reference ²².”

Reviewers' comment:

- Line 290-293: the effect of trace SNR on trace extraction performance is explored (and compared to CaImAn), but not discussed anywhere else in the paper. It appears cells with low SNR can not have their traces reliably extracted (as both methods are not correlated), does this mean those cells with low SNR shouldn't be analysed online? The authors should discuss this

Author response: we thank the referee for this comment. To address it, we added the following paragraph to the discussion (lines 564-578):

“Cross correlation values between traces extracted with CITE-On and those extracted with seeded CaImAn depended on the trace SNR (i.e., correlations were lower for small values of SNR). This result may indicate that CITE-On is less accurate in extracting functional traces from low SNR cells. However, this effect may also be due to the reduced accuracy of CaImAn in extracting functional traces from low SNR cells ¹⁰. A way to discriminate between these possibilities would be to pair single-cell electrophysiology (to record the cell's spiking activity) with two-photon calcium imaging and test which method (CITE-On vs. CaImAn) extracts functional traces that best match the AP firing profile of cells with low SNR. However, low SNR cells are typically absent in current available datasets of combined imaging and electrophysiological measurements due to

the difficulty of performing imaging-guided electrophysiological recordings on low SNR cells. Although the pairwise correlations between low SNR cells extracted by CITE-On and CaImAn are low, we believe that these should not be removed from the CITE-On online analysis, for two reasons. First, excluding these identities altogether may lead to a substantial bias towards more active cells or cells with higher levels of indicator expression. Second, SNR is a dynamic property, and each cell may display both high and low SNR periods during chronic imaging.”

Reviewers’ comment:

- Lines 313/314: *it sounds like internal ground truth annotations were used for CITE-On but different ground truth annotations were used to compare to the other methods in Fig 7, is that true? If not, please edit for clarity. If so, this will become a major issue where internally generated ground truth annotations was not used for all methods.*

Author response: we thank the referee for giving us the opportunity to clarify the text and we apologise for not being clear in the previous version of the manuscript. In new Fig. 6, we present the performance of CITE-On (in both the offline and the online modalities) compared to those of the other tested methods (STNeuroNET, CaImAn online, CaImAn batch, Suite2P, HNCorr, Unet2DS) using the same ground truth annotations for all approaches (i.e., the one described in (Soltanian-Zadeh et al. PNAS 2019)). CITE-On performance (both offline and online) tended to be lower than that of STNeuroNET, while it was similar or higher than that of other methods.

In new Fig. 6, we also include the performance of CITE-On (offline and online) computed using our own consensus ground truth. We observed a slight increase in performance when our consensus ground truth, rather than the one provided in (Soltanian-Zadeh et al. PNAS 2019), was used. This result is in line with the observation that CITE-On detects neurons in the ABO dataset (Supplementary Fig. 8) that are not present in the set of true positives of (Soltanian-Zadeh et al. PNAS 2019). The difference in performance between CITE-On and STNeuroNET when different ground truth annotations are used, may therefore originate from this discrepancy in the annotations rather than from a systematic difference in processing efficiency. We think these additional data are valuable for the reader and we included them in the results.

We modified the text in the Results section (lines 359-368) to clarify our approach. The new text reads:

“We compared the offline detection performance of CITE-On (Supplementary Table 3) with state-of-the-art alternative segmentation approaches such as STNeuroNET²², CaImAn online¹⁰, CaImAn batch¹⁰, Suite2P¹⁶, HNCcorr¹⁹, UNet2DS²³ on the ABO and NF datasets provided in reference²². To this aim, we did not run the alternative approaches ourselves. Rather, we used the data reported in reference²² for all of them. To carry out this comparison, CITE-On was run on the GT annotations provided in reference²². Detection performance (Precision, Recall, and F-1 score) is similar or better for CITE-On (both online and offline) when compared with detection performance reported on the same GT for CaImAn (CaImAn online and CaImAn batch) and most other algorithms (Fig. 6). Moreover, CITE-On performance (online and offline) using our consensus GT tends to be higher than CITE-On performance computed using the GT provided in reference²².”

Reviewers’ comment:

- Line 339-341: ABO calcium event frequency distribution is described, but CITE-On-only calcium event frequency distribution is missing, it should also be provided for the reader to see the comparison made in the sentence

Author response: the distribution of calcium events frequency in CITE-On only cells is shown in Supplementary Fig. 8d. We modified the sentence at lines 394-397 to make this clear. The sentence now reads:

“Moreover, the distribution of calcium events *per* detection for CITE-On only cells (Supplementary Fig. 8d) shows that some identities were silent (as expected), but also that a large fraction of them displayed detectable activity (91 % displayed at least one calcium event and 69 % showed at least ten calcium events in the whole ABO dataset).”

Reviewers' comment:

- Line 344: Fig 7g is compared to 7f, is it fair to compare dendrogram sorted correlation matrices between two different sized groups of cells across the same area? The authors should include a correlation matrix from a random set of bounding boxes of the same size as the smaller group to prove that this result doesn't occur by chance

Author response: we agree with the Referee that the comparison is not fair between dendrogram sorted correlation matrices computed from different numbers of neurons. We thus removed the comparison by eliminating the matrix showing the dendrogram sorting of CITE-On only cells from Fig. 7h and by editing the text. We thank the Reviewer for raising this point.

Reviewers' comment:

- Line 395-398: CITE-On can be applied efficiently, but the authors should be honest that the accuracy/precision/recall/F-1 etc. is not good, additionally there is no opportunity to see any breakdown of the performance results from mesoscopic imaging, but this is seen as a novelty of the paper

Author response: we modified the text according to the suggestions of the referee. At lines 456-460, we now state:

“With this strategy, we achieved an online F-1 score of 0.54 (Precision: 0.77; Recall: 0.42) with a score threshold of 0.25 (quantified on four patches from the entire FOV). Although performance was lower in mesoscopic data compared to other datasets, these results demonstrate that CITE-On can be applied to fast processing of mesoscopic two-photon t-series with good efficiency.”

Reviewers' comment:

- Line 402: the authors make a strong claim that CITE-On is accurate, however in some cases it has a precision in the region of 0.6. The authors should leave out the word 'accurate' here and directly compare CITE-On's accuracy to state-of-the-art methods below

Author response: following the reviewer's comment, we remove the word “accurate” from line 464 (line 402 in the previous version of the manuscript).

Reviewers' comment:

- Line 435-437: while it is true that cell detection updates frame-by-frame, there is a lag between the real data and the cell detection bounding box updates (due to the sliding/step frame average). The authors should make this distinction for the reader, as cell positions are not updated 'frame-by-frame'

Author response: we appreciate the referee's request for clarification. We now added a paragraph (lines 528-539) to explain this point:

“CITE-On online works on individual images. These were either updated frame-by-frame after obtaining them as the result of a sliding average approach, or they were updated every n -frames, when a step average strategy was used. In both cases, CITE-On online has an initial lag in detecting identities, due to the time required to compute the first sliding average or step average. During this initial lag, detections are not available, and, if no previous detections had been computed, no functional trace extraction is performed. In the case of the LIV, CA1, VPM, and ABO datasets the initial lag was 6.6 s, 14 s, 7 s, and 0.3 s, respectively, using the sliding average approach. For the NF and mesoscope datasets, on which a step averaged approach was taken, a time window of respectively 28.5 s and 12.6 s was required for the computation to be performed. In both cases, the shape of each bounding box was updated every time active detections were updated. This process occurred in real time for the LIV, CA1, VPM and ABO datasets. In all cases, dynamic segmentation and functional trace extraction were performed at 100 Hz, which was faster than real time.”

Reviewers' comment:

- Line 509: the authors make speculation of CITE-On only cells being low SNR, why not measure and report this in the results section?

Author response: following the reviewer's comment, we now show the SNR values for all CITE-On only cells (new Supplementary Fig. 8f) and, compare them to those of all true positive cells. We found no significant difference between the medians of the two distributions. We now report this finding at lines 401-404 and removed the sentence indicated by the referee. The sentence in the Results reads:

“We observed no significant difference between the medians of the distribution of SNR values for CITE-On only cells and for all CITE-On TP cells (Supplementary Fig. 8f, Wilcoxon rank sum test, $p = 0.20$, $N = 439$ total number of CITE-On only cells and $N = 4934$ total number of CITE-On TP cells).”

Reviewers' comment:

- Line 854: the authors should mention somewhere here that ground truth annotation was used to train the network and used for the loss function

Author response: to address the referee's comment, we modified the text as follows (lines 855-860):

“The network was trained with a regression L1 loss function (Mean Absolute Errors, MAE, <https://rishy.github.io/ml/2015/07/28/11-vs-l2-loss/>) and with focal loss (<http://arxiv.org/abs/1708.02002>) using the Adam optimizer⁵⁹ with learning rate 10^{-5} and clipnorm 10^{-3} (<http://github.com/keras-team/keras/issues/510>)³⁴ modified by reducing the learning rate on loss plateau with a factor of 0.1. The network was trained for 17 epochs, each consisting of 1000 training steps of batch size 1.”

Reviewers' comment:

- Line 865: 'score threshold' is not described anywhere, the authors should add a few lines describing it

Author response: The score threshold is now defined in the Methods as follows (lines 876-877):

“The score threshold was defined as the minimum value of score needed for each box to be considered as true.”

Reviewers' comment:

- Line 866/867: what is meant by 'smallest feature in each image', the authors should describe this further to help with the understanding of the upscaling factor

Author response: we thank the referee for their request of clarification. To address it, we modified the text (lines 874-896) as follows:

“The parameter “upscaling factor” was defined as the geometric transformation of the input image, before it is fed to the CNN input layer. (...). The upscaling factor retained the original aspect ratio of the input image, while the absolute size of all image features changed (e.g. neuronal somata). In RetinaNET³⁴, a set of anchor boxes were used to predict the size and position of the bounding box for an object, independently from the input image size. Moreover, each location on a given feature map in RetinaNet had nine anchor boxes (at three scales and three ratios). The relationship between the anchor boxes and the dimension of the features on which the CNN performed the detection (i.e., neuronal somata) was therefore important. The upscaling process was set out to optimize this relationship and improve detection performance. We did not perform an *ab initio* training of a CNN because of the lack of suitably large two-photon datasets annotated for neuronal somata morphology. Rather, because RetinaNET is trained on millions of natural images, we opted for a transfer learning approach. This strategy prevented the modification of the anchor boxes size. The upscaling factor to be used for each input image (or groups of images with features of similar size) was empirically defined. Specifically, while exploring a range of upscaling factors and score thresholds, we used a grid search approach aimed to maximize the F-1 score (Fig. 5a-e). This optimization step can be refined for any new dataset containing features of size different from those used in this study. Alternatively, new CITE-On users can optimize their upscaling factor by using the simple empirical relation described in Fig. 5f. Indeed, the upscaling factor linearly depends on the square root of the ratio between the FOV area and the average feature surface (Fig. 5f). Upscaling factor was adjusted in order to have the smallest feature in each image inscribed in a 32 pixels x 32 pixels box. This was because the smallest anchor box encoded in the network was 32 pixels x 32 pixels.”

Reviewers' comment:

- Line 932-934: potentially move to/repeat in the main text as this is useful information to know

Author response: following the referee's comment, we now repeated in the main text (lines 319-322) that:

“Using this strategy, we obtained fluorescent traces extracted by CaImAn from putative cells detected in the same locations as those detected by CITE-On, allowing for a one-to-one trace comparison between algorithms.”

Reviewers' comment:

CLARIFICATIONS AND GRAMMAR

MAIN TEXT

Reviewers' comment: - Line 44 definition: 't-series' as 'timeseries' the first time the word is used

Author response: done (line 44).

Reviewers' comment: - Line 44 clarity: 'heavy' is meant as 'large', 'uses up a lot of disk space', 'high bandwidth', 'requires a lot of storage space'

Author response: the word “large” was used instead of “heavy” (line 44).

Reviewers' comment: - Line 45 clarity: 'time and computational power' requires more accurate wording - how much?

Author response: we now specify that 50 GB/h - 1 TB/h is needed for mesoscopic imaging (Giovannucci et al. eLife 2019) (line 45).

Reviewers' comment: - Line 46/47 clarity: 'truthful and reliable' implies that calcium imaging is truthful and reliable, but rather it is 'representative'

Author response: we removed the word “truthful and reliable” (line 47).

Reviewers' comment: - Line 48 clarity: does 'staining' mean 'expression' or 'labelling'?

Author response: we substituted the word “staining” with “labelling” (line 48).

Reviewers' comment: - Line 53 clarity: does 'static and dynamic' mean 'spatial and temporal'?

Author response: yes. The sentence has been changed according to the Referee's suggestion (line 53).

Reviewers' comment: - Line 54 grammar: 'in case of' missing 'the' for 'in the case of'

Author response: fixed (line 54).

Reviewers' comment: - Line 60 grammar: 'associated to' = 'associated with'?

Author response: fixed (line 60).

Reviewers' comment: - Line 69-89 clarity: the first sentence is on motion artifacts, but most of the paragraph is really about online segmentation of neurons and not specifically motion artifacts, potentially re-order for clarity

Author response: we thank the Referee for this comment. The indicated paragraph is about the challenges of tracking cells online. One of these challenges is the presence of motion artifacts and it is introduced and discussed first. We think this is fair and would prefer to keep the text as is.

Reviewers' comment: - Line 81 clarity: describe background signal as neuropil fluorescence?

Author response: fixed (line 81).

Reviewers' comment: - Line 84 clarity: 500 x 500 um, not um^2?

Author response: we refer to the area of the FOV, i.e. 500 μm x 500 μm = 500 x 500 μm^2 . We now indicate 500 μm x 500 μm (line 84).

Reviewers' comment: - Line 84 grammar: missing 'of' in 'thousands frames'

Author response: fixed (line 84).

Reviewers' comment: - Line 92 clarity: 'light-weight' in what way

Author response: we replaced the words "light-weight" with the word "fast" (line 97).

Reviewers' comment: - Line 103 clarity: 'RetinaNet dedicated to the identification of neuronal somata' as a novice, this reads as though RetinaNet is setup for neuronal somata, consider restructuring the sentence

Author response: thanks for this comment. We edited the text according to the Referee's suggestion. The new sentence reads (line 108-110):

"..an image detector based on the publicly available convolutional neural network (CNN) RetinaNet³⁴ used to identify neuronal somata and a custom-built downstream fast analysis pipeline, designed for functional trace extraction"

Reviewers' comment: - Line 104 clarity: 'light-weight' in what way

Author response: we replaced the words "light-weight" with the word "fast" (line 109).

Reviewers' comment: - Line 111 clarity: does 'surface' means 'area'?

Author response: yes. The text has been edited according to the reviewer's suggestion (line 116).

Reviewers' comment: - Line 117 clarity: does 'surfaces' means 'areas'?

Author response: yes, same as previous point (line 117).

Reviewers' comment: - Line 118 clarity: does 'dynamic' means 'frame-by-frame'?

Author response: yes. The word “dynamic” has been substituted with the word “frame-by-frame” (line 124).

Reviewers' comment: - Line 124 clarity: can the authors correct for all 'planar' motion, or do they mean lateral motion was corrected?

Author response: we meant only for lateral motion. The sentence has been edited to make this clear (line 130).

Reviewers' comment: - Line 136 clarity: 'average', what kind of average?

Author response: the arithmetic mean of the n considered frames. The indicated sentence has been edited to clarify the point raised by the Reviewer (line 144).

Reviewers' comment: - Line 139/140 clarity: 'average projection' what kind of average across what axis?

Author response: the arithmetic mean of the n considered frames along the temporal axis. The indicated sentence has been edited to clarify the point raised by the Reviewer (lines 148-149).

Reviewers' comment: - Line 162/163 grammar: comma placement should be after 'Moreover'

Author response: fixed (line 173).

Reviewers' comment: - Line 164 clarity: 'functional' is probably the wrong word here, potentially 'uncontaminated'?

Author response: we now use the word “bg-corrected” (line 175).

Reviewers' comment: - Line 169 clarity: 'Training of the image detector and ground truth generation' title suggestion as they are talked about in that order

Author response: thanks, the title has been modified according to the Referee's suggestion (line 180).

Reviewers' comment: - Line 180-183 clarity: make it clear the dataset was made internally

Author response: we edited the text to clarify that the dataset was made internally. The new text reads:

“We thus decided to use a dedicated dataset including only t-series acquired in our laboratory for training and internal validation.” Lines 191-193.

Reviewers’ comment: - Line 220-221 clarity: sounds like similar FOVs were grouped and all put in to only one of either the training or validation datasets, but doesn’t currently read like that

Author response: we edited the text to address the Reviewer's comment (lines 232-234).

Old sentence: “To avoid data leakage and to decrease overfitting, t-series from the same or similar FOVs were included in either the training dataset or the validation dataset.”

New sentence:

“To avoid data leakage and to decrease overfitting³⁸, the t-series relative to a given FOV were first grouped together, and then included only in the training dataset or the validation dataset.”

Reviewers’ comment: - Line 239 clarity: extra word 'value'?

Author response: We edited the indicated sentence (lines 255-256).

Old sentence:

“F-1 increased with n between 1 and 20 frames, value at which the absolute maximum was observed (Fig. 3a).”

New sentence:

“F-1 increased with n between 1 and 20 frames. At this latter value, a local maximum in F-1 was observed (Fig. 3a).”

Reviewers’ comment: - Line 264 clarity: 'averaged' with what method?

Author response: the arithmetic mean. The indicated sentence has been edited to clarify the point raised by the Reviewer (line 291).

Reviewers’ comment: - Line 273-275 clarity: were CaImAn and CITE-On used in the 'online' mode, or offline? Were both using motion-corrected data?

Author response: CaImAn and CITE-On were used offline and on motion corrected data. This is now specified at lines 315-316.

Reviewers’ comment: - Line 313 clarity: potentially include the word 'ref' to indicate that the authors are referring to reference 22, or mention the authors of ref 22 by name?

Author response: done (line 361).

Reviewers' comment: - Line 326 clarity: is 'ABO' a method as well as a dataset, I didn't see it introduced

Author response: thanks for this comment. We added a sentence to explain that the ABO dataset, besides the raw data, also contains the coordinates of segmentation performed on that dataset (lines 795-797).

Reviewers' comment: - Line 329 grammar: 'Similarly to what described' missing the word 'is', 'to what is described'

Author response: fixed (line 384).

Reviewers' comment: - Line 329: description of dendrogram could be mentioned when dendrograms are first introduced, rather than later on here

Author response: Done at line 308, in the paragraph where dendrograms are first introduced.

Reviewers' comment: - Line 331-332 clarity: supp fig 7 shows number of detected calcium events, not cells, and supp table 1 isn't related to this sentence, maybe supp table 3 was meant?

Author response: we apologize for the mistake. The indicated sentence referred to Supplementary Table 5. This has now been fixed (line 387).

Reviewers' comment: - Line 342 grammar: 'resulted' should be 'was'?

Author response: the indicated sentence was removed from the text (see response to point regarding line 344 in the "Minor Comments general list" of this referee).

Reviewers' comment: - Line 347/348 clarity: 'detections' here refers to 'calcium event detections'?

Author response: No, it refers to detection of cells. We modified the indicated sentence to make this clear.

Old sentence:

"The number of detections in the different datasets is reported in Supplementary Table 5"

New sentence (lines 404-405):

"The average number of cell detections in the different datasets is reported in Supplementary Table 5"

Reviewers' comment: - Line 351 clarity: 'ran CITE-On online using out GT annotation' - do the authors mean comparing it to their GT annotations?

Author response: correct. We edited the text to clarify this point (lines 408-409).

Old sentence:

“We ran CITE-On online using our GT annotation on each frame of the ABO, NF, and VPM datasets.”

New sentence:

“We ran CITE-On online and compared the results to our consensus GT annotation on each frame of the ABO, NF, and VPM datasets.”

Reviewers' comment: - Line 367 grammar: 'what' is meant to be 'that'?

Author response: fixed (line 425).

Reviewers' comment: - Line 371 grammar: 'light-weight of' rewording needed, 'applied to' instead of 'applied for'?

Author response: both fixed (lines 432-433).

Old sentence:

“Given the speed and light-weight of the CITE-On architecture, we tested if it could be applied for detecting cells in the mesoscopic imaging t-series described in 7.”

New sentence:

“Given the speed of the CITE-On architecture, we tested if it could be applied to detect cells in the mesoscopic imaging t-series described in Sofroniew et al. ⁷.”

Reviewers' comment: - Line 372 clarity: 'ref 7' or refer to authors names?

Author response: we now refer to the name (line 432, see also previous point).

Reviewers' comment: - Line 373 clarity: 1792 x 1682 pixels at 1 um pixel size isn't 4.8 mm x 4.8 mm, some clarity needed here?

Author response: the referee is correct. We now modified the indicated values and inserted the corrected ones, which are: 1792 pixels x 1682 pixels corresponding to 4.3 mm x 4.0 mm (0.42 pixels / um) (line 434).

Reviewers' comment: - Line 411 clarity: what is meant by the word 'readiness' here?

Author response: we meant “availability”. We edited the text to make this clear (lines 472-473).

Old sentence:

“On one side, this choice was justified by RetinaNET’s excellent performance in object recognition and by its readiness.”

New sentence:

“On one side, this choice was justified by RetinaNET’s excellent performance in object recognition and by its availability”

Reviewers’ comment: - Line 412 clarity: remove 'images,'

Author response: fixed.

Reviewers’ comment: - Line 431 clarity: 'CITE-On performed as state-of-the-art' should be 'CITE-On performed as well as'?

Author response: fixed (lines 503-504).

Old sentence:

“CITE-On performed as state-of-the-art algorithms ^{10, 11, 16, 19, 22, 39} on publicly available datasets”

New sentence:

“CITE-On performed similarly to state-of-the-art algorithms ^{10, 11, 16, 19, 22, 40} on publicly available datasets...”

Reviewers’ comment: - Line 438 clarity: 'cost-effective' in what way? Computational or monetary?

Author response: we meant from a computational point of view. We edited the indicated sentence to make this clear.

Old sentence:

“Third, once bounding boxes were identified in individual frames, we used a simple cost-effective strategy to extract pixels belonging to neuronal ROIs based on pixel’s brightness.”

New sentence (lines 509-511):

“Third, once bounding boxes were identified in individual frames, we used a simple computationally effective strategy to extract pixels belonging to neuronal ROIs based on their brightness.”

Reviewers' comment: - Line 444-454: repetition of previous paragraph, suggest removing one or merging them and removing repetition

Author response: we substantially trimmed the second paragraph and merged it with the first one.

Reviewers' comment: - Line 461 clarity: repetition between 'CITE-On analyzed full mesoscopic images' and the previous sentence

Author response: the repetition was removed.

Reviewers' comment: - Line 462 grammar: 'image in subfields' should be 'in to'?

Author response: fixed (line 520).

Reviewers' comment: - Line 466 grammar: 'it' should be 'its'

Author response: fixed (line 524).

Reviewers' comment: - Line 467 grammar: 'thousands neurons' missing 'of'?

Author response: fixed (line 526).

Reviewers' comment: - Line 503 clarity: 'beard'?

Author response: we meant “carried”. The indicated sentence now reads (line 641-643):

“Thus, being able to track cells regardless of their activity level is key, for instance, for investigating the sensory information carried by neurons that significantly change their activity throughout longitudinal imaging experiments.”

Reviewers' comment: - Line 504-506 clarity: sentence is confusing, last half needs rewording, maybe 'skews the detection of cells to those that are responsive...'?

Author response: fixed following the Referee's suggestion. The new sentence reads (lines 643-645):

“Biasing the cell identification toward active neurons, as currently done by most approaches, intrinsically skews the proportion of analyzed cells towards those that are responsive to a given stimulation in a certain brain region.”

Reviewers' comment: - Line 728 grammar: 'what' should be 'that'

Author response: fixed (line 714).

Reviewers' comment: - Line 797-798 clarity: what is meant by 'manually split t-series including different FOVs in the datasets'?

Author response: we edited the text to make this clear (lines 784-786).

Old sentence:

“To avoid data leakage between training and validation datasets, we manually split t-series including different FOVs in the datasets.”

New sentence:

“To avoid data leakage between training and validation datasets, we grouped together t-series acquired from the same FOV and included these data either in the training or validation datasets.”

Reviewers' comment: - Line 818-819 clarity: '... as described for the training and validation dataset' add the word 'below' as it is yet to be described at this point in the methods

Author response: done (line 813).

Reviewers' comment: - Line 849 grammar: 'were' should be 'where'?

Author response: fixed (line 852).

Reviewers' comment: - Line 856/857: super-script the number -5 and -3?

Author response: done (line 858).

Reviewers' comment: - Line 866/867: 32 x 32 pixels, or < 6 x 6 pixels?

Author response: 32 pixels x 32 pixels (lines 895-896).

Reviewers' comment: - Line 896 extra word: 'anyway'?

Author response: the indicated word was removed.

Reviewers' comment: - Line 915 clarity: 'net size multipliers' == 'upscaling factors'?

Author response: yes. We edited the sentence to clarify (lines 973-974).

Reviewers' comment: - Line 921 clarity: which 'acquisition parameters'?

Author response: This is now indicated in the edited text (lines 978-980), which reads:

“Therefore, we determined the upscaling factor according to the acquisition parameters (FOV area and mean bounding box area, Fig. 5f) and the relative score thresholds, in order to maximize the F-1 score for each dataset.”

Reviewers' comment: - Line 922 clarity: 'appropriate upscaling factors' based on what?

Author response: we edited the sentence to make it clear. It now reads (lines 980-982):

“For the online pipeline, we used the same upscaling factors utilized in the offline pipeline and proceeded by exploring the dependency of F-1 on the score threshold and on the number of averaged frames in each detection.”

Reviewers' comment: - Line 923 grammar: should 'from' be 'on' in both cases?

Author response: agreed and fixed (lines 981-982).

Reviewers' comment: - Line 944 clarity: 'local noise' means 'local background'?

Author response: yes, fixed (line 1018).

Reviewers' comment: - Line 947 clarity: '($>3000 \text{ pixel}^2$)' is meant to be 3000 x 3000 pixels?

Author response: yes, 3000 pixel^2 meant 3000 pixels x 3000 pixels. However, the correct numbers are 1792 pixels x 1682 pixels. The text has been amended to correct for this mistake (line 434).

Reviewers' comment: - Line 947 grammar: 'as' should be 'such as'?

Author response: fixed (line 1045).

Reviewers' comment: - Line 949 grammar: 'in a' should be 'in to a'?

Author response: fixed (line 1047).

Reviewers' comment: - Line 952 grammar: 'were' should be 'where'?

Author response: fixed (line 1051).

MAIN FIGURES AND LEGENDS

Reviewers' comment:

- Fig 1F extra scale bar not required?

Author response: done, extra scale bar removed.

Reviewers' comment:

- Line 987: 'acquisition' should be 'acquisitions'?

Author response: fixed.

Reviewers' comment:

- Line 991/992: add comma between 'squares' and 'black'?

Author response: done.

Reviewers' comment:

- Line 1003: this grayscale bar is actually in panel B, not C

Author response: fixed.

Reviewers' comment:

- Line 1011/1012: the traces aren't shown in green, they are purple + orange?

Author response: fixed.

Reviewers' comment:

- Fig 2 title: add 'cell detection' as this relates to that performance only?

Author response: done.

Reviewers' comment:

- Fig 3 title: add 'cell detection' as above

Author response: done.

Please note that we found a mistake in the positioning of some of the bounding boxes in Fig. 3c and Supplementary Fig. 2, 6. Bounding boxes in those figures have now been corrected.

Reviewers' comment:

- Line 1027: referring to panel B, the two FOVs are swapped, the middle is actually jRCaMP1a

Author response: thank for pointing our attention to this issue. In each figure, we now reports jRCaMP1a before GCaMP6f and we corrected all the corresponding figure legends.

Reviewers' comment:

- Line 1028: capitalisation of 'DET' not needed?

Author response: we removed capitalization as requested by the referee.

Reviewers' comment:

- Line 1028: 'the online detections', is this the result at the end of the t-series, mid-way or the start?

Author response: it refers to the end of the t-series. This is now stated at line 1294.

Reviewers' comment:

- Line 1030/1031: bring 'validation' and 'datasets' back together and put the datasets in parentheses?

Author response: done.

Reviewers' comment:

- Line 1039: missing hyphen in 'not-motion corrected'

Author response: fixed.

Reviewers' comment:

- Fig 4A/B/C: the cells in panel B/C do not look like any of the cells in A (left/right panels), are they the cells highlighted? If so, why are the images different? Explain in figure legend

Author response: thanks for pointing our attention to this mistake. In the previous version of the figure, we erroneously assigned cells shown in b-c to the FOV in a. We now updated panels a-c to fix the mistake.

Reviewers' comment:

- Line 1047 clarity: mention that they were thresholded as well as background subtracted (i.e. the threshold was 80th-95th percentile intensity values)

Author response: done.

Reviewers' comment:

- Line 1050 grammar: 'shown' should be 'shows'?

Author response: fixed

Reviewers' comment:

- Fig 5 title: include 'cell detection' as with Fig 2/3 titles

Author response: done.

Reviewers' comment:

- Fig 6 title: again, include 'cell detection'

Author response: done.

Reviewers' comment:

- Line 1061 grammar: should 'On Line' be 'online'?

Author response: fixed.

Reviewers' comment:

- Fig 7E does not appear to be referenced in the text

Author response: fixed.

Reviewers' comment:

- Line 1074 rewording to say 'not counted in the GT of ABO or STNeuroNET'?

Author response: done.

Reviewers' comment:

- Line 1075-1077: there isn't a dendrogram shown, but cross correlation matrix looks to have been sorted by a dendrogram? Rewording required

Author response: The referee is correct. Fixed.

Reviewers' comment:

- Fig 8 title: add 'cell detection'

Author response: done.

Reviewers' comment:

- Line 1094: should 'SPM' be 'VPM'?

Author response: fixed.

Reviewers' comment:

- Lines 1099/1100: there are no yellow squares, red + white instead?

Author response: fixed.

Reviewers' comment:

- Fig 9C: the cells in each of the cropped images appear to be brighter than every cell around them, how were the crops made, are they a different average/substack? Report in the figure legend

Author response: cells in the cropped area look brighter because we picked the brightest cell in each FOV. This is now stated in the legend (lines 1375-1376).

SUPPLEMENTAL FIGURES AND LEGENDS

Reviewers' comment:

- *Supp fig 1: SNR calculation method is not mentioned, is it mean/std throughout the paper, mention this in the methods?*

Author response: the method we used to compute the SNR is now stated in the methods (at lines 1028-1033. Specifically:

"Computation of the (SNR)

To compute the SNR of a t-series, we divided the average fluorescence intensity of all pixels by the standard deviation of the fluorescence intensity of all pixels. To compute the SNR of single ROIs, we divided the average fluorescence intensity of all pixels within the ROI by the standard

deviation of the fluorescence intensity of the selected pixels. We ran this computation on background-subtracted traces extracted by either CITE-On or CaImAn.”

Reviewers' comment:

- *Supp fig 1 legend: underscore used instead of left parentheses*

Author response: fixed.

Reviewers' comment:

- *Supp fig 1 legend: 'sharpened median projection (cyan)' mentioned, but not shown in the figure*

Author response: we removed that statement from the legend.

Reviewers' comment:

- *Supp fig 3 title: 'artefacts', but have used 'artifacts' in main text*

Author response: fixed.

Reviewers' comment:

- *Supp fig 3C legend: should 'Percentage of total' be 'Distribution of total'?*

Author response: fixed.

Reviewers' comment:

- *Supp fig4A-C fig6A-C: make the y-axis matched across the three panels in each figure, wasn't immediately obvious that the correlation was so good on panel A for both figures*

Author response: fixed.

Reviewers' comment:

- *Supp fig 7E: The SD is so large compared to the mean for all of the samples that I don't think any of the statistical tests are providing much information, remove them? If the authors want to keep them, did they do a multiple test correction here?*

Author response: according to the referee's comment, we removed the statistical test in Supplementary Fig. 7e.

Reviewers' comment:

- *Supp table 3 + 4 titles: include 'cell detection' i.e. '...cell detection performance'?*

Author response: done.

Reviewers' comment:

- *Supp table 4: Precision and Recall mean + SDs are replicates of each other, I don't think this is possible?*

Author response: this was a mistake and it has been fixed. The correct values are now reported in the new Supplementary Table 4.

Reviewers' comment:

- Supp table 5 title: are the detections referring to 'calcium events' here? Mention that in the title and legend

Author response: In Supplementary Table 5, we refer to detection of cells. This is now clearly stated in the legend and in the table (lines 1518-1521).

Reviewer #2

Reviewers' comment:

The manuscript by Sità et al. describes a new software suite aimed at analyzing two-photon calcium imaging data online at rates up to 100 Hz. The strategy uses neural network-based algorithm for fast automatic cell identification and segmentation, as well as lean strategies for cell identity tracking and trace extraction online. Neural network was trained on datasets acquired in the lab and carefully tested on data repository available from different sources. The performance is equal or superior to all other existing suites, suggesting this is a valuable addition to the available toolbox. I have however a few questions relative to the testing and performance.

- Cross-correlation of traces extracted by CITE-On pipeline and by the best-performing available pipeline CaImAn is used as an important indicator of the reliability of CITE-On. However, the main determinant for cross-correlation appears to be the SNR of the trace. While this may seem trivial at first, it somewhat invalidates the approach. Indeed, each signal is composed of uncorrelated noise at each pixel of the image (mostly shot-noise and instrumental noise) on top of correlated modulations in time and space (whether they are true signals or motion artefacts). Because the two algorithms have separate estimates of the fluorescence, uncorrelated noise for the same cell and the same frame will depend on different pixel weights and should therefore remain mostly uncorrelated between the traces, the proportion of uncorrelated noise over trace true signal modulation (in amplitude and frequency of occurrence), the SNR, will dictate the overall cross-correlation. Thus, measuring cross-correlation without any normalization relative to the uncorrelated noise does not give any indication on the similarity of the CITE-On and CaImAn traces. Although high SNR gives excellent correlation, nothing warrants that true trace correlation is as good at lower signal to noise levels. This analysis should be redone accordingly by estimating the correlations of the denoised traces.

Author response: we agree with the point raised by the referee and we did what they suggested. We re-ran the analysis estimating pairwise cross-correlations between traces extracted by CITE-On and by seeded CaImAn after denoising. Denoising was performed by smoothing fluorescence values using a Gaussian filter with a 1 frame kernel standard deviation in each t-series. We chose the smallest kernel width in order to reduce the impact of the smoothing approach on the waveform of calcium events. Indeed, longer sliding averages would have resulted in spurious increase of cross-correlation values because of time dilation of fluorescence waveforms. This effect would have been particularly relevant for some of the used datasets, which were recorded at low frame rate (e.g., 2-5 Hz). New Supplementary Fig. 5 and 7 now show representative traces before and

after denoising and the pairwise cross-correlations among traces extracted by CITE-On and seeded CaImAn after denoising. Cross correlation values computed on denoised traces were comparable, if not slightly higher, compared to those previously obtained on non-denoised traces. We therefore conclude that reduction of uncorrelated noise did not alter the results and the conclusions relative to this section of the manuscript.

The description of these new results (lines 329-331 and 370-372) now replaces the old cross correlation analysis.

Reviewers' comment:

- It is not entirely clear from the manuscript how the online version of the pipeline works. The authors state that there is no seed needed. Does it mean that the neural network can identify cells from the first averaged frame without any pre-acquisition? It is stated that cells are gradually identified as they become active. If so, is the signal recalculated for the first frames online?

Author response: we thank the referee for their request for clarification and we apologize for not being sufficiently clear in the first version of the manuscript. To address the referee's comment regarding the necessity of pre-acquisition, we edited the text and added the following paragraphs to the manuscript:

Lines 528-539

“CITE-On online works on individual images. These were either updated frame-by-frame after obtaining them as the result of a sliding average approach, or they were updated every n -frames, when a step average strategy was used. In both cases, CITE-On online has an initial lag in detecting identities, due to the time required to compute the first sliding average or step average. During this initial lag, detections are not available, and, if no previous detections had been computed, no functional trace extraction is performed. In the case of the LIV, CA1, VPM, and ABO datasets the initial lag was 6.6 s, 14 s, 7 s, and 0.3 s, respectively, using the sliding average approach. For the NF and mesoscope datasets, on which a step averaged approach was taken, a time window of respectively 28.5 s and 12.6 s was required for the computation to be performed. In both cases, the shape of each bounding box was updated every time active detections were updated. This process occurred in real time for the LIV, CA1, VPM and ABO datasets. In all cases, dynamic segmentation and functional trace extraction were performed at 100 Hz, which was faster than real time.”

Lines 580-590

“Since online CITE-On detection efficiency increased with increasing SNR of the input image (Supplementary Fig. 9), we decided to feed the CNN with images resulting from averaging a subset of frames from the running t-series, rather than individual frames. This approach increased the detectability of neuronal somata (see Supplementary Fig. 1c-d) because the average across frames reduced uncorrelated noise emerging from each individual frame. In order to process data online, we opted for a sliding average approach. Here, the input image fed to the CNN was obtained by averaging the last n frames of the t-series, and it was updated at every new incoming frame. The input image was then appropriately upscaled and processed as described for the offline pipeline.

In the presence of large FOVs and small pixel size, the upscaling process could be slower than the time required for the calculation of the sliding average. In this case, a step average approach was used, where the input image was computed on blocks of n frames and updated every n frames.”

Regarding the question whether the signal was recalculated for the first frames online once a new detection was identified: CITE-On does not currently support this feature and functional traces start being extracted only when the associated cell is detected. Since CITE-On was designed for closed-loop experiments, we did not retrospectively update functional traces when all detections were identified. Rather, we relied on the constant updating of detections as the t-series progressed. We added the following sentence in the discussion (lines 539-541) to clarify this point.

“CITE-On did not retrospectively update functional traces corresponding to a newly identified ROI, and functional traces started being extracted only when the associated ROI was detected.”

Reviewers’ comment:

- What is the dependence of cell tracking and signal quality on the amplitude of brain motion? One may anticipate that if the movement exceeds the overlap criterion for cell box identity a new cell will be created. This could be tested on a synthetic set of data obtained by tempering with an experimental set, by adding known image shifts

Author response: to address the referee’s point and explore the performance of CITE-On in the presence of larger motion artifacts, we created a set of t-series ($N = 90$) with artificial motion artifacts ranging between $4 \mu\text{m}$ and $20 \mu\text{m}$. The artifacts were obtained using frame-by-frame lateral drifts from one ABO motion-corrected t-series (ABO #501271265). The consensus ground truth annotation was translated frame-by-frame according to the imposed artificial drift. Using this strategy, we were able to study CITE-On performance (precision, recall, and F-1 score) as a function of the amplitude and duration of the artificial shift (new Supplementary Fig. 4). We observed that the F-1 score was highest for lateral displacements $\leq 8 \mu\text{m}$ for all the values of motion artifact durations that we considered. In contrast, CITE-On performance decreased for displacements $> 8 \mu\text{m}$ and this drop was more evident for faster artifacts ($< 2 \text{ s}$). Importantly, motion artifacts typically observed in awake, behaving mice, are $< 6 \mu\text{m}$ (Griffiths et al. Nat. Methods 2020, Dombeck et al. Neuron 2007, Greenberg and Kerr J Neurosci. Methods 2009). The results of these new simulations and analyses are reported at lines 275-283 and new Supplementary Fig. 4. The new text reads:

“To explore the performance of CITE-On in the presence of larger motion artifacts, we created a set of t-series ($N = 90$) with artificial motion artifacts ranging between $4 \mu\text{m}$ and $20 \mu\text{m}$. The artifacts were obtained using frame-by-frame lateral drifts from one ABO motion-corrected t-series (ABO #501271265). The consensus GT annotation was translated frame-by-frame according to the imposed artificial drift. Using this strategy, we were able to study CITE-On performance (Precision, Recall, and F-1 score) as a function of the amplitude and duration of the artificial shift (Supplementary Fig. 4). We observed that the F-1 score was highest for lateral displacements $\leq 8 \mu\text{m}$ for all the values of motion artifact duration that we considered. In contrast, CITE-On performance decreased for displacements $> 8 \mu\text{m}$ and this drop was larger for faster artifacts ($< 2 \text{ s}$).”

A section in the Material and Methods related to this new analysis has also been added (lines 984-994). Specifically:

“Generation of artificial motion artifacts

Artificial motion artifacts were generated on a representative t-series from the ABO dataset. The FOV was first cropped by 20 μm (i.e., the size of the maximal displacement tested) on each side to remove the black bands introduced by the shift. A parameter search was then run to determine the best combination of upscaling factor, number of averages, and score threshold for online analysis. We simulated a planar shift from left to right using an affine transformation with a translation matrix. Starting at the 10000th frame of the acquisition, we gradually applied the shift at each frame following a linear profile in time ranging from 30 ms to 333.33 s with logarithmic sampling. This procedure was repeated for each value of the shift ranging between 4 and 20 μm . In Supplementary Fig. 4, we report Precision, Recall, and F-1 score values for the online pipeline run on the shifted and cropped t-series against the cropped version of the corresponding GT annotation.”

REVIEWERS' COMMENTS

Reviewer #2 (Remarks to the Author):

The revised version of the manuscript has been substantially improved.

Concerning the effect of uncorrelated noise on functional traces cross-correlation (now sup. Fig.5), one could have expected a slightly more quantitative approach, for example by measuring the uncorrelated noise in each trace through autocorrelation and then subtracting the estimated impact on the cross-correlation. However, the marginal change in the results that is induced by smoothing is a convincing demonstration that noise is not the factor dominating cross-correlation between functional traces obtained by different methods.

The rest of my concerns have been properly addressed.

Reviewer #3 (Remarks to the Author):

The amendments that the authors have made to the manuscript were thorough and have clarified their description of the approach, improving accessibility. I recommend publication without a further round of review, however, I think it would be beneficial for the authors to consider these few comments:

- Line 108-110 still sounds like RetinaNet is used to identify neuronal somata, consider 'image detector used to identify neuronal somata based on the publicly available convolutional neural network (CNN) RetinaNet'
- Supp movies 4 + 5 don't appear to have magenta ground truth annotation boxes

I thank the authors for responding to every comment, congratulate them on their development, and look forward to seeing the tool being used broadly.

Robert Lees

Point-by-point response to the referees' comments

Reviewer #2

Referee's comment: The revised version of the manuscript has been substantially improved. Concerning the effect of uncorrelated noise on functional traces cross-correlation (now sup. Fig.5), one could have expected a slightly more quantitative approach, for example by measuring the uncorrelated noise in each trace through autocorrelation and then subtracting the estimated impact on the cross-correlation. However, the marginal change in the results that is induced by smoothing is a convincing demonstration that noise is not the factor dominating cross-correlation between functional traces obtained by different methods. The rest of my concerns have been properly addressed.

Author response: we thank the referee for the positive evaluation of our work. Regarding Suppl. Fig. 5, we agree with them that “the marginal change in the results that is induced by smoothing is a convincing demonstration that noise is not the factor dominating cross-correlation between functional traces obtained by different methods”.

Reviewer #3 (Remarks to the Author):

Referee's comment: The amendments that the authors have made to the manuscript were thorough and have clarified their description of the approach, improving accessibility. I recommend publication without a further round of review, however, I think it would be beneficial for the authors to consider these few comments:

- Line 108-110 still sounds like RetinaNet is used to identify neuronal somata, consider 'image detector used to identify neuronal somata based on the publicly available convolutional neural network (CNN) RetinaNet'*
- Supp movies 4 + 5 don't appear to have magenta ground truth annotation boxes*

I thank the authors for responding to every comment, congratulate them on their development, and look forward to seeing the tool being used broadly.

Author response: we thank for the positive evaluation of the work, which we did during the revision. Regarding the two additional comments, we addressed them as follows:

- 1) We modified the sentence on page 107-109 as suggested by the referee. It now reads: “an image detector used to identify neuronal somata based on the publicly available convolutional neural network (CNN) RetinaNet³⁴ and a custom-built downstream fast analysis pipeline, designed for functional trace extraction.”*

- 2) We corrected the legend of Supplementary movies 4-5, which erroneously indicated the presence of magenta ground truth annotations in the movies. The ground truth annotations are shown in green. We modified the legends to fix the mistake.